# The Landscape of Agentic Reinforcement Learning for LLMs: A Survey

**Guibin Zhang**[3†]    **Hejia Geng**[1†]    **Xiaohang Yu**[8†]    **Zhenfei Yin**[1*]    **Zaibin Zhang**[9,1]
**Zelin Tan**[7,2]    **Heng Zhou**[7,2]    **Zhongzhi Li**[10]    **Xiangyuan Xue**[11,2]    **Yijiang Li**[13]
**Yifan Zhou**[12]    **Yang Chen**[2]    **Chen Zhang**[7]    **Yutao Fan**[2]    **Zihu Wang**[14]    **Songtao Huang**[6,2]
**Piedrahita-Velez, Francisco**[5]    **Yue Liao**[3]    **Hongru Wang**[11]    **Mengyue Yang**[9]
**Heng Ji**[4]    **Jun Wang**[6]    **Shuicheng Yan**[3]    **Philip Torr**[1]    **Lei Bai**[2*]

[1]University of Oxford    [2]Shanghai AI Laboratory    [3]National University of Singapore
[4]University of Illinois Urbana-Champaign    [5]Brown University    [6]University College London
[7]University of Science and Technology of China    [8]Imperial College London
[9]Dalian University of Technology [10]Chinese Academy of Sciences
[11]The Chinese University of Hong Kong    [12]University of Georgia
[13]University of California, San Diego    [14]University of California, Santa Barbara

[†] *Equal contribution*,    [*] *Corresponding Author*

Reviewed on OpenReview: **https://openreview.net/forum?id=RY19y2RI1O**

## Abstract

The emergence of agentic reinforcement learning (Agentic RL) marks a paradigm shift from conventional reinforcement learning applied to large language models (LLM RL), reframing LLMs from passive sequence generators into autonomous, decision-making agents embedded in complex, dynamic worlds. This survey formalizes this conceptual shift by contrasting the degenerate single-step Markov Decision Processes (MDPs) of LLM RL with the temporally extended Partially Observable Markov Decision Processes (POMDPs) that define Agentic RL. Building on this foundation, we propose a comprehensive twofold taxonomy: one organized around core agentic capabilities, including planning, tool use, memory, reasoning, self-improvement, and perception, and the other around their applications across diverse task domains. Central to our thesis is that reinforcement learning serves as the critical mechanism for transforming these capabilities from static, heuristic modules into adaptive, robust agentic behavior. To support and accelerate future research, we consolidate the landscape of open-source environments, benchmarks, and frameworks into a practical compendium. By synthesizing over five hundred recent works, this survey charts the contours of this rapidly evolving field and highlights the opportunities and challenges that will shape the development of scalable, general-purpose AI agents.

## 1 Introduction

The rapid convergence of large language models (LLMs) and reinforcement learning (RL) has precipitated a fundamental transformation in how language models are conceived, trained, and deployed. Early LLM RL paradigms largely treated these models as static conditional generators, optimized to produce single-turn outputs aligned with human preferences or benchmark scores. While successful for alignment and instruction-following, such approaches overlook the broader spectrum of sequential decision-making that underpins realistic, interactive settings. These limitations have prompted a shift in perspective: rather than viewing LLMs as passive text emitters, recent developments increasingly frame them as *Agents*, *i.e.*, autonomous decision-makers capable of perceiving, reasoning, planning, invoking tools, maintaining memory, and adapting

strategies over extended horizons in partially observable, dynamic environments. We define this emerging paradigm as **Agentic Reinforcement Learning (Agentic RL)**. To more clearly delineate the distinction between the concept of Agentic RL studied in this work and conventional RL approaches, we provide the following definition:

> **Agentic Reinforcement Learning (Agentic RL)** refers to a paradigm in which LLMs, rather than being treated as *static conditional generators* optimized for single-turn output alignment or benchmark performance, are conceptualized as *learnable policies* embedded within sequential decision-making loops, where RL endows them with autonomous agentic capabilities, such as planning, reasoning, tool use, memory maintenance, and self-reflection, enabling the emergence of long-horizon cognitive and interactive behaviors in *partially observable, dynamic environments.*

In Section 2, we present a more formal, symbolically grounded distinction between Agentic RL and conventional RL. Prior research relevant to Agentic RL can be broadly grouped into two complementary threads: **Synergy between RL and LLMs** and **LLM Agents**, detailed as follows:

**Synergy between RL and LLMs**  The second line of research investigates how reinforcement learning algorithms are applied to improve or align LLMs. A primary branch, RL for training LLMs, leverages on-policy (*e.g.*, proximal policy optimization (PPO) (Schulman et al., 2017) and Group Relative Policy Optimization (GRPO) (Shao et al., 2024b)) and off-policy (*e.g.*, actor–critic, Q-learning (Mnih et al., 2013)) methods to enhance capabilities such as instruction-following, ethical alignment, and code generation (Srivastava & Aggarwal, 2025; Wang et al., 2025m; 2024c). A complementary direction, LLMs for RL, examines the deployment of LLMs as planners, reward designers, goal generators, or information processors to improve sample efficiency, generalization, and multi-task planning in control environments, with systematic taxonomies provided by (Cao et al., 2025c). RL has also been integrated throughout the LLM lifecycle: from data generation (Guo et al., 2025b; Wan et al., 2025a) and pretraining (Dong et al., 2025a) to post-training and inference (Chow et al., 2025), as surveyed by (Guo & Wang, 2025). The most prominent branch here is post-training alignment, notably Reinforcement Learning from Human Feedback (RLHF) (Christiano et al., 2017), along with extensions such as Reinforcement Learning from AI Feedback (RLAIF) (Bai et al., 2022) and Direct Preference Optimization (DPO) (Rafailov et al., 2023; Wang et al., 2024j; Xiao et al., 2024; Liu et al., 2025k; Srivastava & Aggarwal, 2025)

**LLM Agents.**  LLM-based agents represent an emerging paradigm in which LLMs act as autonomous or semi-autonomous decision-making entities (Wang et al., 2025d; Li et al., 2025r), capable of reasoning, planning, and executing actions in pursuit of complex goals. Recent surveys have sought to map this landscape from complementary perspectives. Luo et al. (2025a) propose a methodology-centered taxonomy that connects architectural foundations, collaboration mechanisms, and evolutionary pathways, while Plaat et al. (2025) emphasizes the core capabilities of reasoning, acting, and interacting as defining features of *agentic* LLMs. Tool use, encompassing retrieval-augmented generation (RAG) and API utilization, is a central paradigm, extensively discussed in Li (2025) and further conceptualized by Wang et al. (2024k). Planning and reasoning strategies form another pillar, with surveys such as Masterman et al. (2024) and Kumar et al. (2025) highlighting common design patterns like plan-execute-reflect loops, while Tao et al. (2024) extend this to self-evolution, where agents iteratively refine knowledge and strategies without substantial human intervention. Other directions explore collaborative, cross-modal, and embodied settings, from multi-agent systems (Aratchige & Ilmini, 2025) to multimodal integration (Durante et al., 2024), and brain-inspired architectures with memory and perception (Liu et al., 2025a).

**Research Gap and Our Contributions.**  The recent surge in research on LLM agents and RL-enhanced LLMs reflects two complementary perspectives: one explores what large language models can do as the core of autonomous agents, while the other focuses on how reinforcement learning can optimize their behavior. However, despite the breadth of existing work, a unified treatment of *Agentic RL*, which conceptualizes LLMs as policy-optimized agents embedded in sequential decision processes, remains lacking. Current studies often examine isolated capabilities, domains, or custom environments, with inconsistent terminology and evaluation protocols, making systematic comparison and cross-domain generalization difficult. To bridge

this gap, we present a coherent synthesis that connects theoretical foundations with algorithmic approaches and practical systems. We formalize Agentic RL through Markov decision processes (MDPs) and partially observable Markov decision processes (POMDPs) abstractions to distinguish it from classical LLM RL paradigms, and introduce a capability-centered taxonomy that includes planning, tool use, memory, reasoning, reflection (self-improvement), and interaction as RL-optimizable components. Furthermore, we consolidate representative tasks, environments, frameworks, and benchmarks that support agentic LLM training and evaluation, and conclude by discussing open challenges and outlining promising future directions for scalable, general-purpose agentic intelligence. Overall, we aim to further clarify the research scope of this survey:

> **Primary focus:**
>
> ✔ how RL empowers LLM-based agents (or LLMs with agentic characteristics) in *dynamic environments*
>
> **Out of scope (though occasionally mentioned):**
>
> ✗ RL for human value alignment (*e.g.*, RL for harmful query refusal);
>
> ✗ traditional RL algorithms that are not LLM-based (*e.g.*, MARL (Huh & Mohapatra, 2024));
>
> ✗ RL for boosting pure LLM performance on static benchmarks.

**Structure of the Survey.** This survey is organized to progressively build a unified understanding of Agentic RL from conceptual foundations to practical implementations. Section 2 formalizes the paradigm shift to Agentic RL through an MDP/POMDP lens. Section 3 examines Agentic RL from the capability perspective, categorizing key modules such as planning, reasoning, tool use, memory, self-improvement, perception, and others. Section 4 explores applications across domains, including search, GUI navigation, code generation, mathematical reasoning, and multi-agent systems. Section 5 consolidates open-source environments and RL frameworks that underpin experimentation and benchmarking. Section 6 discusses open challenges and future directions towards scalable, adaptive, and reliable agentic intelligence, and Section 7 concludes the survey. The overall structure is also illustrated in Figure 1.

## 2 Preliminary: From LLM RL to Agentic RL

LLMs are initially pre-trained using behavior cloning, which applies maximum likelihood estimation (MLE) to static datasets such as web-scraped text corpora. Subsequent post-training methods enhance capabilities and align outputs with human preferences—transforming them beyond generic web-data replicators. A common technique is supervised fine-tuning (SFT), where models are refined on human-generated (prompt, response) demonstrations. However, procuring sufficient high-quality SFT data remains challenging (Maosongcao et al., 2025; Szep et al., 2025; Han et al., 2025). Reinforcement fine-tuning (RFT) offers an alternative by optimizing models through reward functions, circumventing dependence on behavioral demonstrations.

In early RFT research, the core objective is to optimize LLMs through human feedback (Christiano et al., 2017; Ouyang et al., 2022) or data preferences (Rafailov et al., 2023), aligning them with human preferences (RLHF) or directly with data preferences (as in DPO).[1] This **preference-based RFT (PBRFT)** primarily involves learning reward model optimization for LLMs on a fixed preference dataset, or directly implementing it using data preferences. With the release of LLMs such as OpenAI o1 (OpenAI et al., 2024) and DeepSeek-R1 (DeepSeek-AI et al., 2025) that possess reasoning capabilities, their improved performance and cross-domain generalization have garnered widespread attention. With the release of models like OpenAI o3 (OpenAI Team, 2025), which possess both self-evolving reasoning capabilities and support for tool use, researchers are beginning to contemplate how to deeply integrate LLMs with downstream tasks through reinforcement learning methods. Subsequently, researchers have shifted their focus from PBRFT, aimed at optimizing fixed preference datasets, to agentic reinforcement learning tailored for specific tasks and dynamic environments.

---

[1]Although DPO is another form of optimization objective in RLHF, its complexity is optimized from the perspective of the training process, so it is necessary to distinguish between pure RLHF and DPO.

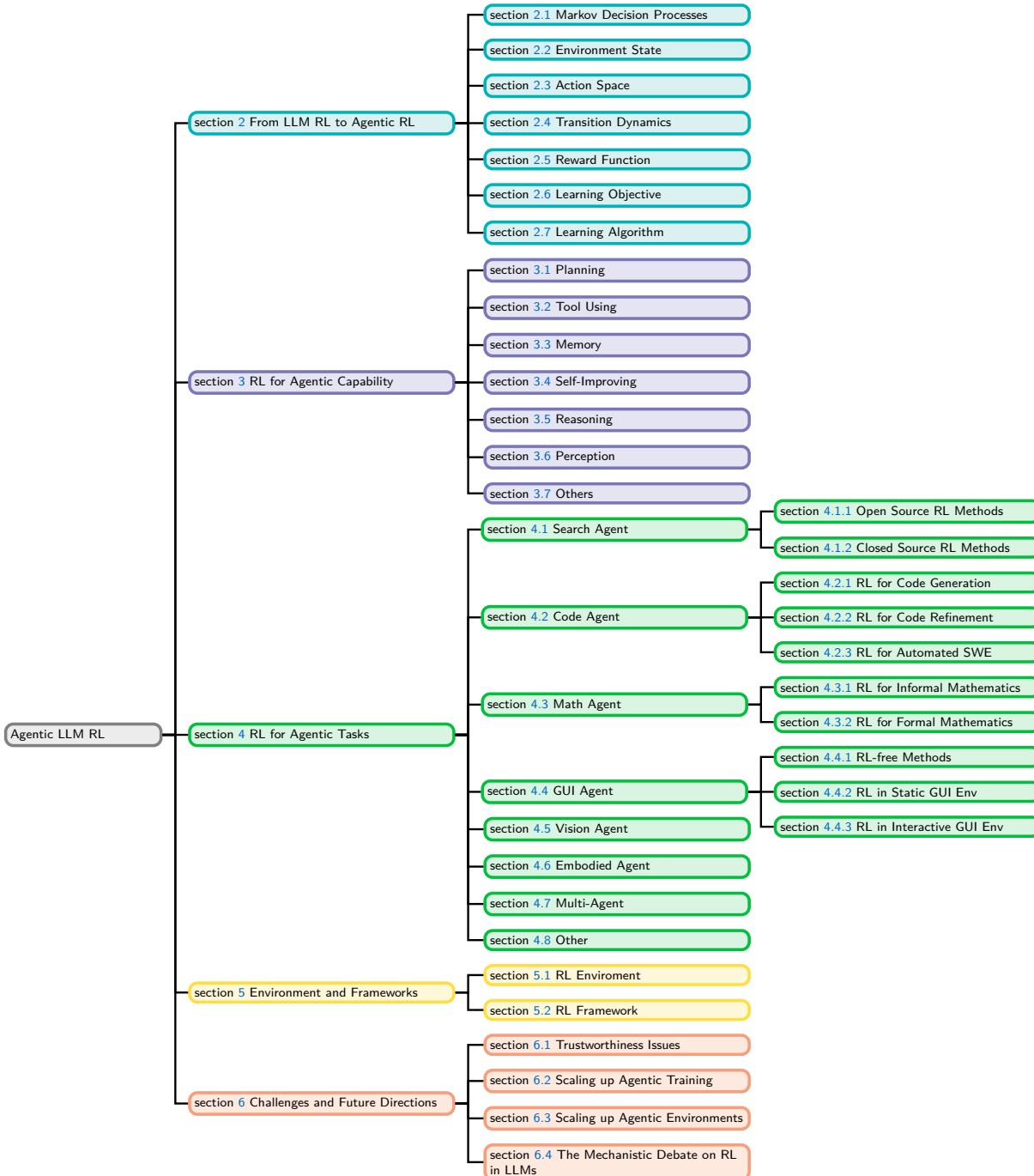

Figure 1: The primary organizational structure of the survey.

In this section, we provide a formalization of the paradigm shift from PBRFT to the emerging framework of **agentic reinforcement learning (Agentic RL)**. While both approaches leverage RL techniques to improve LLMs' performance, they fundamentally differ in their underlying assumptions, task structure, and decision-making granularity. Figure 2 illustrates the paradigm shift from LLM RL to Agentic RL.

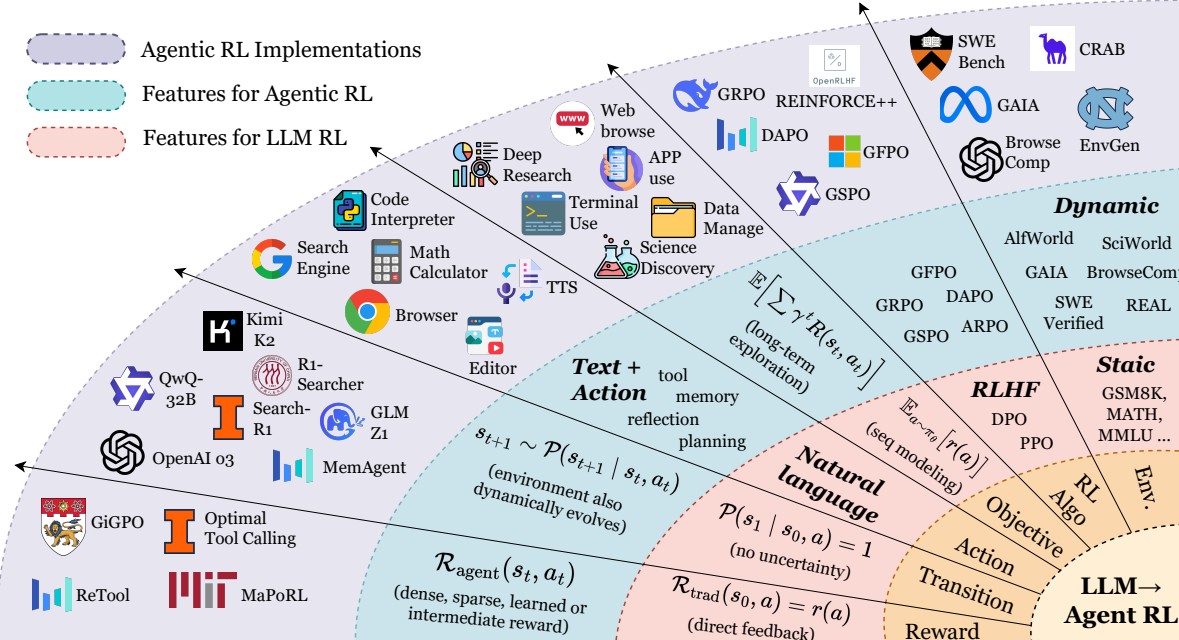

Figure 2: Paradigm shift from LLM RL to Agentic RL. We draw inspiration from (Kumar et al., 2025). The fan-shaped design reflects the outward growth of the RL formulation—from traditional RL (inner), to LLM RL, to full Agentic RL (outer). Color-coded regions represent: red = features specific to LLM RL; teal = features required for Agentic RL; purple = existing Agentic RL implementations. Arrows point outward to indicate increasing interaction breadth (tool use, web browsing, dynamic environments) as one moves toward more agentic settings.

## 2.1 Markov Decision Processes

The Markov decision process (MDP) for the RL fine-tuning process can be formalized as a seven-element tuple $\langle \mathcal{S}, \mathcal{O}, \mathcal{A}, \mathcal{P}, \mathcal{R}, T, \gamma \rangle$, where $\mathcal{S}$ represents the state space and $\mathcal{O}$ is the observation space of the agent. $\mathcal{A}$ denotes the action space. $\mathcal{R}$ is defined as the reward function, $\mathcal{P}$ encapsulates the state transition probabilities, $T$ signifies the task horizon, and $\gamma$ is the discount factor. By casting both preference-based RFT and Agentic RL as MDPs or POMDPs, we clarify the theoretical implications of treating LLMs either as static sequence generators or as interactive, decision-capable agents embedded within dynamic environments.

**PBRFT.** The RL training process of PBRFT is formalized as a degenerate MDP defined by the tuple:

$$\langle \mathcal{S}_{\text{trad}}, \mathcal{A}_{\text{trad}}, \mathcal{P}_{\text{trad}}, \mathcal{R}_{\text{trad}}, T = 1, \gamma = 1 \rangle. \tag{1}$$

**Agentic RL.** The RL training process of Agentic RL is modeled as a POMDP:

$$\langle \mathcal{S}_{\text{agent}}, \mathcal{A}_{\text{agent}}, \mathcal{P}_{\text{agent}}, \mathcal{R}_{\text{agent}}, \gamma, \mathcal{O} \rangle. \tag{2}$$

where the agent receives observations $o_t = O(s_t)$ based on the state $s_t \in \mathcal{S}_{\text{agent}}$. The primary distinctions between PBRFT and Agentic RL are delineated in Table 1. In summary, PBRFT optimizes sequences of output sentences within a fixed dataset under full observations, whereas Agentic RL optimizes semantic-level behaviors in variable environments characterized by partial observations.

## 2.2 Environment State

**PBRFT.** In the training process, each episode starts from a single prompt state $s_0$; the episode terminates immediately after the model emits one response. Formally, the underlying MDP degenerates to a *single-step*

Table 1: Formal comparison between traditional PBRFT and Agentic RL.

| Concept | Traditional PBRFT | Agentic RL |
|---|---|---|
| $\mathcal{S}$: State space | $\{s_0\}$ (single prompt); episode ends immediately. | $s_t \in \mathcal{S}_{\text{agent}}$; $o_t = O(s_t)$; horizon $T > 1$. |
| $\mathcal{A}$: Action space | Pure text sequences. | $\mathcal{A}_{\text{text}} \cup \mathcal{A}_{\text{action}}$. |
| $\mathcal{P}$: Transition | Deterministic transition to the terminal state. | Dynamic transition function $P(s_{t+1} \mid s_t, a_t)$. |
| $\mathcal{R}$: Reward | Single scalar $r(a)$. | Step-wise $R(s_t, a_t)$; combines sparse task and dense sub-rewards. |
| $J(\theta)$: Objective | $\mathbb{E}_{a \sim \pi_\theta}[r(a)]$. | $\mathbb{E}_{\tau \sim \pi_\theta}[\sum_t \gamma^t R(s_t, a_t)]$. |

decision problem with horizon $T = 1$. The state space reduces to a single static prompt input:

$$\mathcal{S}_{\text{trad}} = \{\text{prompt}\}. \tag{3}$$

**Agentic RL.** The LLM agent acts over multiple time-steps in a POMDP. Let $s_t \in \mathcal{S}_{\text{agent}}$ denote the full world state and the LLM agent gets observation $O_t$ based on the current state $o_t = \mathcal{O}(s_t)$. The LLM agent chooses an action $a_t$ based on the current observation $o_t$, and the state evolves over time:

$$s_{t+1} \sim P(s_{t+1} \mid s_t, a_t). \tag{4}$$

as the agent accumulates intermediate signals such as retrieved tool results, user messages, or environment feedback. The interaction is thus inherently dynamic and temporally extended.

## 2.3 Action Space

In the Agentic RL setting, the LLM's action space comprises two distinct subspaces:

$$\mathcal{A}_{\text{agent}} = \mathcal{A}_{\text{text}} \cup \mathcal{A}_{\text{action}}. \tag{5}$$

Here, $\mathcal{A}_{\text{text}}$ denotes the space of free-form natural language tokens emitted via autoregressive decoding, while $\mathcal{A}_{\text{action}}$ denotes the space of abstract, non-linguistic actions, which is usually delimited in the output stream by special tokens `<action_start>` and `<action_end>`. These actions may invoke external tools (e.g., `call("search", "Einstein")`) or interact with an environment (e.g., `move("north")`), depending on task requirements.

Notably, $\mathcal{A}_{\text{action}}$ is recursively constructed, such that an element $a \in \mathcal{A}_{\text{action}}$ may itself represent a sequence $(a_1, \ldots, a_k)$ of primitive actions, thus unifying primitive and composite actions within the same space.

Formally, the two subspaces differ in semantics and functional role: $\mathcal{A}_{\text{text}}$ defines the space of outputs intended for human or machine interpretation without directly altering the external state, whereas $\mathcal{A}_{\text{action}}$ defines the space of environment-interactive behaviors that either (i) acquire new information through tool invocations, or (ii) modify the state of a physical or simulated environment. This distinction enables a unified policy jointly to model language generation and environment interaction within the same RL formulation.

## 2.4 Transition Dynamics

**PBRFT.** In conventional PBRFT, the transition dynamics are deterministic: the next state is determined once an action is taken, as follows:

$$\mathcal{P}(s_1 \mid s_0, a) = 1, \quad \text{where there is no uncertainty.} \tag{6}$$

**Agentic RL.** In Agentic RL, the environment evolves under uncertainty according to

$$s_{t+1} \sim \mathcal{P}(s_{t+1} \mid s_t, a_t), \quad a_t \in \mathcal{A}_{\text{text}} \cup \mathcal{A}_{\text{action}}. \tag{7}$$

Text actions ($\mathcal{A}_{\text{text}}$) generate natural language outputs without altering the environmental state. Structured actions ($\mathcal{A}_{\text{action}}$), delimited by `<action_start>` and `<action_end>`, can either query external tools or directly modify the environment. This sequential formulation contrasts with the one-shot mapping of PBRFT, enabling policies that iteratively combine communication, information acquisition, and environment manipulation.

### 2.5 Reward Function

**PBRFT.** PBRFT commonly features a reward function with verifiable response correctness, which may be implemented using either a rule-based verifier (DeepSeek-AI et al., 2025) or a neural network-parameterized reward model (Zhong et al., 2025). Regardless of the implementation approach, its core follows the equation:

$$\mathcal{R}_{\text{trad}}(s_0, a) = r(a). \tag{8}$$

where $r : \mathcal{A} \to \mathbb{R}$ is a scalar score supplied by a human- or AI-preference model, with no intermediate feedback.

**Agentic RL.** The reward function of the LLM agent is based on the downstream task.

$$\mathcal{R}_{\text{agent}}(s_t, a_t) = \begin{cases} r_{\text{task}} & \text{on task completion,} \\ r_{\text{sub}}(s_t, a_t) & \text{for step-level progress,} \\ 0 & \text{otherwise.} \end{cases} \tag{9}$$

allowing dense, sparse, or learned rewards (*e.g.*, unit-test passes, symbolic verifier success).

### 2.6 Learning Objective

**PBRFT.** The optimization objective of PBRFT is to maximize the response reward based on the policy $\pi_\theta$:

$$J_{\text{trad}}(\theta) = \mathbb{E}_{a \sim \pi_\theta}\big[r(a)\big]. \tag{10}$$

No discount factor is required; optimization resembles maximum-expected-reward sequence modeling.

**Agentic RL.** The optimization objective of Agentic RL is to maximize the discounted reward:

$$J_{\text{agent}}(\theta) = \mathbb{E}_{\tau \sim \pi_\theta}\left[\sum_{t=0}^{T-1} \gamma^t R_{\text{agent}}(s_t, a_t)\right], \qquad 0 < \gamma < 1. \tag{11}$$

This objective is optimized via policy-gradient or value-based methods with exploration and long-term credit assignment.

PBRFT focuses on single-turn text quality alignment without explicit planning, tool use, or environmental feedback, while Agentic RL involves multi-turn planning, adaptive tool invocation, stateful memory, and long-horizon credit assignment, enabling the LLM to function as an autonomous decision-making agent.

### 2.7 RL Algorithms

In contemporary research, RL algorithms constitute a pivotal component in both PBRFT and Agentic RL frameworks. Different RL algorithms demonstrate distinct sample efficiency and performance characteristics, each offering a unique approach to the central challenge of aligning model outputs with complex, often subjective, human goals. The canonical methods, such as REINFORCE, PPO (Schulman et al., 2017), GRPO (DeepSeek-AI et al., 2025), and DPO (Rafailov et al., 2023), form a spectrum from general policy gradients to specialized preference learning. We next introduce each of these four classic algorithms and provide a comparison of popular variants from each family in Table 2.

**REINFORCE: The Foundational Policy Gradient**   As one of the earliest policy gradient algorithms, REINFORCE (Williams, 1992) provides the foundational theory for training stochastic policies. It operates by increasing the probability of actions that lead to high cumulative reward and decreasing the probability of those that lead to low reward. Its objective function is given by:

$$\nabla_\theta J(\theta) = \mathbb{E}_{s_0}\left[\frac{1}{N}\sum_{i=1}^{N}\left(\mathcal{R}(s_0, a^{(i)}) - b(s_0)\right)\nabla_\theta \log \pi_\theta(a^{(i)}|s_0)\right]. \tag{12}$$

where $a^{(i)} \sim \pi_\theta(a|s_0)$ is the $i$-th sampled response, $\mathcal{R}(s_0, a)$ denotes the final rewards received on task completion, and $b(s)$ is a baseline function to reduce the variance of the policy gradient estimate. In general, $b(s)$ can be any function, including random variables. In practice, $b(s)$ is commonly instantiated as the value function $V(s)$. Despite with advantages of the concise formula and easy implementation, REINFORCE suffers from drawbacks such as high variance in gradient estimates, sample inefficiency, sensitivity to learning rate and the lack of a critic (value estimator).

**Proximal Policy Optimization (PPO)**   PPO (Schulman et al., 2017) became the dominant RL algorithm for LLM alignment due to its stability and reliability. It improves upon vanilla policy gradients by limiting the update step to prevent destructively large policy changes. Its primary clipped objective function is:

$$L_{PPO}(\theta) = \frac{1}{N}\sum_{i=1}^{N}\min\left(\frac{\pi_\theta(a_t^{(i)}|s_t)}{\pi_{\theta_{old}}(a_t^{(i)}|s_t)}A(s_t, a_t^{(i)}),\ \ \text{clip}\left(\frac{\pi_\theta(a_t^{(i)}|s_t)}{\pi_{\theta_{old}}(a_t^{(i)}|s_t)}, 1-\epsilon, 1+\epsilon\right)A(s_t, a_t^{(i)})\right). \tag{13}$$

where $a_t^{(i)} \sim \pi_{\theta_{old}}(a|s_t)$ is the $i$-th sampled response from the old policy $\pi_{\theta_{old}}$, whose update is delayed. $A_t$ is the estimated advantage given by

$$A(s_t, a_t) = \mathcal{R}(s_t, a_t) - V(s_t). \tag{14}$$

where $V_\theta(s)$ is the learned value function, i.e., the expectation $\mathbb{E}_{a\sim\pi_\theta(a|s)}[\mathcal{R}(s, a)]$, which is typically, but not necessarily, derived from a critic network that is of the same size as the policy network. The clip term prevents the probability ratio from moving too far from 1, ensuring stable updates.   The estimation of the advantage function plays a predominant role in the performance of PPO. Recent variants have concentrated on reducing the bias (Kazemnejad et al., 2024) or variance (Yue et al., 2025b) in the advantage estimation. Meanwhile, some other variants make improvements from the perspectives of stable policy update mechanisms (Liu et al., 2025s) or mitigating sparse rewards (Dai et al., 2025). Despite these improvements, a remaining drawback is its reliance on a separate critic network for advantage estimation, which substantially increases the parameter count during training.

**Direct Preference Optimization (DPO)**   DPO represents a groundbreaking shift by entirely bypassing the need for a separate explicit reward model. It reframes the problem of maximizing a reward under a KL-constraint as a likelihood-based objective on human preference data. Given a dataset of preferences $D = \{(y_w, y_l)\}$, where $y_w$ is the preferred response and $y_l$ is the dispreferred one, the DPO loss is:

$$L_{DPO}(\pi_\theta; \pi_{ref}) = -\mathbb{E}_{(x,y_w,y_l)\sim D}\left[\log \sigma\left(\beta\log\frac{\pi_\theta(y_w|x)}{\pi_{ref}(y_w|x)} - \beta\log\frac{\pi_\theta(y_l|x)}{\pi_{ref}(y_l|x)}\right)\right]. \tag{15}$$

where $\pi_{ref}$ is a reference policy (usually the initial SFT model), and $\beta$ is a hyperparameter.   While DPO eliminates the critic, its performance is intrinsically tied to the quality and coverage of its static preference dataset. Variants have emerged to address its limitations via involving external or online data (Ethayarajh et al., 2024; Hong et al., 2024a). In addition, some other work attempts to improve by introducing generalized optimization objectives (Gheshlaghi Azar et al., 2024) or sophisticated implicit reward mechanisms (Meng et al., 2024; Lai et al., 2024; Hong et al., 2025a).

**Group Relative Policy Optimization (GRPO)**   The remarkable success achieved by DeepSeek (Guo et al., 2025a) has catalyzed significant research interest in GRPO. Proposed to address the inefficiency of PPO's large critic, GRPO introduces a novel, lightweight evaluation paradigm. It operates on groups of

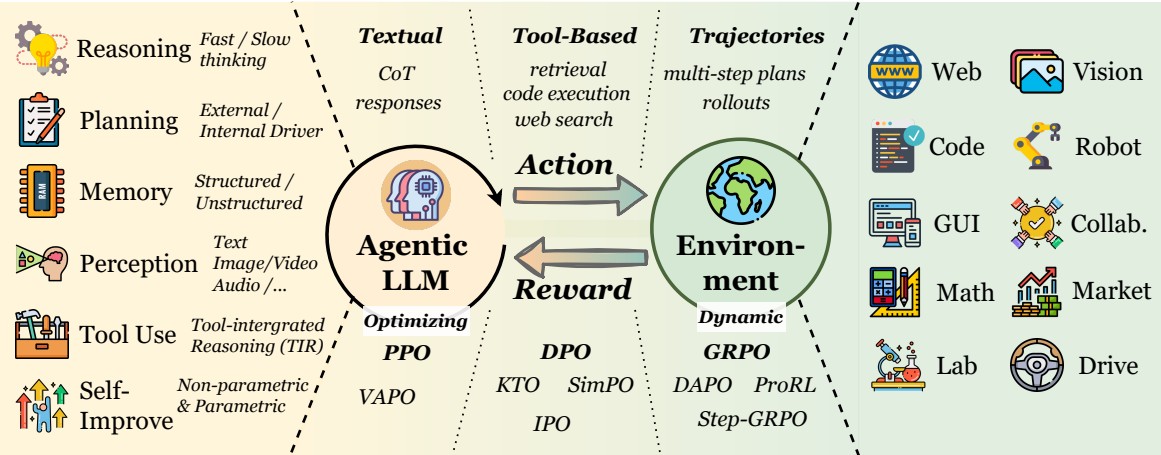

Figure 3: The agent–environment interaction and RL loop for agentic LLMs. Core agentic capabilities drive action generation, while the environment provides feedback and rewards, which are aggregated through RL-based optimization across diverse task domains ("Collab." denotes tasks requiring explicit task division and multi-agent coordination).

responses, using their relative rewards within a group to compute advantages, thus eliminating the need for an absolute value critic. The core GRPO objective can be conceptualized as:

$$L_{GRPO} = \frac{1}{G} \sum_{g=1}^{G} \min \left( \frac{\pi_\theta(a_t^{(g)}|s_t^{(g)})}{\pi_{\theta_{old}}(a_t^{(g)}|s_t^{(g)})} \hat{A}(s_t^{(g)}, a_t^{(g)}), \ \text{clip} \left( \frac{\pi_\theta(a_t^{(g)}|s_t^{(g)})}{\pi_{\theta_{old}}(a_t^{(g)}|s_t^{(g)})}, 1-\epsilon, 1+\epsilon \right) \hat{A}(s_t^{(g)}, a_t^{(g)}) \right). \quad (16)$$

where a group of outputs $\{(s_0^{(g)}, a_0^{(g)}, \ldots, s_{T-1}^{(g)}, a_{T-1}^{(g)})\}_{g=1}^{G}$ is sampled from the old policy $\pi_{\theta_{old}}$. The advantage function is estimated by

$$\hat{A}(s_t, a_t) = \frac{\mathcal{R}(s_t, a_t) - \text{mean}(\mathcal{R}(s_t^{(1)}, a_t^{(1)}), \ldots, \mathcal{R}(s_t^{(G)}, a_t^{(G)}))}{\text{std}(\mathcal{R}(s_t^{(1)}, a_t^{(1)}), \ldots, \mathcal{R}(s_t^{(G)}, a_t^{(G)}))}. \quad (17)$$

This group-relative approach is highly sample-efficient and reduces computational overhead. However, the group-based advantage estimation is vulnerable to high variance and low accuracy. Consequently, a series of novel algorithms derived from the GRPO framework have been subsequently proposed (see Table 2), aiming to substantially improve its advantage estimation.

## 3 Agentic RL: The model capability perspective

In this section, we conceptually characterize **Agentic RL** as the principled training of an autonomous agent composed of a set of key abilities/modules, *i.e.*, planning (Section 3.1), tool use (Section 3.2), memory (Section 3.3), self-improvement (Section 3.4), reasoning (Section 3.5), perception (Section 3.6), and others (Section 3.7), following the classic LLM agent definition (Weng, 2023; Shang et al., 2025b), as demonstrated in Figure 5. Traditionally, an agent pairs an LLM with mechanisms for planning (*e.g.*, task decomposition and plan selection) (Wei et al., 2025a), reasoning (chain-of-thought or multi-turn inference) (Zhang et al., 2024c), external tool invocation (Qin et al., 2024b), long- and short-term memory, and iterative reflection to self-correct and refine behavior. Agentic RL thus treats these components not as static pipelines but as interdependent policies that can be jointly optimized: RL for planning learns multi-step decision trajectories; RL for memory shapes retrieval and encoding dynamics; RL for tool use optimizes invocation timing and fidelity; and RL for reflection drives internal self-supervision and self-improvement. Consequently, our survey systematically examines how RL empowers planning, tool use, memory, reflection, and reasoning in subsequent

Table 2: Comparison of the popular variants of the PPO, DPO, and GRPO families. Clip corresponds to preventing the policy ratio from moving too far from 1 for ensuring stable updates. KL penalty corresponds to penalizing the KL divergence between the learned policy and the reference policy for ensuring alignment.

| Method | Objective Type | Key Mechanism |
|---|---|---|
| *PPO family* | | |
| PPO (Schulman et al., 2017) | Policy gradient | Policy ratio clipping |
| VAPO (Yue et al., 2025b) | Policy gradient | Adaptive KL penalty + variance control |
| LitePPO (Liu et al., 2025s) | Policy gradient | Stable advantage updates |
| PF-PPO (Zhang et al., 2025c) | Policy gradient | Policy filtration |
| VinePPO (Kazemnejad et al., 2024) | Policy gradient | Unbiased value estimates |
| PSGPO (Dai et al., 2025) | Policy gradient | Process supervision |
| *DPO family* | | |
| DPO (Rafailov et al., 2023) | Preference optimization | Implicit reward related to the policy |
| $\beta$-DPO (Wu et al., 2024) | Preference optimization | Dynamic KL coefficient |
| SimPO (Meng et al., 2024) | Preference optimization | Use the average log probability of a sequence as the implicit reward |
| IPO (Gheshlaghi Azar et al., 2024) | A special case of a more general objective exclusively expressed in terms of pairwise preferences | Always regularizes its solution towards a preference policy by controlling the gap between the log-likelihood ratios, which avoids the over-fitting to the preference dataset. |
| KTO (Ethayarajh et al., 2024) | Knowledge transfer optimization | Teacher stabilization |
| ORPO (Hong et al., 2024a) | Online regularized preference optimization | Online stabilization |
| Step-DPO (Lai et al., 2024) | Preference optimization | Step-wise supervision |
| LCPO (Hong et al., 2025a) | Preference optimization | Length preference with limited data and training |
| *GRPO family* | | |
| GRPO (DeepSeek-AI et al., 2025) | Policy Gradient under group-based reward | Group-based relative reward to eliminate value estimates |
| DAPO (Yu et al., 2025e) | Surrogate of GRPO's | Decoupled clip and dynamic sampling |
| GSPO (Zheng et al., 2025a) | Surrogate of GRPO's | Define the importance ratio based on sequence likelihood and performs sequence-level clipping, rewarding, and optimization |
| GMPO (Zhao et al., 2025f) | Surrogate of GRPO's | Geometric mean of token-level rewards |
| ProRL (Liu et al., 2025h) | Same as GRPO's | Reference policy reset |
| Posterior-GRPO (Fan et al., 2025a) | Same as GRPO's | Reward only successful processes |
| Dr.GRPO (Liu et al., 2025r) | Unbiased GRPO's objective | Eliminate the bias in optimization of GRPO |
| Step-GRPO (Zhang et al., 2025j) | Same as GRPO's | Rule-based reasoning rewards |
| SRPO (Zhang et al., 2025s) | Same as GRPO's | Two-staged history-resampling |
| GRESO (Zheng et al., 2025b) | Same as GRPO's | Pre-rollout filtering |
| StarPO (Wang et al., 2025v) | Same as GRPO's | Reasoning-guided actions for multi-turn interactions |
| GHPO (Liu et al., 2025u) | Policy gradient | Adaptive prompt refinement |
| Skywork R1V2 (Wang et al., 2025i) | GRPO's with hybrid reward signal | Selective sample buffer |
| ASPO (Lin & Xu, 2025) | GRPO's with shaped advantage function | Apply a clipped bias directly to advantage function |
| TreePo (Li et al., 2025n) | Same as GRPO's | Self-guided policy rollout for reducing the compute burden |
| EDGE-GRPO (Zhang et al., 2025c) | Same as GRPO's | Entropy-driven advantage and duided error correction to mitigate advantage collapse |
| DARS (Yang et al., 2025h) | Same as GRPO's | Reallocate compute from medium-difficulty to the hardest problems via multi-stage rollout sampling |
| CHORD (Zhang et al., 2025q) | Weighted sum of GRPO's and Supervised Fine-Tuning losses | Reframe Supervised Fine-Tuning as a dynamically weighted auxiliary objective within the on-policy RL process |
| PAPO (Wang et al., 2025u) | Surrogate of GRPO's | Encourage learning to perceive while learning to reason through the Implicit Perception Loss |
| Pass@k Training (Chen et al., 2025l) | Same as GRPO's | Pass@k metric as the reward to continually train a model |

subsections. We aim to provide a high-level conceptual delineation of RL's applications for agent capabilities, rather than an exhaustive enumeration of all related work, which we provide in Section 4.

## 3.1 Planning

Planning, the deliberation over a sequence of actions to achieve a goal, constitutes a cornerstone of artificial intelligence, demanding complex reasoning, world knowledge, and adaptability (Newell et al., 1958). Initial

efforts leveraged the innate capabilities of LLMs through prompting-based methods (Huang et al., 2024a; Yao et al., 2023b). For example, Modular Agentic Planner (MAP) (Webb et al., 2025) introduces a brain-inspired, modular architecture that decomposes planning into specialized LLM modules for conflict monitoring, state evaluation, and coordination. However, these approaches lacked a mechanism for adaptation through experience (Wei et al., 2025a). RL has emerged as a powerful paradigm to address this gap, enabling agents to refine their planning strategies by learning from environmental feedback. The integration of RL into agent planning manifests in two distinct paradigms, distinguished by whether RL functions as an **external guide** to a structured planning process or as an **internal driver** that directly evolves the LLM's intrinsic planning policy, which we will detail below.

**RL as an External Guide for Planning.** One major paradigm frames RL as an external guide to the planning process, where the LLM's primary role is to generate potential actions within a structured search framework. Here, RL is not employed to fine-tune the LLM's generative capabilities directly, but rather to train an auxiliary value or heuristic function (Wei et al., 2025a). This learned function then guides a classical search algorithm, such as Monte Carlo Tree Search (MCTS), by evaluating the quality of different planning trajectories. Representative works like RAP (Hao et al., 2023) and LATS (Zhou et al., 2024a) exemplify this approach. Planning without Search (Hong et al., 2025d) extends this idea by leveraging offline goal-conditioned RL to learn a language-based value critic that guides LLM reasoning and planning without updating the LLM's parameters. In this configuration, the LLM acts as a knowledge-rich action proposer, while RL provides adaptive, evaluative feedback for efficient exploration. Beyond static guidance, Learning When to Plan (Paglieri et al., 2025) formulates dynamic planning as an RL-driven test-time compute allocation problem, training agents to decide when to invoke explicit planning to balance reasoning performance against computational cost. Conversely, MAPF-DT (Atasever et al., 2025) explores the reverse direction, employing Decision Transformer–based offline RL for decentralized multi-agent path planning, with LLM guidance enhancing adaptability and long-horizon efficiency in dynamic environments.

**RL as an Internal Driver of Planning.** A second, more integrated paradigm positions RL as an internal driver of the agent's core planning capabilities. This approach casts the LLM directly as a policy model and optimizes its planning behavior through direct environmental interaction. Instead of guiding an external search algorithm, RL-based feedback from trial and error is used to directly refine the LLM's internal policy for generating plans. This is achieved through methods derived from RLHF, such as leveraging DPO on successful versus failed trajectories as seen in ETO (Song et al., 2024b), or through lifelong learning frameworks. For instance, VOYAGER (Wang et al., 2024a) iteratively builds and refines a skill library from environmental interaction. This paradigm transforms the LLM from a static generator into an adaptive policy that continuously evolves, enhancing its robustness and autonomy in dynamic environments. In a complementary direction, Dynamic Speculative Planning (DSP) (Guan et al., 2025b) embodies an online reinforcement mechanism that adapts the agent's policy to jointly optimize latency and operational cost, demonstrating that internal policy refinement can govern not only task success but also system efficiency. RLTR (Li et al., 2025p) decouples planning from answer generation and introduces tool-use rewards that directly evaluate action sequence quality, enabling focused optimization of the agent's planning capability without relying on verifiable final answers. AdaPlan and its PilotRL framework (Lu et al., 2025c) leverage global plan-based guidance with progressive RL to enhance LLM agents' long-horizon planning and execution coordination in text game environments like AFLWorld and TextCraft. Planner-R1 (Zhu et al., 2025d) examines reward-density effects in Agentic RL, showing that shaped, process-level rewards markedly improve learning efficiency and enable smaller models to attain competitive planning capability.

**Prospective: The Synthesis of Deliberation and Intuition.** The prospective horizon for agentic planning lies in the synthesis of these two paradigms: moving beyond the distinction between external search and internal policy optimization. The ultimate goal is to develop an agent that **internalizes the structured search process itself**, seamlessly blending intuitive, fast plan generation with deliberate, slow, deliberative reasoning. In such a model, RL would not only refine the final plan but also optimize a meta-policy governing the deliberation process: learning when to explore alternative paths, how to prune unpromising branches, and how deeply to reason before committing to an action. This would transform the LLM agent from a component that either proposes actions or acts as a raw policy into an integrated reasoning engine.

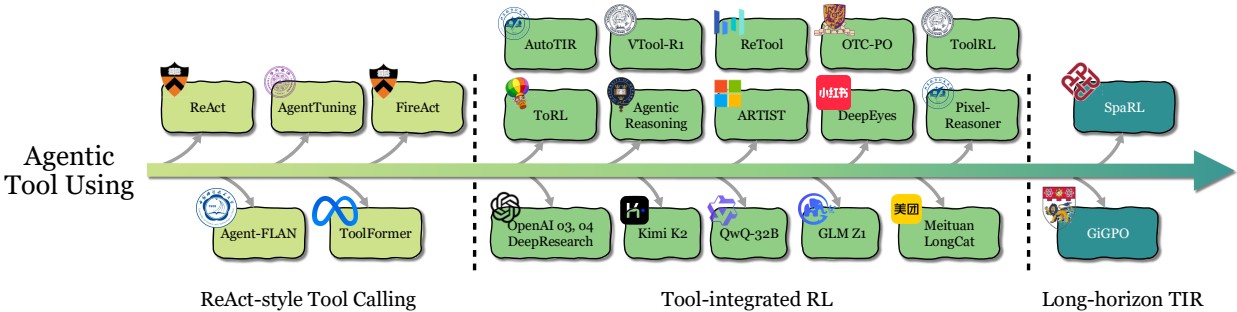

Figure 4: The development of agentic tool use. Note that we only select a small bunch of representative works here to reflect the progress.

## 3.2 Tool Using

RL has emerged as a pivotal methodology for evolving tool-enabled language agents from post-hoc, ReAct-style pipelines to deeply interleaved, multi-turn Tool-Integrated Reasoning (TIR) systems. While early paradigms successfully demonstrated the feasibility of tool invocation, their reliance on SFT or prompt engineering limited agents to mimicking static patterns, lacking the strategic flexibility to adapt to novel scenarios or recover from errors (Chen et al., 2024g; Kavathekar et al., 2025). Agentic RL addresses this by shifting the learning paradigm from imitation to outcome-driven optimization, enabling agents to autonomously discover when, how, and which tools to deploy. This evolution charts a clear trajectory, which we explore in three stages. We begin with (1) early ReAct-style tool calling, then examine (2) modern tool-integrated reasoning (TIR) that deeply embeds tool use within cognitive loops, and finally, discuss the prospective challenge of (3) multi-turn TIR, focusing on temporal credit assignment for robust, long-horizon performance.

**ReAct-style Tool Calling.** Early paradigms for tool invocation predominantly relied on either prompt engineering or SFT to elicit tool-use behaviors. The **(I) prompt engineering** approach, exemplified by ReAct (Yao et al., 2023b), leveraged few-shot exemplars to guide an LLM to interleave reasoning traces and actions within a "Thought-Action-Observation" cycle, capitalizing on the model's in-context learning abilities. Going beyond, **(II) SFT-based methods** were introduced to internalize models' tool-use capabilities. Frameworks like Toolformer (Schick et al., 2023) employed a self-supervised objective to teach models where to insert API calls, while others like FireAct (Chen et al., 2023), AgentTuning (Zeng et al., 2024), Agent-FLAN (Chen et al., 2024h) fine-tuned models on expert-generated or curated datasets of tool-interaction trajectories (e.g., AgentBank (Song et al., 2024a), APIBank (Li et al., 2023b)). Although SFT improved the reliability of tool invocation, both of these early approaches are fundamentally constrained by their imitative nature. They train agents to replicate static, pre-defined patterns of tool use, thereby lacking the strategic flexibility to adapt to novel scenarios or recover from unforeseen errors, a limitation that RL-centric approaches directly address by shifting the learning objective from imitation to outcome-driven optimization.

**Tool-integrated RL.** Building on the limitations of purely imitative paradigms, RL-based approaches for tool use shift the objective from replicating fixed patterns to optimizing end-task performance. This transition enables agents to *strategically* decide *when*, *how*, and *in what combination* to invoke tools, adapting dynamically to novel contexts and unforeseen failures. At the foundation, frameworks such as ToolRL (Qian et al., 2025) demonstrate that, even when initialized from base models without any imitation traces, RL training can elicit emergent capabilities, e.g., self-correction of faulty code, adaptive adjustment of invocation frequency, and the composition of multiple tools for complex sub-tasks. Subsequently, a recent surge in research has produced works such as OTC-PO (Wang et al., 2025e), ReTool (Feng et al., 2025a), AutoTIR (Wei et al., 2025c), VTool-R1 (Wu et al., 2025g), DeepEyes (Zheng et al., 2025g), Pixel-Reasoner (Su et al., 2025a), Agentic Reasoning (Wu et al., 2025e), ARTIST (Singh et al., 2025), ToRL (Li et al., 2025l) and numerous other works (Hao et al., 2025a; Feng et al., 2024a; Wei et al., 2025f; Li et al., 2025f; Wu et al., 2025a; Li et al., 2025i; Chen et al., 2025d; Song et al., 2025d; Ye et al., 2025a), which employ RL policies that interleave

symbolic computation (e.g., code execution, image editing) with natural-language reasoning within a single rollout. This integrated control loop allows the agent to balance precise, tool-mediated operations with flexible verbal inference, tailoring the reasoning process to the evolving task state. Lin & Xu (2025) theoretically proves that TIR fundamentally expands LLM capabilities beyond the "invisible leash" of pure-text RL by introducing deterministic tool-driven state transitions, establishes token-efficiency arguments for feasibility under finite budgets, and proposes Advantage Shaping Policy Optimization (ASPO) to stably guide agentic tool use.

Today, such tool-integrated reasoning is no longer a niche capability but a baseline feature of advanced agentic models. Mature commercial and open-source systems, such as OpenAI's DeepResearch and o3 (OpenAI, 2025), Kimi K2 (Kimi, 2025), Qwen QwQ-32B (Team, 2025c), Zhipu GLM Z1 (AI, 2025), Microsoft rStar2-Agent (Shang et al., 2025a) and Meituan LongCat (Meituan, 2025), routinely incorporate these RL-honed strategies, underscoring the centrality of outcome-driven optimization in tool-augmented intelligence.

**Prospective: Long-horizon TIR.** While tool-integrated RL has proven effective for optimizing actions within a single reasoning loop, the primary frontier lies in extending this capability to robust, long-horizon tasks that require multi-turn reasoning (Gao et al., 2025c). This leap is fundamentally bottlenecked by the challenge of temporal credit assignment (Pignatelli et al., 2024). Current RL approaches often depend on sparse, trajectory-level/outcome-based rewards, making it difficult to pinpoint which specific tool invocation in a long, interdependent sequence contributed to success or failure. While nascent research has begun to explore more granular reward schemes, such as turn-level advantage estimation in GiGPO (Feng et al., 2025b) and SpaRL (Wang et al., 2025b), these are still early steps. Consequently, developing more granular credit assignment mechanisms that can accurately guide the agent through complex decision chains without inadvertently punishing useful exploration or promoting reward hacking remains a critical and largely unsolved problem for advancing agentic systems.

### 3.3 Memory

Agentic RL transforms memory modules from passive data stores into dynamic, RL-controlled subsystems, deciding what to store, when to retrieve, and how to forget similar to humans (Wu et al., 2025k). This section traces this evolution through four representative phases.

**RL in RAG-style Memory.** Early systems (*e.g.*, retrieval-augmented generation) treated memory as an external datastore; when RL was employed at all, it solely regulated when to perform queries. Several classic memory systems without RL involvement, such as MemoryBank (Zhong et al., 2024), MemGPT (Packer et al., 2023), and HippoRAG (Gutiérrez et al., 2024), adopt predefined memory management strategies that specify how to *store*, *integrate*, and *retrieve* information (*e.g.*, storage via vector databases or knowledge graphs; retrieval based on semantic similarity or topological connectivity). Subsequently, RL was incorporated into the memory management pipeline as a functional component. A notable example is the framework proposed in Tan et al. (2025b), where the RL policy adjusts retrieval behavior through *prospective reflection* (multi-level summarization) and *retrospective reflection* (reinforcing retrieval outcomes). Nevertheless, the memory medium itself remained static (*e.g.*, simple vector store or summary buffer), and the agent exerted no control over the write processes. Recently, Memory-R1 (Yan et al., 2025b) introduced an RL-based memory-augmented Agent framework where a Memory Manager learns to perform structured operations (ADD/UPDATE/DELETE/NOOP) via PPO or GRPO based on downstream QA performance, while an Answer Agent employs a Memory Distillation policy over RAG-retrieved entries to reason and answer. Follow-up works like Mem-$\alpha$ (Wang et al., 2025t) and Memory-as-action (Zhang et al., 2025v) have also explored RL for training agents into automatic memory managers.

**RL for Token-level Memory.** Subsequent advancements introduced models equipped with explicit, trainable memory controllers, enabling agents to regulate their own memory states (often stored in token form) without relying on fixed, external memory systems. Notably, such memory is commonly instantiated in two forms. The first is **(I) explicit tokens**, corresponding to human-readable natural language. For example, in MemAgent (Yu et al., 2025d), the agent maintains a natural-language memory pool alongside the LLM, with an RL policy determining, at each segment, which tokens to retain or overwrite, effectively compressing

Table 3: An overview of three classic categories of agent memory; works marked with $^\dagger$ directly employ RL. The list here is not exhaustive, and we refer readers interested in broader agent memory to Wu et al. (2025k). The shaded rows indicate the use of reinforcement learning algorithms.

| Method | Type | Key Characteristics |
|---|---|---|
| *RAG-style Memory* | | |
| MemoryBank (Zhong et al., 2024) | External Store | Static memory with predefined storage/retrieval rules |
| MemGPT (Packer et al., 2023) | External Store | OS-like agent with static memory components |
| HippoRAG (Gutiérrez et al., 2024) | External Store | Neuro-inspired memory with heuristic access |
| Prospect$^\dagger$ (Tan et al., 2025b) | RL-guided Retrieval | Uses RL for reflection-driven retrieval adjustment |
| Memory-R1† (Yan et al., 2025b) | RL-guided Retrieval | RL-driven memory ADD/UPDATE/DELETE/NOOP |
| Mem-$\alpha$† (Wang et al., 2025t) | RL-guided Retrieval | RL-guided agents for memory retrieval |
| Memory-as-action (Zhang et al., 2025v) | RL-guided Management | End-to-end training agents for memory management |
| *Token-level Memory* | | |
| MemAgent† (Yu et al., 2025d) | Explicit Token | RL controls which NL tokens to retain or overwrite |
| MEM1$^\dagger$ (Zhou et al., 2025g) | Explicit Token | Memory pool managed by RL to enhance context handling |
| Memory Token (Jin et al., 2025b) | Explicit Token | Structured memory for reasoning disentanglement |
| ReSum† (Wu et al., 2025i) | Explicit Token | Turn-wise Interaction summary for ReAct agents |
| Context Folding† (Sun et al., 2025c) | Explicit Token | Context folding for ReAct agents |
| MemoryLLM (Wang et al., 2024h) | Latent Token | Latent tokens repeatedly integrated and updated |
| M+ (Wang et al., 2025s) | Latent Token | Scalable memory tokens for long-context tracking |
| IMM (Orlicki, 2025) | Latent Token | Decouples word representations and latent memory |
| Memory (Hongkang Yang et al., 2024) | Latent Token | Forget-resistant memory tokens for evolving context |
| MemGen† (Zhang et al., 2025e) | Latent Token | Context-sensitive latent token as memory carriers |
| *Structured Memory* | | |
| Zep (Rasmussen et al., 2025) | Temporal Graph | Temporal knowledge graph enabling structured retrieval |
| A-MEM (Xu et al., 2025d) | Atomic Memory Notes | Symbolic atomic memory units; structured storage |
| G-Memory (Zhang et al., 2025d) | Hierarchical Graph | Multi-level memory graph with topological structure |
| Mem0 (Chhikara et al., 2025) | Structured Graph | Agent memory with full-stack graph-based design |

long-context inputs into concise, informative summaries. Similar approaches include MEM1 (Zhou et al., 2025g) and Memory Token (Jin et al., 2025b), both of which explicitly preserve a pool of natural-language memory representations. More frequently, works like ReSum (Wu et al., 2025i), context folding (Sun et al., 2025c) have also explored RL for context memory management. The second form is **(II) implicit tokens**, where memory is maintained in the form of latent embeddings. A representative line of work includes MemoryLLM (Wang et al., 2024h) and M+ (Wang et al., 2025s), in which a fixed set of latent tokens serves as "memory tokens." As the context evolves, these tokens are repeatedly retrieved, integrated into the LLM's forward computation, and updated, thereby preserving contextual information and exhibiting strong resistance to forgetting. Unlike explicit tokens, these memory tokens are not tied to human-readable text but rather constitute a machine-native form of memory. Related efforts include IMM (Orlicki, 2025) and Memory (Hongkang Yang et al., 2024). Across both paradigms, these approaches empower agents to autonomously manage their memory banks, delivering significant improvements in long-context understanding, continual adaptation, and self-improvement. MemGen (Zhang et al., 2025e) for the first time proposes the paradigm of leveraging latent memory tokens for carrying and generating experiential knowledge, posing promising directions for RL-based latent memory.

**Prospective: RL for Structured Memory.** Building on token-level approaches, recent trends are moving toward *structured* memory representations, which organize and encode information beyond flat token sequences. Representative examples include the temporal knowledge graph in Zep (Rasmussen et al., 2025), the atomic memory notes in A-MEM (Xu et al., 2025d), and the hierarchical graph-based memory designs in G-Memory (Zhang et al., 2025d) and Mem0 (Chhikara et al., 2025). These systems capture richer relational, temporal, or hierarchical dependencies, enabling more precise retrieval and reasoning. However, their management, spanning insertion, deletion, abstraction, and linkage updates, has thus far been governed by handcrafted rules or heuristic strategies. To date, little work has explored the use of RL to dynamically control the construction, refinement, or evolution of such structured memory, making this an open and promising direction for advancing agentic memory capabilities.

### 3.4 Self-Improvement

As LLM agents evolve, recent research increasingly emphasizes RL as a mechanism for ongoing reflection, enabling agents to learn from their own mistakes across planning, reasoning, tool use, and memory (ang Gao et al., 2025). Rather than relying exclusively on data-driven training phases or static reward models, these systems incorporate *iterative, self-generated feedback loops*, ranging from prompt-level heuristics to fully fledged RL controllers, to guide agents toward continual self-improvement.

**RL for Verbal Self-correction.** Initial methods in this vein leveraged prompt-based heuristics, sometimes referred to as *verbal reinforcement learning*, where agents generate an answer, linguistically reflect on its potential errors, and subsequently produce a refined solution, all within a single inferential pass without gradient updates. Prominent examples include Reflexion (Shinn et al., 2023), Self-refine (Madaan et al., 2023), CRITIC (Gou et al., 2024), and Chain-of-Verification (He et al., 2024). For instance, the Self-Refine (Madaan et al., 2023) protocol directs an LLM to iteratively polish its output using three distinct prompts for generation, feedback, and refinement, proving effective across domains like reasoning and programming. To enhance the efficacy and robustness of such self-reflection, several distinct strategies have been developed: **(I) multiple sampling**, which involves generating multiple output rollouts by sampling from the model's distribution. By aggregating critiques or solutions from multiple attempts, the agent can improve the consistency and quality of its self-reflection. This method has been widely studied in works like If-or-Else (Li et al., 2024b), UALA (Han et al., 2024) and Multi-agent Verification (Lifshitz et al., 2025). This approach is conceptually analogous to test-time scaling techniques, so we refer the reader to (Pignatelli et al., 2024) for more details; **(II) structured reflection workflows**, rather than prompting for a monolithic reflection on a final answer, prescribe a more dedicated and granular workflow. For example, Chain-of-Verification (He et al., 2024) manually decomposes the process into distinct "Retrieving, Rethinking, and Revising" stages; **(III) external guidance**, which grounds the reflection process in verifiable, objective feedback by incorporating external tools. These tools include code interpreters, as seen in Self-Debugging (Chen et al., 2024f), CAD modeling programs in Luban (Guo et al., 2024), mathematical calculators in T1 (Kang et al., 2025b), step-wise reward models (Xiong et al., 2025), and tool-interactive critiquing mechanisms (Gou et al., 2024).

**RL for Internalizing Self-correction.** While verbal self-correction offers a potent inference-time technique, its improvements are ephemeral and confined to a single session. To instill a more durable and generalized capability for self-improvement, subsequent research has employed RL with gradient-based updates to internalize these reflective feedback loops directly into the model's parameters and to fundamentally enhance the model's inherent ability to identify and correct its own errors. This paradigm has been applied across multiple domains. For instance, KnowSelf (Qiao et al., 2025) leverages DPO and RPO (Pang et al., 2024) to enhance agents' self-reflection capabilities in text-based game environments, while Reflection-DPO (Patel et al., 2025) focuses on user–agent interaction scenarios, enabling agents to better infer user intent through reflective reasoning. DuPo (She et al., 2025) employs RL with dual-task feedback to enable annotation-free optimization, enhancing LLM agents' self-correction across translation, reasoning, and reranking tasks. SWEET-RL (Zhou et al., 2025e) and ACC-Collab (Estornell et al., 2025b) adopt a slightly different setting from the above works: they train an external critic model to provide higher-quality revision suggestions for the actor agent's actions. Nonetheless, the underlying principle remains closely aligned.

**RL for Iterative Self-training.** Moving toward full agentic autonomy, the third and most advanced class of models combines reflection, reasoning, and task generation into a self-sustaining loop, enabling unbounded self-improvement without human-labeled data. These methods can be distinguished by the architecture of their learning loops: **(I) Self-play and search-guided refinement**, which emulates classic RL paradigms like AlphaZero. R-Zero (Huang et al., 2025a), for instance, employs a Monte Carlo Tree Search (MCTS) to explore a reasoning tree, using the search results to iteratively train both a policy LLM (the actor) and a value LLM (the critic) entirely from scratch. Similarly, the ISC framework (Tian et al., 2024) operationalizes a cycle of "Imagination, Searching, and Criticizing," where the agent generates potential solution paths, uses a search algorithm to explore them, and applies a critic to refine its reasoning strategy before producing a final answer. **(II) Execution-guided curriculum generation**, where the agent creates its own problems and learns from verifiable outcomes. Absolute Zero (Zhao et al., 2025a) exemplifies this by proposing its own

tasks, attempting solutions, verifying them via execution, and using the outcome-based reward to refine its policy. Similarly, Self-Evolving Curriculum (Chen et al., 2025f) enhances this process by framing problem selection itself as a non-stationary bandit task, allowing the agent to strategically generate a curriculum that maximizes its learning gains over time. TTRL (Zuo et al., 2025) applies this principle for on-the-fly adaptation to a single problem. At test time, it uses execution-based rewards to rapidly fine-tune a temporary copy of the agent's policy for the specific task at hand; this specialized policy is then used to generate the final answer before being discarded. Though differing in whether the learning is permanent or ephemeral, all these methods underscore a powerful, unified strategy: harnessing execution-based feedback to autonomously guide the agent's reasoning process. ALAS (Atreja, 2025) constructs an autonomous pipeline that crawls web data, distills it into training signals, and continuously fine-tunes LLMs, thereby enabling self-training and self-evolution without manual dataset curation. **(III) Collective bootstrapping**, where learning is accelerated by aggregating shared experience. SiriuS (Zhao et al., 2025e), for example, constructs and augments a live repository of successful reasoning trajectories from multi-agent interactions, using this growing knowledge base to bootstrap its own training curriculum. MALT (Motwani et al., 2025) shares a similar motivation, yet its implementation is limited to a three-agent setup. Nevertheless, all these methods define feedback loops that are internally generated and continuously evolving, representing a significant step toward truly autonomous agents.

**Prospective: Meta Evolution of Reflection Ability.** While current research successfully uses RL to refine an agent's behavior through reflection, the reflection process itself remains largely handcrafted and static. The next frontier lies in applying RL at a higher level of abstraction to enable **meta-learning for adaptive reflection**, focusing not just on correcting an error, but on learning how to self-correct more effectively over time. In this paradigm, the agent may learn a meta-policy that governs its own reflective strategies. For instance, it could learn to dynamically choose the most appropriate form of reflection for a given task, deciding whether a quick verbal check is sufficient or if a more costly, execution-guided search is necessary. Furthermore, an agent could use long-term outcomes to evaluate and refine the very heuristics it uses for self-critique, effectively learning to become a better internal critic. By optimizing the reflective mechanism itself, this approach moves beyond simple self-correction and toward a state of continuous self-improvement in the learning process, representing a crucial step toward agents that can not only solve problems but also autonomously enhance their fundamental capacity to learn from experience.

### 3.5 Reasoning

Reasoning in large language models can be broadly categorized into *fast reasoning* and *slow reasoning*, building on the dual-process cognitive theory (Kahneman, 2011; Kahneman & Tversky, 1974; Stanovich & West, 2000), as discussed in recent surveys (Ke et al., 2025; Kumar et al., 2025). Fast reasoning corresponds to rapid, heuristic-driven inference with minimal intermediate steps, while slow reasoning emphasizes deliberate, structured, and multi-step reasoning. Understanding the trade-offs between these two paradigms is crucial for designing models that balance efficiency and accuracy in complex problem-solving.

**Fast Reasoning: Intuitive and Efficient Inference** Fast reasoning models operate in a manner analogous to System 1 (Li et al., 2025r) cognition: quick, intuitive, and pattern-driven. They generate immediate responses without explicit step-by-step deliberation, excelling in tasks that prioritize fluency, efficiency, and low latency. Most conventional LLMs fall under this category, where reasoning is implicitly encoded in next-token prediction (Shao et al., 2024b; Yang et al., 2024a). However, this efficiency comes at the cost of systematic reasoning, making these models more vulnerable to factual errors, biases, and shallow generalization.

To address the severe hallucination problems in fast reasoning, current research has largely focused on various direct approaches. Some studies attempt to mitigate errors and hallucinations in the next-token prediction paradigm by leveraging internal mechanisms (Wang et al., 2023b; Yao et al., 2023a; Besta et al., 2024) or by simulating human-like cognitive reasoning. Other works propose introducing both external and internal confidence estimation methods (Lightman et al., 2023; Wang et al., 2024d) to identify more reliable reasoning paths. However, constructing such external reasoning frameworks often risks algorithmic adaptivity issues and can easily fall into the complexity trap.

**Slow Reasoning: Deliberate and Structured Problem Solving** In contrast, slow reasoning models are designed to emulate System 2 cognition (Li et al., 2025r) by explicitly producing intermediate reasoning traces. Techniques such as chain-of-thought prompting, multi-step verification (Qin et al., 2024a), and reasoning-augmented reinforcement learning allow these models to engage in deeper reflection and achieve greater logical consistency. While slower in inference due to extended reasoning trajectories, they achieve higher accuracy and robustness in knowledge-intensive tasks such as mathematics, scientific reasoning, and multi-hop question answering (Chu et al., 2025a). Representative examples include OpenAI's o1 (OpenAI et al., 2024) and o3 series (OpenAI Team, 2025), DeepSeek-R1 (DeepSeek-AI et al., 2025), as well as methods that incorporate dynamic test-time scaling (Aggarwal & Welleck, 2025; Zhang et al., 2024a; Xu et al., 2025a; Yao et al., 2023a) or reinforcement learning (Zeng et al., 2025c; Yu et al., 2025e; Wang et al., 2025k; Liang et al., 2025a;b; Yue et al., 2025a) for reasoning.

Modern slow reasoning exhibits output structures that differ substantially from fast reasoning. These include a clear exploration and planning structure, frequent verification and checking behaviors, and generally longer inference lengths and times. Past work has explored diverse patterns for constructing long-chain reasoning outputs. Some methods—Macro-o1, HuatuoGPT-o1, and AlphaZero—have attempted to synthesize long chains-of-thought via structured, agentic search (Zhao et al., 2024; Chen et al., 2024c;b). Other approaches focus on generating long-CoT datasets that embody specific deliberative or reflective thinking patterns; examples include HiICL-MCTS, LLaVA-CoT, rStar-Math, and ReasonFlux (Wu et al., 2025d; Xu et al., 2025b; Guan et al., 2025a; Yang et al., 2025d). Recent approaches that perform reasoning in the latent space leverage latent representations to conduct parallel reasoning and explore diverse reasoning trajectories (Zhang et al., 2025x; Hao et al., 2024). With the progress of pretrained foundation models, more recent work has shifted toward self-improvement paradigms—frequently instantiated with reinforcement learning—to further enhance models' reasoning capabilities (Zeng et al., 2025c; Yu et al., 2025e).

**Prospective: Integrating Slow Reasoning Mechanisms into Agentic Reasoning** The dichotomy between fast and slow reasoning highlights an open challenge in agentic reasoning: how to employ reinforcement learning for reliably training slow-thinking reasoning capabilities in agentic scenarios. Reinforcement learning in agentic scenarios faces greater challenges in training stability, such as ensuring compatibility with diverse environments. Agentic reasoning itself is also susceptible to overthinking. Purely fast models may overlook critical reasoning steps, while slow models often suffer from excessive latency or **overthinking behaviors**, such as unnecessarily long chains of thought. Emerging approaches seek hybrid strategies (Yang et al., 2025a) that combine the efficiency of fast reasoning with the rigor of slow reasoning (Yang et al., 2025g; Hou et al., 2025; Li et al., 2025q; Chen et al., 2025g). For instance, adaptive test-time scaling allows a model to decide whether to respond quickly or to engage in extended deliberation depending on task complexity. Developing such cognitively aligned mechanisms is a key step toward building reasoning agents that are both efficient and reliable.

## 3.6 Perception

By bridging visual perception with linguistic abstraction, Large Vision–Language Models (LVLMs) have demonstrated unprecedented capabilities for perceiving and understanding multimodal content (Team et al., 2023; Liu et al., 2023a; Wang et al., 2024e; Li et al., 2024d; Chen et al., 2024j; OpenAI, 2023; Zhang et al., 2025p; 2024b). Central to this progress is the incorporation of explicit reasoning mechanisms into multimodal learning frameworks (Shao et al., 2024a; Zhang et al., 2023), moving beyond passive perception toward active visual cognition (Su et al., 2025c). RL has emerged as a powerful paradigm for this purpose, enabling the alignment of vision–language–action models with complex, multi-step reasoning objectives that go beyond the constraints of supervised next-token prediction (Zhou et al., 2025a; Wu et al., 2025h).

**From Passive Perception to Active Visual Cognition** Multimodal content often requires nuanced, context-dependent interpretation. Inspired by the remarkable success of RL in enhancing reasoning within LLMs (DeepSeek-AI et al., 2025; Team et al., 2025b), researchers have increasingly sought to transfer these gains to multimodal learning (Shen et al., 2025a; Peng et al., 2025). Early efforts focused on preference-based RL to strengthen the Chain-of-Thought (CoT) reasoning ability of MLLMs (Wang et al., 2024g; Dong et al., 2025d; Zhu et al., 2025b). Visual-RFT (Liu et al., 2025v) and Reason-RFT (Tan et al., 2025a)

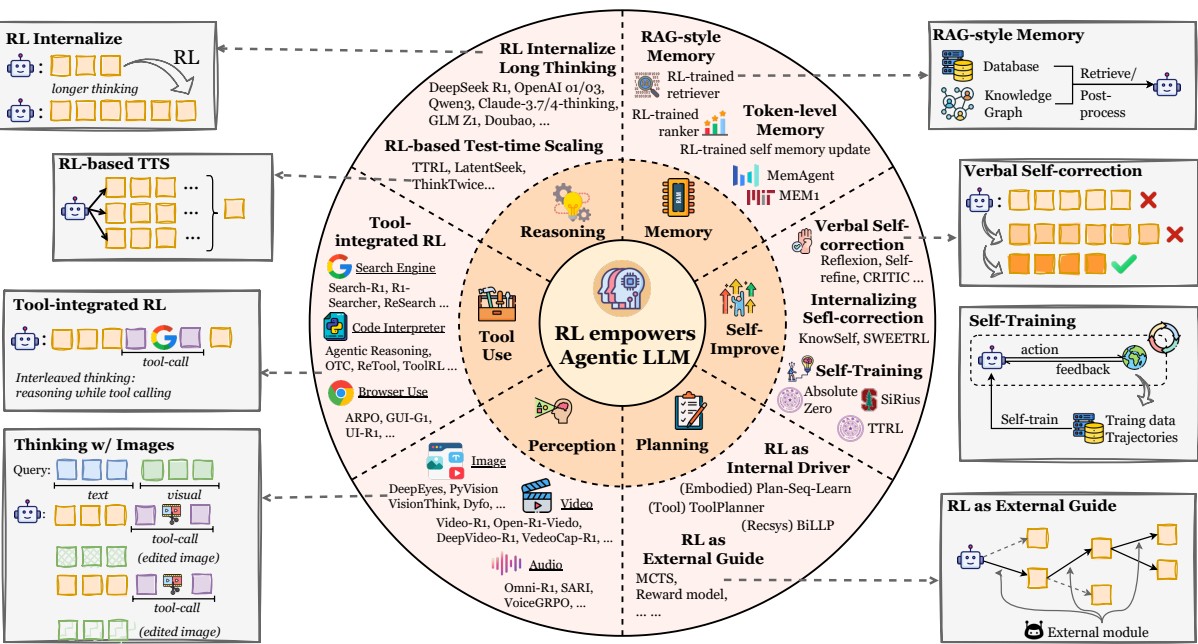

Figure 5: A conceptual overview of how RL empowers agentic LLMs across six core capabilities. The central panel summarizes the capability taxonomy, while the side panels illustrate representative RL mechanisms and interaction patterns. Listed methods are illustrative rather than exhaustive; see the main text for details.

directly apply GRPO to the vision domain, adaptively incorporating vision-specific metrics such as IoU as verifiable reward signals, while STAR-R1 (Li et al., 2025t) extended this idea by introducing partial rewards tailored for visual GRPO. Building upon this, a series of approaches—Vision-R1 (Huang et al., 2025c), VLM-R1 (Shen et al., 2025a), LMM-R1 (Peng et al., 2025), and MM-Eureka (Meng et al., 2025)—developed specialized policy optimization algorithms designed to incentivize step-wise visual reasoning, demonstrating strong performance even on smaller 3B-parameter models. SVQA-R1 (Wang & Ling, 2025) introduced Spatial-GRPO, a novel groupwise RL method that enforces view-consistent and transformation-invariant objectives. Visionary-R1 (Xia et al., 2025a) enforces image captioning as a prerequisite step before reasoning, mitigating shortcut exploitation during reinforcement finetuning. A line of curriculum-learning methods have also been proposed to ease and smooth the RL training process of vision reinforcement finetuning (Yang et al., 2025c; Chen et al., 2025b; Zhan et al., 2025; Guo et al., 2025d; Dong et al., 2025d). R1-V (Chen et al., 2025b) introduces VLM-Gym and trains G0/G1 models via scalable, pure RL self-evolution with a perception-enhanced cold start, yielding emergent perception–reasoning synergy across diverse visual tasks. R1-Zero (Zhou et al., 2025d) shows that even simple rule-based rewards can induce self-reflection and extended reasoning in non-SFT models, surpassing supervised baselines. PAPO (Wang et al., 2025u) proposes a perception-aware policy optimization framework that augments RLVR methods with an implicit perception KL loss and double-entropy regularization, while Li et al. (2025s) proposes a summarize-and-then-reason framework under RL training to mitigate visual hallucinations and improve reasoning without dense human annotations. Collectively, these approaches demonstrate that R1-style RL can be successfully transferred to the vision domain, provided that well-designed, verifiable reward metrics are used—yielding significant improvements in performance, robustness, and out-of-distribution generalization.

More recent work explores another key advantage of RL: moving beyond the formulation of tasks as passive perception, where static, verifiable rewards are computed only on the text-based outputs of LVLMs. Instead, RL can be used to incentivize active cognition over multimodal content—treating visual representations as manipulable and verifiable intermediate thoughts. This paradigm empowers models not merely to "look and answer," but to actively see, manipulate, and reason with visual information as part of a multi-step cognitive process (Su et al., 2025c).

**Grounding-Driven Active Perception.** To advance from passive perception to active visual cognition, a key direction is enabling LVLMs to repeatedly look back and query the image while generating their reasoning process. This is achieved through grounding (Nagaraja et al., 2016; Mao et al., 2016), which anchors each step of the generated chain-of-thought (CoT) to specific regions of the multimodal input—facilitating more valid and verifiable reasoning by explicitly linking text with corresponding visual regions.

To begin with, GRIT (Fan et al., 2025c) interleaves bounding-box tokens with textual CoT and uses GRPO with both verifiable rewards and bounding-box correctness as supervision. Chung et al. (2025) introduces a simple point-and-copy mechanism that allows the model to dynamically retrieve relevant image regions throughout the reasoning process. Ground-R1 (Cao et al., 2025a) and BRPO (Chu et al., 2025c) highlight evidence regions (via IoU-based or reflection rewards) prior to text-only reasoning, while DeepEyes (Zheng et al., 2025g) demonstrates that end-to-end RL can naturally induce such grounding behaviors. Chain-of-Focus further refines this approach by grounding CoT steps followed by zooming in operations.

**Tool-Driven Active Perception.** Another promising direction for enabling active perception is to frame visual cognition as an agentic process, where external tools, code snippets, and runtime environments assist the model's cognitive workflow (Gupta & Kembhavi, 2023; Zhao et al., 2025d). For instance, VisTA (Huang et al., 2025d) and VTool-R1 (Wu et al., 2025g) focus on teaching models how to select and use visual tools through RL, while OpenThinkIMG (Su et al., 2025b) provides standardized infrastructure for training models to "think with images." Finally, Visual-ARFT (Liu et al., 2025v) leverages RL to facilitate tool creation, harnessing the code-generation capabilities of MLLMs to dynamically extend their perceptual toolkit. Pixel Reasoner (Su et al., 2025a) expands the model's action space with operations such as crop, erase, and paint, and introduces curiosity-driven rewards to discourage premature termination of exploration.

**Generation-Driven Active Perception.** In addition to grounding and tool use, humans employ one of their most powerful cognitive abilities—imagination—to produce sketches or diagrams that aid problem-solving. Inspired by this, researchers have begun equipping LVLMs with the ability to generate sketches or images interleaved with chain-of-thought (CoT) reasoning, enabling models to externalize intermediate representations and reason more effectively (Xu et al., 2025e; Fang et al., 2025a; Li et al., 2025c). Visual Planning (Xu et al., 2025e) proposes to use imagined image rollouts only as the CoT images thinking, using downstream task success as the reward signal. GoT-R1 (Duan et al., 2025) applies RL within the Generation-CoT framework, allowing models to autonomously discover semantic–spatial reasoning plans before producing the image. Similarly, T2I-R1 (Jiang et al., 2025b) explicitly decouples the process into a semantic-level CoT for high-level planning and a token-level CoT for patch-wise pixel generation, jointly optimizing both stages with RL.

**Audio.** RL has also been extended beyond vision–language models to a diverse range of modalities, including audio. Within the audio–language domain, we categorize RL applications into two broad classes. (1) Reasoning enhancement for large audio–language models: RL is leveraged to guide models in producing structured, step-by-step reasoning chains for tasks such as audio question answering and logical inference (Wen et al., 2025; Diao et al., 2025; Li et al., 2025d;d; Wen et al., 2025). (2) Fine-grained component optimization in speech synthesis (TTS): RL is employed to directly refine system components—for example, improving duration predictors—using perceptual quality metrics such as speaker similarity and word error rate as reward signals, thereby yielding more natural and intelligible speech (Li et al., 2025m). Some other works such as EchoInk-R1 (Xing et al., 2025) further enrich visual reasoning by integrating audio–visual synchrony under GRPO optimization.

## 3.7 Others

Beyond optimizing the above core cognitive modules, Agentic RL also strengthens the ability to maintain strategic coherence over extended, **multi-turn interactions**. Here, RL is applied to support long-horizon reasoning and effective credit assignment.

For long-horizon interactions, the central challenge is temporal credit assignment (Pignatelli et al., 2024), where sparse and delayed feedback obscures the link between an agent's actions and a distant outcome. Agentic

RL directly confronts this by evolving both the learning signal and the optimization framework. One major approach is the **(I) integration of process-based supervision with final outcome rewards.** Rather than relying on a single reward at a trajectory's conclusion, this paradigm uses *auxiliary models* or *programmatic rules* to evaluate the quality of intermediate steps, providing a denser and more immediate learning signal that guides the agent's multi-turn strategy. For example, EPO (Liu et al., 2025m), ThinkRM (Hong et al., 2025c), SPO (Guo et al., 2025c), and AgentPRM (Choudhury, 2025) introduce external reward models to provide step-wise signals for agents; in contrast, RLVMR (Zhang et al., 2025z) designs manually defined, programmatic rules to guide the intermediate supervision. A second, complementary strategy is to **(II) extend preference optimization from single turns to multi-step segments.** Techniques like Segment-level DPO (SDPO) (Kong et al., 2025) move beyond comparing isolated responses and instead construct preference data over entire conversational snippets or action sequences. This allows the model to directly learn how early decisions influence long-term success, thereby refining its ability to maintain strategic coherence in extended dialogues and complex tasks.

## 4 Agentic RL: The Task Perspective

Agentic RL manifests through a wide spectrum of concrete tasks that test and shape its evolving capabilities. This section surveys representative application domains where Agentic RL has demonstrated remarkable potential and unique challenges. We begin with *search and information retrieval* (Section 4.1), followed by *code generation and software engineering* (Section 4.2), and *mathematical reasoning* (Section 4.3). We then discuss its role in *GUI navigation* (Section 4.4), *vision understanding tasks* (Section 4.5), as well as *VLM embodied interaction* (Section 4.6). Beyond single-agent scenarios, we extend the perspective to *multi-agent systems* (Section 4.7) and conclude with other emerging domains (Section 4.8). Together, these applications highlight how Agentic RL transitions from abstract paradigms into actionable, real-world problem-solving, as illustrated in Figure 6.

### 4.1 Search & Research Agent

Search has been central to extending LLMs with external knowledge, with Retrieval-Augmented Generation (RAG) as a widely used approach (Gao et al., 2024; Fan et al., 2024). The paradigm is now evolving beyond simple information retrieval towards creating autonomous agents capable of *deep research*: complex, multi-step processes that involve not just finding information but also performing in-depth analysis, synthesizing insights from numerous sources, and drafting comprehensive reports (Kimi, 2025; Perplexity, 2025). This shift elevates the objective from answering queries to tackling complex research tasks. Early prompt-driven methods relied on brittle query strategies and manual engineering. While more recent works like Search-o1 (Li et al., 2025i) leverage large reasoning models for agentic, inference-time retrieval, and multi-agent systems such as DeepResearch (Zhang et al., 2025r) coordinate querying and summarization sub-agents, they still lack learning signals. These prompt-based methods lack any fine-tuning signal, leading to limited generalization and poor effectiveness in multi-turn settings that demand a tight loop of search, reasoning, and synthesis. These limitations have led to the adoption of reinforcement learning to directly optimize the end-to-end process of query generation and search–reasoning coordination for advanced research objectives. Table 4 presents the majority of works studied in this section. In the following, we will detail how RL empowers these agents.

#### 4.1.1 Open Source RL Methods

**Search from the external Internet** A major line of work builds on the RAG foundation but relies on *real-time web search APIs* as the external environment, using reinforcement learning to optimize query generation and multi-step reasoning. Early progress was spearheaded by DeepRetrieval (Jiang et al., 2025c), which framed one-shot query generation as a GRPO-trained policy and directly rewarded recall and relevance against live search results. Motivated by its gains, subsequent methods extended the paradigm into multi-turn, reasoning-integrated, and multi-modal search. Search-R1 (Jin et al., 2025a) and DeepResearcher (Zheng et al., 2025e) integrate retrieved-token masking with outcome-based rewards to interleave query formulation and answer generation. AutoRefine (Shi et al., 2025b) further advances this trajectory by inserting refinement

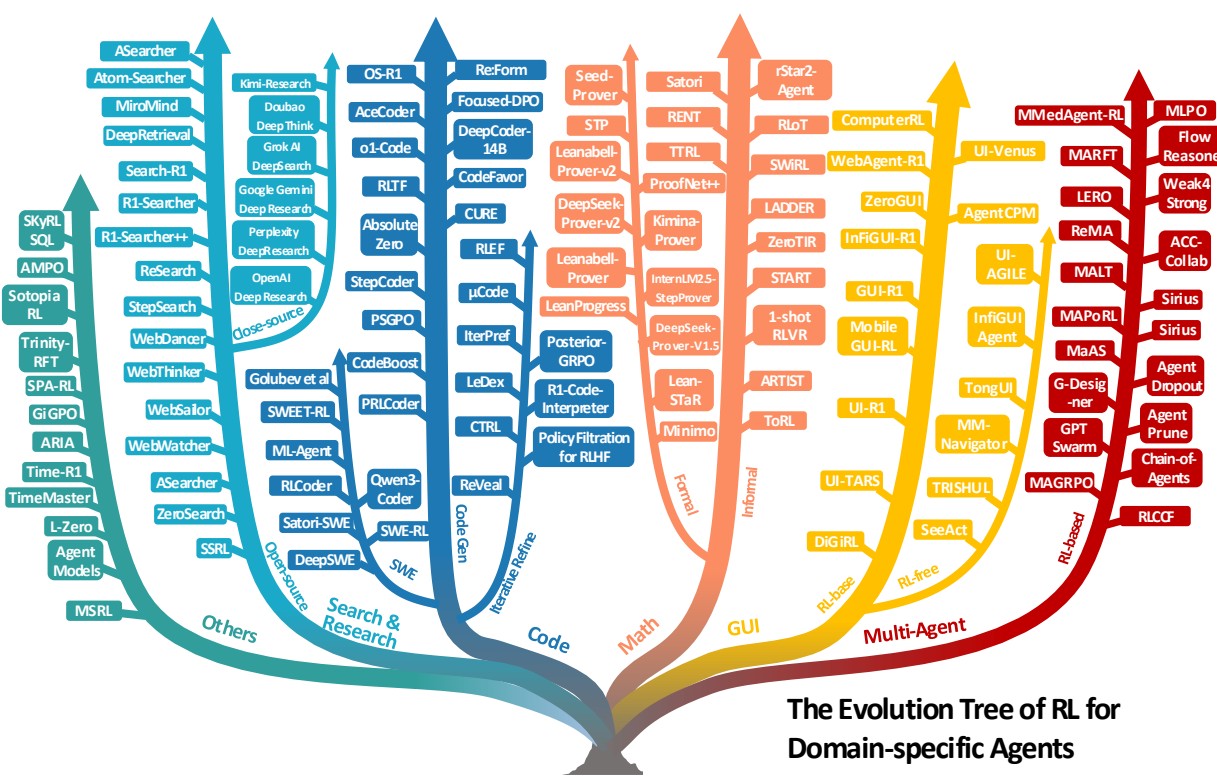

Figure 6: The evolution tree of RL for domain-specific agents, illustrating the chronological progression of representative domains and methods.

phases between successive search calls, using GRPO to reward not only answer correctness but also retrieval quality, enabling agents to iteratively filter and structure noisy evidence during long-horizon reasoning. R1-Searcher (Song et al., 2025a) employs a two-stage, cold-start PPO strategy—first learning when to invoke web search, then how to exploit it—while its successor R1-Searcher++ (Song et al., 2025b) adds supervised fine-tuning, internal-knowledge rewards to avoid redundancy, and dynamic memory for continual assimilation. ReSearch (Chen et al., 2025d) pursues fully end-to-end PPO without supervised tool-use trajectories, while StepSearch (Wang et al., 2025w) accelerates convergence on multi-hop QA by assigning intermediate step-level rewards. Atom-Searcher (Deng et al., 2025b) is an agentic deep research framework that significantly improves LLM problem-solving by refining the reasoning process itself, not just the final outcome. WebDancer (Wu et al., 2025a) leverages human browsing trajectory supervision plus RL fine-tuning to produce autonomous ReAct-style agents, excelling on GAIA (Mialon et al., 2024) and WebWalkerQA (Wu et al., 2025b). WebThinker (Li et al., 2025j) embeds a Deep Web Explorer into a think-search-draft loop, aligning via DPO with human feedback to tackle complex report-generation. WebSailor (Li et al., 2025f) is a complete post-training methodology designed to teach LLM agents sophisticated reasoning for complex web navigation and information-seeking tasks. WebWatcher (Geng et al., 2025) further extends to multimodal search, combining visual-language reasoning, tool use, and RL to outperform text-only and multimodal baselines on BrowseComp-VL and VQA benchmarks. ASearcher (Gao et al., 2025c) uses large-scale asynchronous reinforcement learning with synthesized QA data, enabling long-horizon search (40+ tool calls) and outperforming prior open-source methods. MiroMind Open Deep Research (MiroMind ODR) (MiroMind Team, 2025) aims to build a high-performance, fully open-sourced, open-collaborative deep research ecosystem — with an agent framework, model, data, and training infrastructure all fully accessible and open.

**Search from LLM internal knowledge**   However, these training methods that rely on external APIs face two major challenges: (1) the document quality of real-time internet document searching is uncontrolled, and

Table 4: A summary of RL-based methods for search and research agents.

| Method | Category | Base LLM | Resource Link |
|---|---|---|---|
| *Open Source Methods* | | | |
| DeepRetrieval (Jiang et al., 2025c) | External | Qwen2.5-3B-Instruct, Llama-3.2-3B-Instruct | ⭘ GitHub |
| Search-R1 (Jin et al., 2025a) | External | Qwen2.5-3B/7B-Base/Instruct | ⭘ GitHub |
| R1-Searcher (Song et al., 2025a) | External | Qwen2.5-7B, Llama3.1-8B-Instruct | ⭘ GitHub |
| R1-Searcher++ (Song et al., 2025b) | External | Qwen2.5-7B-Instruct | ⭘ GitHub |
| ReSearch (Chen et al., 2025d) | External | Qwen2.5-7B/32B-Instruct | ⭘ GitHub |
| StepSearch (Wang et al., 2025w) | External | Qwen2.5-3B/7B-Base/Instruct | ⭘ GitHub |
| DeepResearcher (Zheng et al., 2025e) | External | Qwen2.5-7B-Instruct | ⭘ GitHub |
| WebDancer (Wu et al., 2025a) | External | Qwen2.5-7B/32B, QWQ-32B | ⭘ GitHub |
| WebThinker (Li et al., 2025j) | External | QwQ-32B, DeepSeek-R1-Distilled-Qwen, Qwen2.5-32B | ⭘ GitHub |
| WebSailor (Li et al., 2025f) | External | Qwen2.5-3B/7B/32B/72B | ⭘ GitHub |
| WebWatcher (Geng et al., 2025) | External | Qwen2.5-VL-7B/32B | ⭘ GitHub |
| WebShaper (Tao et al., 2025) | External | Qwen-2.5-32B/72B, QwQ-32B | ⭘ GitHub |
| ASearcher (Gao et al., 2025c) | External | Qwen2.5-7B/14B, QwQ-32B | ⭘ GitHub |
| Atom-Searcher (Deng et al., 2025b) | External | Qwen2.5-7B-Instruct | ⭘ GitHub |
| MiroMind Open Deep Research (MiroMind Team, 2025) | External | - | ⊕ Website |
| SimpleDeepResearcher (Sun et al., 2025b) | External | QwQ-32B | ⭘ GitHub |
| AWorld (Yu et al., 2025a) | External | Qwen3-32B | ⭘ GitHub |
| SFR-DeepResearch (Nguyen et al., 2025b) | External | QwQ-32B, Qwen3-8B, GPT-oss-20b | - |
| ZeroSearch (Sun et al., 2025a) | Internal | Qwen2.5-3B/7B-Base/Instruct | ⭘ GitHub |
| SSRL (Fan et al., 2025b) | Internal | Qwen2.5, Llama-3.2/Llama-3.1, Qwen3 | ⭘ GitHub |
| *Closed Source Methods* | | | |
| OpenAI Deep Research (OpenAI, 2025) | External | OpenAI Models | ⊕ Website |
| Perplexity's DeepResearch (Perplexity, 2025) | External | - | ⊕ Website |
| Google Gemini's DeepResearch (Google, 2025) | External | Gemini | ⊕ Website |
| Kimi-Researcher (Kimi, 2025) | External | Kimi K2 | ⊕ Website |
| Grok AI DeepSearch (x.ai, 2025) | External | Grok3 | ⊕ Website |
| Doubao with Deep Think (Doubao, 2025) | External | Doubao | ⊕ Website |
| Manus WideResearch | External | - | ⊕ Website |

noisy information brings instability to the training process. (2) The API cost is too high and severely limits scalability. To enhance the efficiency, controllability, and stability of training, some recent studies have used controllable simulated search engines such as LLM internal knowledge. For example, ZeroSearch (Sun et al., 2025a) replaces live web retrieval with a pseudo search engine distilled from LLMs themselves, combining curriculum RL to gradually approach live-engine performance without issuing real queries. SSRL (Fan et al., 2025b) takes this idea further: the agent performs entirely offline "self-search" during training, without explicit search engines, yet transfers seamlessly to online inference, where real APIs can still boost performance. Though still at an early stage, offline self-search enhances stability and scalability beyond API limits, pointing toward more self-reliant research agents.

### 4.1.2 Closed Source RL Methods

**Industrial Research Agents.** Despite progress in combining RAG and RL, most open-source models still fail on OpenAI's BrowseComp (Wei et al., 2025b), a challenging benchmark that measures the ability of AI agents to locate hard-to-find information, revealing gaps in long-horizon planning, page-grounded tool use, and cross-source verification. In contrast, recent closed source systems are markedly stronger, having shifted from mere query optimization to fully autonomous research agents that navigate the open web, synthesize information from multiple sources, and draft comprehensive reports. This is likely due to the industry's more powerful foundation models and the availability of more high-quality data. OpenAI Deep Research (OpenAI, 2025) achieves 51.5% pass@1 on BrowseComp. Other prototypes, Perplexity's DeepResearch (Perplexity, 2025), Google Gemini's DeepResearch (Google, 2025), Kimi-Researcher (Kimi, 2025), Grok AI DeepSearch (x.ai, 2025), Doubao with Deep Think (Doubao, 2025), combine RL-style fine-tuning with advanced tool integration and memory modules, ushering in a new era of interactive, iterative research assistants.

**Case Study: OpenAI Deep Research.** Deep Research provides a concrete example of how capabilities from Section 3 combine with the RL-shaped search strategies. The agent begins with long-horizon multi-step

reasoning and planning, decomposing a user request into sub-goals. It then performs RL-shaped web search: issuing queries, selecting which pages to open, and refining its search trajectory. These search policies are shaped during training using research-oriented benchmarks such as BrowseComp (Wei et al., 2025b). Throughout the process, the agent maintains persistent memory in the form of scratchpad notes and performs cross-source verification before synthesis. These capabilities—reasoning, planning, tool use, memory, and verification—are coupled with RL-shaped control decisions over search depth, branch selection, and evidence integration, forming a unified research agent.

## 4.2 Code Agent

Code generation, or more broadly, software engineering, provides an ideal testbed for LLM-based Agentic RL: execution semantics are explicit and verifiable, and automated signals (compilation, unit tests, and runtime traces) are readily available (Dong et al., 2025b). Early multi-agent frameworks (*e.g.*, MetaGPT, AutoGPT, AgentVerse) coordinated roles through prompting without parameter updates, showcasing the promise of modular role allocation (Hong et al., 2024b; Gravitas, 2023; Chen et al., 2024e). Initial RL for code, such as CodeRL, incorporated execution-based reward modeling and actor–critic training (Le et al., 2022), catalyzing a wave of studies that exploit execution feedback to guide policy updates. Table 5 presents the majority of works studied in this section. We structure the literature along increasing *task complexity*, progressing from *code generation* (Section 4.2.1) to *code refinement* (Section 4.2.2) and *software engineering* (Section 4.2.3).

### 4.2.1 RL for Code Generation

Early research focused on relatively simple, single-round code generation (*e.g.*, completing a function or solving a coding challenge in one go), which lays the foundation for subsequent large-scale software engineering.

**Outcome reward RL.** Methods in this class optimize directly for final correctness, typically measured by pass@k or unit-test success. AceCoder (Zeng et al., 2025b) introduces a data-efficient RLHF pipeline for code generation, constructing large-scale preference pairs from existing code fragments to train a reward model via Bradley–Terry loss, which then guides RFT on the synthesized dataset. Beyond early actor–critic formulations, recent open-source efforts scale outcome-based RL on large pre-trained code models. DeepCoder-14B (Luo et al., 2025c) stabilizes GRPO training via iterative context lengthening and DAPO-inspired filtering, and employs a sparse Outcome Reward Model (ORM) to prevent reward hacking on curated coding data. RLTF employs an online RL loop that uses unit test results as multi-granularity reward signals, from coarse pass/fail outcomes to fine-grained fault localization, directly guiding code refinement (Liu et al., 2023b). CURE formalizes coder–tester co-evolution: a tester generates or evolves unit tests while a coder iteratively patches code; a reward-precision objective mitigates low-quality test effects during joint training (Wang et al., 2025q). Absolute Zero applies self-play RL without human data. It generates coding tasks for itself and uses execution outcomes as verifiable rewards to bootstrap reasoning ability (Zhao et al., 2025a). Re:Form (Yan et al., 2025a) leverages formal language-based reasoning with RL and automated verification to reduce human priors, enabling reliable program synthesis and surpassing strong baselines on formal verification tasks. In (Feng et al., 2025c), the authors propose a two-stage training pipeline: first fine-tuning for a high-correctness baseline, then performing efficiency-driven online RL optimization.

**Process reward RL.** To mitigate sparsity and credit assignment, several works design process-level supervision by integrating compilation and execution feedback. StepCoder (Dou et al., 2024) decomposes compilation and execution into step-level signals for shaping; Process Supervision-Guided Policy Optimization (PSGPO) (Dai et al., 2025) leverages intermediate error traces and process annotations for dense rewards; and CodeBoost (Wang et al., 2025n) mines raw repositories to unify heterogeneous execution-derived signals, ranging from output correctness to error-message quality, under a single PPO framework. Further, PRLCoder (Ye et al., 2025b) introduces process-supervised RL by constructing reward models that score each partial snippet: a teacher model mutates lines of reference solutions and assigns positive/negative signals based on compiler and test feedback. This fine-grained supervision yields faster convergence and +10.5% pass-rate improvements over the base model, illustrating how dense shaping at the line-level can guide code synthesis more effectively than outcome-only signals. o1-Coder (Zhang et al., 2024d) combines RL with

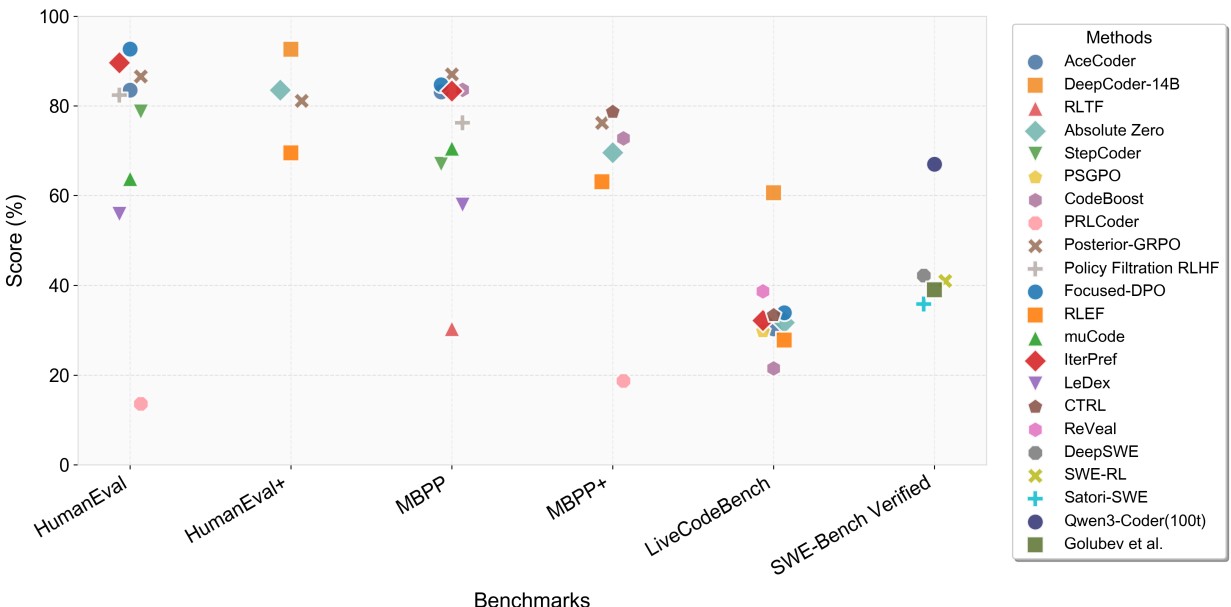

Figure 7: Benchmark Performance of RL-Enhanced Code and SWE Methods. Scores are pass@1 unless otherwise specified.

Monte Carlo Tree Search, where the policy learns from exploration guided by test case rewards and gradually improves from pseudocode to executable code. Posterior-GRPO (Fan et al., 2025a) rewards intermediate reasoning but gates credit by final test success to prevent speculative reward exploitation; Policy Filtration for RLHF (Zhang et al., 2025c) improves reward-correctness alignment by filtering low-confidence pairs before policy updates. Scaling preference supervision beyond costly human annotation has proven effective as well. CodeFavor (Liu et al., 2024a) constructs CodePrefBench from code evolution histories, covering correctness, efficiency, security, and style to improve preference modeling and alignment. Focused-DPO (Zhang et al., 2025o) adapts preference-based RL by weighting preference optimization on error-prone regions of the code, making feedback more targeted and improving robustness across benchmarks. Yang et al. (2025f) studies how RL-trained small-scale agents surpass large-scale prompt-based models in MLE environments via duration-aware gradient updates in a distributed asynchronous RL.

### 4.2.2 RL for Iterative Code Refinement

A second line of research targets more complex coding tasks that require debugging and iterative refinement. In these scenarios, an agent may need multiple attempts to improve solutions, using feedback from human requirements or failed test results, which is closer to real-world tasks.

**Outcome reward RL.** A representative line treats the entire refinement loop as a trajectory while optimizing for final task success. RLEF (Gehring et al., 2025) (Reinforcement Learning from Execution Feedback) grounds correction loops in real error messages as context while optimizing for ultimate pass rates; this reduces the number of attempts needed and improves competitive-programming performance relative to single-shot baselines. $\mu$Code (Jain et al., 2025a) jointly trains a generator and a learned verifier under single-step reward feedback, showing that verifier-guided outcome rewards can outperform purely execution-feedback baselines. R1-Code-Interpreter (Chen et al., 2025h) harnesses multi-turn supervised fine-tuning and reinforcement learning to train LLMs to decide when and how to invoke a code interpreter during step-by-step reasoning.

**Process reward RL.** Process-supervised approaches explicitly guide *how* the model debugs. IterPref (Wu et al., 2025c) constructs localized preference pairs from iterative debugging traces and applies targeted prefer-

Table 5: A summary of RL methods for code and software engineering agents.

| Method | Reward | Base LLM | Resource |
|---|---|---|---|
| *RL for Code Generation* | | | |
| AceCoder (Zeng et al., 2025b) | Outcome | Qwen2.5-Coder-7B-Base/Instruct, Qwen2.5-7B-Instruct | ⭕ GitHub |
| DeepCoder-14B (Luo et al., 2025c) | Outcome | DeepSeek-R1-Distilled-Qwen-14B | ⭕ GitHub |
| RLTF (Liu et al., 2023b) | Outcome | CodeGen-NL 2.7B, CodeT5-770M | ⭕ GitHub |
| CURE (Wang et al., 2025q) | Outcome | Qwen2.5-7B/14B-Instruct, Qwen3-4B | ⭕ GitHub |
| Absolute Zero (Zhao et al., 2025a) | Outcome | Qwen2.5-7B/14B, Qwen2.5-Coder-3B/7B/14B, Llama-3.1-8B | ⭕ GitHub |
| StepCoder (Dou et al., 2024) | Process | DeepSeek-Coder-Instruct-6.7B | ⭕ GitHub |
| PSGPO (Dai et al., 2025) | Process | Qwen2.5-Coder-7B-Instruct | - |
| CodeBoost (Wang et al., 2025n) | Process | Qwen2.5-Coder-7B-Instruct, Llama-3.1-8B-Instruct, Seed-Coder-8B-Instruct, Yi-Coder-9B-Chat | ⭕ GitHub |
| PRLCoder (Ye et al., 2025b) | Process | CodeT5+, Unixcoder, T5-base | - |
| o1-Coder (Zhang et al., 2024d) | Process | DeepSeek-1.3B-Instruct | ⭕ GitHub |
| Posterior-GRPO (Fan et al., 2025a) | Process | Qwen2.5-Coder-3B-Base, Qwen2.5-Coder-7B-Instruct, Qwen2.5-Math-7B | - |
| Policy Filtration for RLHF (Zhang et al., 2025c) | Process | DeepSeek-Coder-6.7B, Qwen1.5-7B | ⭕ GitHub |
| CodeFavor (Liu et al., 2024a) | Process | Mistral-NeMo-12B-Instruct, Gemma-2-9B-Instruct, Llama-3-8B-Instruct, Mistral-7B-Instruct-v0.3 | ⭕ GitHub |
| Focused-DPO (Zhang et al., 2025o) | Process | DeepSeek-Coder-6.7B-Base/Instruct, Magicoder-S-DS-6.7B, Qwen2.5-Coder-7B-Instruct | - |
| Re:Form (Yan et al., 2025a) | Outcome | Qwen2.5 (0.5B–14B) | ⭕ GitHub |
| Qwen Team (Feng et al., 2025c) | Outcome | Qwen2.5-Coder-7B/32B-Instruct | - |
| *RL for Iterative Code Refinement* | | | |
| RLEF (Gehring et al., 2025) | Outcome | Llama-3.0-8B-Instruct, Llama-3.1-8B/70B-Instruct | - |
| μCode (Jain et al., 2025a) | Outcome | Llama-3.1-8B-Instruct, Llama-3.2-1B-Instruct | ⭕ GitHub |
| R1-Code-Interpreter (Chen et al., 2025h) | Outcome | Qwen2.5-7B/14B-Instruct-1M, Qwen2.5-3B-Instruct | ⭕ GitHub |
| IterPref (Wu et al., 2025c) | Process | Deepseek-Coder-7B-Instruct, Qwen2.5-Coder-7B, CodeQwen1.5-7B-Chat, StarCoder2-15B | - |
| LeDex (Jiang et al., 2024) | Process | StarCoder-15B, CodeLlama-7B/13B | - |
| CTRL (Xie et al., 2025) | Process | Qwen2.5-Coder-7B/14B/32B-Instruct | ⭕ GitHub |
| ReVeal (Jin et al., 2025c) | Process | DAPO-Qwen-32B | - |
| *RL for Automated Software Engineering (SWE)* | | | |
| DeepSWE (Luo et al., 2025b) | Outcome | Qwen3-32B | ⭕ GitHub |
| SWE-RL (Wei et al., 2025e) | Outcome | Llama-3.3-70B-Instruct | ⭕ GitHub |
| Satori-SWE (Zeng et al., 2025a) | Outcome | Qwen2.5-Coder-32B-Instruct | ⭕ GitHub |
| RLCoder (Wang et al., 2025p) | Outcome | CodeLlama-7B, StarCoder-7B, StarCoder2-7B, DeepSeekCoder-1B/7B | ⭕ GitHub |
| Qwen3-Coder (Team, 2025b) | Outcome | Qwen3-Coder-480B-A35B-Instruct | ⭕ GitHub |
| ML-Agent (Liu et al., 2025q) | Outcome | Qwen2.5-7B-Base/Instruct, DeepSeek-R1-Distill-Qwen-7B | ⭕ GitHub |
| OS-R1 (Lin et al., 2025b) | Outcome | Qwen2.5-3B/7B-Instruct | ⭕ GitHub |
| Golubev et al. (2025) | Process | Qwen2.5-72B-Instruct | - |
| SWEET-RL (Zhou et al., 2025e) | Process | Llama-3.1-8B/70B-Instruct | ⭕ GitHub |

ence optimization to penalize faulty regions, improving correction accuracy with minimal collateral updates. LeDex (Jiang et al., 2024) couples explanation-driven diagnosis with self-repair: it automatically curates explanation–refinement trajectories and applies dense, continuous rewards to jointly optimize explanation quality and code correctness via PPO, yielding consistent pass@1 gains over SFT-only coders. Beyond explanation-driven shaping, some works like CTRL (Xie et al., 2025) explicitly train separate critic models to evaluate each attempted refinement and provide gradient signals to the policy, though at the cost of added inference overhead. ReVeal (Jin et al., 2025c) extends process-level refinement into a self-evolving agent that autonomously generates tests and learns from per-turn rewards to enhance reasoning and recovery from errors.

### 4.2.3 RL for Automated Software Engineering

**Outcome reward RL.** End-to-end training in realistic environments demonstrates that sparse—but validated—success signals can scale. DeepSWE performs large-scale RL on software engineering missions using verified task completion as the sole reward, achieving leading open-source results on SWE-bench–style evaluations (Luo et al., 2025b). SWE-RL extracts rule-based, outcome-oriented signals from GitHub commit histories, enabling training on authentic improvement patterns and generalization to unseen bug-fixing tasks (Wei et al., 2025e). Satori-SWE introduces an evolutionary RL-enabled test-time scaling method (EvoScale) that trains models to self-improve generations across iterations for sample-efficient software engineering tasks (Zeng et al., 2025a). OS-R1 (Lin et al., 2025b) presents a rule-based reinforcement learning framework for Linux kernel tuning, enabling efficient exploration, accurate configuration, and superior performance over heuristic methods. RLCoder frames retrieval-augmented repository-level code completion as an RL problem, using perplexity-based feedback to train a retriever to fetch helpful context without labeled data (Wang et al., 2025p). Qwen3-Coder performs large-scale execution-driven reinforcement learning on long-horizon, multi-turn interactions across 20,000 parallel environments, yielding state-of-the-art performance

on benchmarks like SWE-Bench Verified (Team, 2025b). In machine learning domains, ML-Agent executes multi-step pipelines (*e.g.*, automated ML), optimizing performance-based terminal rewards (Liu et al., 2025q).

**Process reward RL.** Dense supervision over agentic trajectories improves credit assignment across many steps. From the optimization perspective, long-context, multi-turn software agents benefit from stabilized policy-gradient variants; e.g., Decoupled Clip and Dynamic sAmpling Policy Optimization (DAPO) improves training stability and performance on SWE-bench Verified through multi-turn code generation and debugging interactions, leveraging long-context feedback (Golubev et al., 2025). SWEET-RL trains multi-turn agents on ColBench (backend and frontend tasks), leveraging privileged information during RL to reduce exploration noise and improve long-horizon generalization (Zhou et al., 2025e).

**Remark on closed-source systems.** Commercial systems such as OpenAI's Codex and Anthropic's Claude Code have emphasized preference-aligned fine-tuning and reinforcement learning to improve usefulness and safety in code generation and editing workflows (OpenAI, 2025a; Anthropic, 2025). While concrete training details are limited publicly, these systems underscore the growing role of RL in aligning agentic behavior with developer-centric objectives in practical IDE and terminal environments.

### 4.2.4 Emerging Paradigms

**Code World Models** A recent paradigm shift departs from traditional neural approximations by framing the world model itself as executable code. In these Code World Models (CWMs), agents synthesize programs to explicitly define transition and reward dynamics, enabling model-based planning via verifiable, symbolic simulation rather than opaque latent states.

GIF-MCTS (Dainese et al., 2024) formulates world-model construction as program induction: the LLM edits an "Environment" class and a search procedure selects versions that best explain offline transitions, yielding executable models suitable for downstream planning. WorldCoder (Tang et al., 2024) represents dynamics and rewards as explicit Python functions and refines them through an iterative synthesize–repair process guided by transition consistency and optimism constraints. Meta's 32B CWM (team et al., 2025) strengthens this paradigm by providing an open-weights model trained on interpreter traces and agentic trajectories to improve program synthesis and execution fidelity. Recent work further applies CWMs to general game environments (Lehrach et al., 2025), where an LLM induces complete rule-based simulators and planning is performed directly on the executable model.

Implementing these programmatic world-model paradigms often incurs substantial inference cost, since agents repeatedly synthesize, refine, and query executable simulators. In practice, low-bit quantization (e.g., 8-bit for workstation GPUs like RTX 4500 Ada or 4-bit for consumer hardware) is frequently adopted to make large code-oriented models feasible to deploy. Collectively, CWMs establish programmatic world models as a coherent direction for code agents, coupling LLM-based program synthesis with structured, verifiable simulation for model-based reasoning.

## 4.3 Mathematical Agent

Mathematical reasoning is widely regarded as a gold standard for assessing LLM agents' reasoning ability, owing to its symbolic abstraction, logical consistency, and long-horizon deductive demands. We structure the research efforts around two complementary paradigms: *informal reasoning* (Section 4.3.1), which operates without formal verification support and includes natural-language reasoning and programming-language tool use; and *formal reasoning* (Section 4.3.2), which relies on rigorously specified formal languages and proof assistants.

We note that RLVR methods such as DAPO (Yu et al., 2025e), GRPO (Ren et al., 2025), and GRESO (Zheng et al., 2025b) have consistently played a substantial role in recent enhancements of mathematical reasoning in LLMs. However, given their broader relevance across reasoning tasks, we discuss them in Section 2.7, instead of elaborating here.

### 4.3.1 RL for Informal Mathematical Reasoning

Informal mathematics essentially refers to reasoning and expression in natural language. Such reasoning may incorporate symbols or function names, but no finite and explicit set of logical rules defines their syntactic validity, and no formal semantics precisely determines their interpretation and meaning (Yang et al., 2024b; Asperti et al., 2025).

While informal mathematical reasoning relaxes strict rigor at the detail level, it affords greater expressive flexibility and better captures the high-level structure of arguments. This makes it particularly suited for a variety of math tasks such as mathematical word-problem solving, equation manipulation, and symbolic computation (Singh et al., 2025; Mai et al., 2025). Although general-purpose programming languages are symbolic, they lack the rigor and formal semantics of proof-assistant languages, and are therefore regarded as informal when applied to mathematical reasoning (Yang et al., 2024b), typically through tool invocation of executors such as Python with numerical or symbolic libraries.

**Outcome reward RL.** Outcome-only methods define rewards solely by final numerical or symbolic correctness (e.g., algebraic equations) during RL. Empirically, such training often leads to emergent agentic behaviors, including adaptive tool use interleaved with natural language reasoning. ARTIST (Singh et al., 2025) introduces a framework for tool-integrated agentic reasoning, interleaving tool invocations, e.g. code execution, directly within the reasoning chain. Trained with outcome-only rewards, it achieves strong performance and observes emergent agentic behaviors, including self-reflection, and context-aware CoT, which further shows that by integrating dynamic tool use with RL, agentic tool-integrated reasoning could learn optimal strategies for interacting with environments, highlighting the potential of RL to internalize tool-integrated reasoning strategies in LLMs. Similarly, ToRL (Li et al., 2025l) improves performance by exploiting the scaling of tool-integrated reasoning RL and encouraging code execution behaviour, and experiments show emergent cognitive behaviors, such as adaptive tool-use, self-correction based on tool feedback, and adaptive computational reasoning. ZeroTIR (Mai et al., 2025) investigates the scaling law of RL from outcome-based rewards for Tool-Integrated Reasoning with Python code execution settings, revealing a strong correlation between training computational effort and the spontaneous code execution frequency, the average response length, and the final task accuracy, which corroborates the empirical emergence of tool-integrated reasoning strategies. TTRL (Zuo et al., 2025) leverages majority voting to estimate rewards, enabling training on unlabeled data. Fine-tuned on these majority-vote rewards, it not only surpasses the base model's maj@n accuracy but also achieves an empirical performance curve and upper bound that, surprisingly, closely approach those of direct RL training with labeled test answers on MATH-500, underscoring its practical value and potential. However, RENT (Prabhudesai et al., 2025) suggests that majority voting is limited in generalization, it applies only to questions with deterministic answers, and will not work on free-response outputs. To address this limitation, it extends the entropy minimization idea (Wang et al., 2021) to RL, using the token-level average negative entropy as a reward to guide learning, achieving improvements on an extensive suite of benchmarks including math problem solving, suggesting that confidence-based reward shaping can serve as a path toward continual improvement. Alternatively, Satori (Shen et al., 2025b) proposes Chain-of-Action-Thought (COAT), a variant of CoT that explicitly integrates action choices, and modularizes reasoning into 3-fold meta-actions, including continuation, reflection, and exploration of alternatives, and internalizes this behavior via RL with outcome-only rewards. In particular, 1-shot RLVR (Wang et al., 2025r) studies data efficiency of outcome-only RL with verifier signals. Surprisingly, they found that RL with only 1 example performs close to using a 1.2k-example dataset, and with 2 examples comes close to using the 7.5k MATH training dataset. They also highlight an intriguing phenomenon, named post-saturation generalization, that test accuracy continues to improve even after the training accuracy on the single example approaches 100%. In addition to correctness, hallucination remains a major challenge in informal mathematical reasoning, motivating methods that explicitly promote trustworthiness. For instance, Kirchner et al. (2024) propose a game-theoretic training algorithm that jointly optimizes for both correctness and legibility. Inspired by Prover-Verifier Games (Anil et al., 2021), the method alternates between training a small verifier that predicts solution correctness, a "helpful" prover that generates solutions accepted by the verifier, and a "sneaky" prover that aims to fool it. Empirically, this increases the helpful prover accuracy, verifier robustness and legibility (measured by human accuracy in time-constrained verification tasks). This result suggests that verifier-guided legibility optimization can enhance the interpretability and trustworthiness of LLM-generated informal

reasoning. Recent rStar2-Agent (Shang et al., 2025a) is a 14B-parameter math reasoning model trained with agentic reinforcement learning using a high-throughput Python execution environment, a novel GRPO-RoC algorithm to resample on correct rollouts amid tool-noise, and a multi-stage training recipe—achieving state-of-the-art results in just 510 RL steps, achieving average pass@1 scores of 80.6% on AIME24 and 69.8% on AIME25.

**Process reward RL.** Process-aware methods leverage intermediate evaluators (e.g. unit tests, assertions, sub-task checks) to provide denser feedback, shaping credit assignment and improving tool-integrated reasoning (TIR). START (Li et al., 2025b) guides TIR by injecting handcrafted hint text into Long CoT traces, typically after conjunction words or before the CoT stop token, to encourage code executor calls during inference. This enables test-time scaling that improves reasoning accuracy. The collected trajectories are then used to fine-tune the model, internalizing the tool-invocation behavior. LADDER (Simonds & Yoshiyama, 2025) introduces a training-time framework where an LLM recursively generates and solves progressively simpler variants of a complex problem, using verifiable reward signals to guide a difficulty-based curriculum, and achieves substantial improvements in mathematical reasoning. An additional test-time RL step (TTRL) further enhances performance. The authors suggest that this approach of self-generated curriculum learning with verifiable feedback may generalize beyond informal mathematical tasks to any domain with reliable automatic verification. To improve performance on complex problems, SWiRL (Goldie et al., 2025) synthesizes step-wise tool use reasoning data by iteratively decomposing solutions, and then adopts a preference-based step-wise RL approach to fine-tune the base model on the multi-step trajectories. While many of these approaches exploit inference-time interventions, they often suffer from generalization limitations due to their reliance on manually designed logical structures. To overcome this, RLoT (Hao et al., 2025b) instead trains a lightweight navigator agent model with RL to adaptively enhance reasoning, showing improved generalization across diverse tasks.

While informal approaches excel at word problems and symbolic computations, they struggle to extend effectively to advanced mathematical tasks such as automated theorem proving. This limitation arises from two fundamental challenges: evaluation difficulty, which demands machine-verifiable feedback unavailable to informal methods, and scarcity of high-quality formal proof data (Yang et al., 2024b; Asperti et al., 2025).

### 4.3.2 RL for Formal Mathematical Reasoning

Formal mathematical reasoning refers to reasoning carried out in a formal language with precisely defined syntax and semantics, yielding proof objects that are mechanically checkable by a verifier. This paradigm is particularly suited for advanced tasks such as automated theorem proving (ATP) (Xin et al., 2025), where an agent, given a statement (theorem, lemma, or proposition), must construct a proof object that the verifier accepts, thereby ensuring machine-verifiable correctness. From a reinforcement learning perspective, formal theorem proving is commonly modeled as a Markov Decision Process (MDP): proof states transition via the application of tactics[2], each of which is treated as a discrete action in RL-based proof search (Wu et al., 2021). Under this formulation, formal theorem proving can be cast as a search problem over a vast, discrete, and parameterized action space.

Formal proofs are verified by proof assistants such as Lean, Isabelle, Coq, and HOL Light. These systems, often referred to as Interactive Theorem Provers (ITPs), deterministically accept or reject proof objects, producing binary pass/fail signals as the primary reward for RL training, while some works also explore leveraging error messages as auxiliary signals (Ambati, 2025; Ji et al., 2025).

**Outcome reward RL.** The outcome-only paradigm was demonstrated at scale in 2024 with DeepSeek-Prover-v1.5 (Xin et al., 2025), which releases an end-to-end RL pipeline in Lean based solely on binary verifier feedback, resulting in significant improvements in proof success on benchmarks like miniF2F (Zheng et al., 2022) and ProofNet (Azerbayev et al., 2023). The authors propose a variant of MCTS, i.e. RMaxTS, that incorporates intrinsic rewards for discovering novel tactic states to encourage diversity of proof exploration during inference-time search and mitigate the sparse-reward issue. Building on this direction, Leanabell-

---

[2]In Lean-style Interactive Theorem Provers (ITPs), a tactic is a command or small script that instructs the system to refine the current proof goal, with the resulting proof term checked by the ITP kernel for correctness.

Prover (Zhang et al., 2025k) scales up DeepSeek-Prover-v1.5 by aggregating an expansive hybrid dataset of statement-proof pairs and informal reasoning sketches from multiple sources and pipelines such as Mathlib4 (The mathlib Community, 2020–2025), LeanWorkbook (Ying et al., 2025), NuminaMath (Li et al., 2024a), STP (Dong & Ma, 2025), etc., covering well over 20 mathematical domains. This broad coverage mitigates the scarcity of aligned informal-to-formal (NL to Lean4) training examples, which are crucial for bridging natural-language reasoning and formal proof generation. At the same time, Kimina-Prover (Wang et al., 2025a) Preview further emphasizes the critical challenge of aligning informal and formal reasoning. It implements a structured "formal reasoning pattern," where natural-language reasoning and Lean 4 code snippets are interleaved within thinking blocks. To reinforce this alignment, the output is constrained—to include at least one tactic block and to reuse no less than 60% of the Lean 4 snippets in the final proof, ensuring close correspondence between internal reasoning and formal output. A recent work, Seed-Prover (Chen et al., 2025c), integrates multiple techniques. It first adopts a lemma-centered proof paradigm, which enables systematic problem decomposition, cross-trajectory lemma reuse, and explicit progress tracking. It then enriches RL training with a diverse prompting strategy that randomly incorporates both informal and formal proofs, successful and failed lemmas, and Lean compiler feedback, thereby enhancing adaptability to varied inputs. At inference, it employs a conjecture–prover pipeline that interleaves proving conjectures into lemmas and generating new conjectures from the evolving lemma pool, substantially improving its capacity to tackle difficult problems. Complementarily, the accompanying Seed-Geometry system extends formal reasoning to geometry, providing state-of-the-art performance on Olympiad benchmarks. Together, these efforts demonstrate that sparse but explicit reward signals can yield nontrivial gains, particularly when paired with effective exploration strategies.

**Process reward RL.** To improve credit assignment and reduce wasted exploration, several works extend the outcome-only paradigm with denser, step-level signals. DeepSeek-Prover-v2 (DeepSeek-AI et al., 2024) designs a dual-model pipeline to unify both informal (natural-language) and formal (Lean4) mathematical reasoning to reinforce the formal reasoning ability. It introduces subgoal decomposition, where a prover model solves recursively decomposed subgoals and receives binary Lean feedback at the subgoal level, effectively providing denser supervision and improving both accuracy and interpretability. Following this dual-role collaborative mindset, ProofNet++ (Ambati, 2025) implements a neuro-symbolic RL framework featuring a Symbolic Reasoning Interface, which maps LLM-generated reasoning into formal proof trees, and a Formal Verification Engine, which verifies these proofs with Lean or HOL Light and routes error feedback back to the LLM for self-correction. Leanabell-Prover-v2 (Ji et al., 2025) integrates verifier messages into reinforcement updates within a long CoT framework, enabling explicit verifier-aware self-monitoring that stabilizes tactic generation and reduces repeated failure patterns.

**Hybrid reward RL.** Although both outcome-only and process-aware reward paradigms have demonstrated encouraging advances, the scarcity of high-quality theorem-proving data further amplifies the challenges of reinforcement learning under sparse rewards as well as the design of step-level preference signals (Zeng & Zhong, 2024; Wang et al., 2024f; Dong & Ma, 2025). To mitigate these limitations, a prominent line of work adopts expert iteration (ExIt) (Anthony et al., 2017), a framework that combines search with policy learning. This paradigm provides an alternative to outcome-only or process-aware RL, alleviating data scarcity by producing high-quality supervised trajectories. Instead of directly optimizing against sparse verifier signals, ExIt performs *search-guided data augmentation*: valid proof trajectories discovered by search and checked by a verifier are reused as expert demonstrations in an imitation-learning loop. It usually employs a two-role system: the *expert* collects valid and progressive trajectories via MCTS under outcome-only verifier feedback, while the *apprentice* trains a policy on these process-level trajectories and then shares the improved policy back with the expert, thereby bootstrapping subsequent rounds of search and accelerating convergence. Prior work (Polu & Sutskever, 2020) introduces ExIt into formal theorem proving, demonstrating that search-generated expert data can bootstrap models toward tackling complex multi-step proving challenges. Later works adapt this design to Lean and other ITPs.

When applied to formal theorem proving, naive tree search methods often face severe search space explosion when navigating the vast parameterized tactic space. To mitigate this, InternLM2.5-StepProver (Wu et al., 2025m) introduces a preference-based critic model, trained with RLHF-style optimization, to guide expert

search, effectively providing a curriculum that directs exploration toward problems of suitable difficulty. Lean-STaR (Lin et al., 2025a) further enhances ExIt by integrating Self-Taught Reasoner (STaR) (Zelikman et al., 2022). It first trains a thought-augmented tactic predictor on synthesized *(proof state, generated thought, ground-truth tactic)* triples. Then, in the expert-iteration loop, the model produces trajectories that interleave thoughts with tactics; trajectories with tactics successfully validated by Lean are retained and reused for imitation learning. Empirically, the inclusion of thoughts increases the diversity of exploration in the sample-based proof search.

A recent work, STP (Dong & Ma, 2025), points out that solely relying on expert iteration will quickly plateau due to the sparse positive rewards. To address this, it extends the conjecturer–prover self-play idea from Minimo (Poesia et al., 2024) to practical formal languages (Lean/Isabelle) with an open-ended action space and starts from a pretrained model. STP instantiates a dual-role loop in which a conjecturer proposes statements that are barely provable by the current prover, and a prover is trained with standard expert iteration; this generates an adaptive curriculum and alleviates sparse training signals. Empirically, STP reports large gains on LeanWorkbook (Ying et al., 2025) and reports competitive results among whole-proof generation methods on miniF2F (Zheng et al., 2022) and ProofNet (Azerbayev et al., 2023).

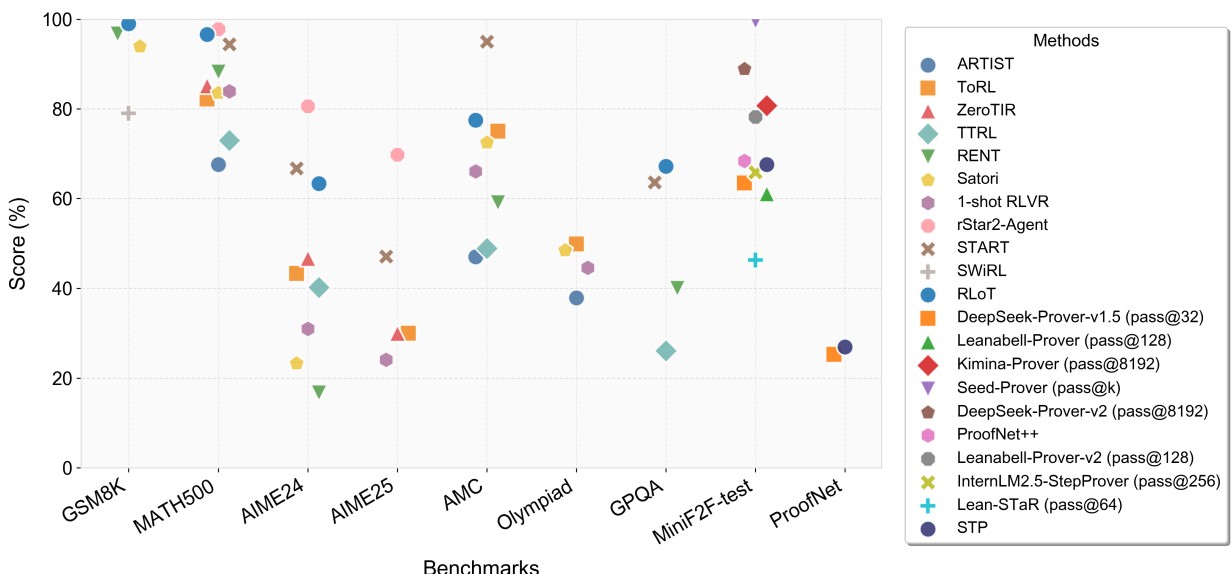

Figure 8: Benchmark Performance of RL-Enhanced Math Methods. Scores are pass@1 unless otherwise specified.

## 4.4 GUI Agent

GUI agents have progressed through distinct training paradigms. Early systems used pre-trained vision–language models (VLMs) in a pure zero-shot fashion, mapping screenshots and prompts directly to single-step actions. Later, SFT on static (screen, action) trajectories improved grounding and reasoning, but were limited by scarce human operation traces. Reinforcement fine-tuning (RFT) reframes GUI interaction as sequential decision-making, allowing agents to learn via trial-and-error with sparse or shaped rewards, and has advanced from simple single-task settings to complex, real-world, long-horizon scenarios. Table 7 presents the majority of works studied in this section.

### 4.4.1 RL-free Methods

**Vanilla VLM-based GUI Agents** Early GUI agents directly leveraged pre-trained Vision–Language Models (VLMs) in a purely zero-shot manner, mapping screenshots and prompts to single-step actions without

Table 6: A summary of RL methods for mathematical reasoning agents.

| Method | Reward | Resources |
|---|---|---|
| *RL for Informal Mathematical Reasoning* | | |
| ARTIST (Singh et al., 2025) | Outcome | - |
| ToRL (Li et al., 2025l) | Outcome | ⭕ GitHub 🤗 HuggingFace |
| ZeroTIR (Mai et al., 2025) | Outcome | ⭕ GitHub 🤗 HuggingFace |
| TTRL (Zuo et al., 2025) | Outcome | ⭕ GitHub |
| RENT (Prabhudesai et al., 2025) | Outcome | ⭕ GitHub 🌐 Website |
| Satori (Shen et al., 2025b) | Outcome | ⭕ GitHub 🤗 HuggingFace 🌐 Website |
| 1-shot RLVR (Wang et al., 2025r) | Outcome | ⭕ GitHub 🤗 HuggingFace |
| Prover-Verifier Games (Kirchner et al., 2024) | Outcome | - |
| rStar2-Agent (Shang et al., 2025a) | Outcome | ⭕ GitHub |
| START (Li et al., 2025b) | Process | - |
| LADDER (Simonds & Yoshiyama, 2025) | Process | - |
| SWiRL (Goldie et al., 2025) | Process | - |
| RLoT (Hao et al., 2025b) | Process | ⭕ GitHub |
| *RL for Formal Mathematical Reasoning* | | |
| DeepSeek-Prover-v1.5 (Xin et al., 2025) | Outcome | ⭕ GitHub 🤗 HuggingFace |
| Leanabell-Prover (Zhang et al., 2025k) | Outcome | ⭕ GitHub 🤗 HuggingFace |
| Kimina-Prover (Wang et al., 2025a) | Outcome | ⭕ GitHub 🤗 HuggingFace |
| Seed-Prover (Chen et al., 2025c) | Outcome | ⭕ GitHub |
| DeepSeek-Prover-v2 (DeepSeek-AI et al., 2024) | Process | ⭕ GitHub 🤗 HuggingFace |
| ProofNet++ (Ambati, 2025) | Process | - |
| Leanabell-Prover-v2 (Ji et al., 2025) | Process | ⭕ GitHub |
| InternLM2.5-StepProver (Wu et al., 2025m) | Hybrid | ⭕ GitHub |
| Lean-STaR (Lin et al., 2025a) | Hybrid | ⭕ GitHub 🤗 HuggingFace 🌐 Website |
| STP (Dong & Ma, 2025) | Hybrid | ⭕ GitHub 🤗 HuggingFace |

any task-specific fine-tuning. Representative systems include MM-Navigator (Yan et al., 2023), SeeAct (Zheng et al., 2024), and TRISHUL (Kunal Singh, 2025), which differ in interface domains or parsing strategies but share the same reliance on off-the-shelf VLMs. While showcasing the generality of foundation models, these approaches suffer from limited grounding accuracy and reliability, restricting their applicability to complex tasks (Zhang et al., 2025b; Nguyen et al., 2025a).

**Supervised Fine-Tuning (SFT) with Static Trajectory Data** The SFT paradigm adapts pre-trained vision–language models to GUI tasks by minimizing cross-entropy loss on offline (screen, action) pairs, without online interaction. InfiGUIAgent (Liu et al., 2025n) employs a two-stage pipeline that first improves grounding and then incorporates hierarchical and reflective reasoning. UI-AGILE (Lian et al., 2025) enhances supervised fine-tuning by incorporating continuous rewards, simplified reasoning, and cropping-based resampling, while further proposing a decomposed grounding mechanism for handling high-resolution displays. TongUI (Zhang et al., 2025a) instead emphasizes data scale, constructing the 143K-trajectory GUI-Net from multimodal web tutorials to enhance generalization. While differing in focus, these approaches all face the limitation of scarce human operation traces.

### 4.4.2 RL in Static GUI Environments

In static settings, reinforcement learning is applied on pre-collected datasets with deterministic execution traces, using rule-based criteria for outcome evaluation in the absence of live environment interactions. GUI-R1 (Luo et al., 2025d) adopts an R1-style reinforcement fine-tuning pipeline over a unified action schema, using simple format and correctness rewards to improve step-level action prediction with modest

data. UI-R1 (Lu et al., 2025d) applies group-relative policy optimization to stabilize policy updates and improve exact parameter matching through a compact action interface and reward shaping for action-type and argument accuracy. InFiGUI-R1 (Liu et al., 2025o) introduces a two-stage training paradigm that first distills spatial reasoning to enhance grounding, followed by reinforcement learning with sub-goal supervision and recovery mechanisms to improve long-horizon reasoning. AgentCPM-GUI (Zhang et al., 2025y) combines grounding-aware pre-training, supervised imitation, and GRPO-based reinforcement fine-tuning with a concise JSON action space, reducing decoding overhead while improving robustness on long-horizon sequences. UI-Venus (Gu et al., 2025) is a multimodal screenshot-based UI agent fine-tuned via RFT with custom reward functions and a self-evolving trajectory framework, achieving a new state-of-the-art performance in both UI grounding and navigation.

### 4.4.3 RL in Interactive GUI Environments

In interactive settings, reinforcement learning agents are optimized through online rollouts in dynamic environments, requiring robustness to stochastic transitions and long-horizon dependencies. WebAgent-R1 (Wei et al., 2025f) conducts end-to-end multi-turn reinforcement learning with asynchronous trajectory generation and group-wise advantages, improving success on diverse web tasks. Vattikonda et al. (2025) studies reinforcement learning for web agents under realistic page dynamics and large action spaces, highlighting challenges in credit assignment and safe exploration. UI-TARS (Qin et al., 2025) integrates pre-training for GUI understanding with reinforcement learning for native desktop control, coupling milestone tracking and reflection to enhance long-horizon execution. DiGiRL (Bai et al., 2024) introduces an offline-to-online reinforcement learning pipeline on real Android devices, combining advantage-weighted updates, doubly robust advantage estimation, and instruction-level curricula to cope with non-stationarity. ZeroGUI (Yang et al., 2025b) automates task generation and reward estimation with a vision-language evaluator, then applies two-stage online reinforcement learning (training on generated tasks followed by test-time adaptation) to reduce human supervision. MobileGUI-RL (Shi et al., 2025c) scales training on Android virtual devices with trajectory-aware GRPO, a decaying efficiency reward, and curriculum filtering, improving execution efficiency and generalization while keeping the system practical for large rollout volumes. ComputerRL (Lai et al., 2025) introduces an API-GUI hybrid interaction paradigm paired with a massively parallel, fully asynchronous RL infrastructure and the novel Entropulse training strategy—alternating RL with supervised fine-tuning—to empower GUI-based agents to operate efficiently and scalably in desktop environments.

## 4.5 Vision Agents

RL has been applied to a wide range of vision tasks (including, but not limited to, image, video, 3D perception and generation). Since the number of related papers is substantial, this section does not aim to provide an exhaustive overview; for a more comprehensive survey on RL for various vision tasks, we refer readers to two dedicated surveys in vision (Wu et al., 2025h; Zhou et al., 2025a).

**Image Tasks.** The success of DeepSeek-R1 (DeepSeek-AI et al., 2025) has sparked widespread interest in applying RL to incentivize long-form reasoning behavior, encouraging LVLMs to produce extended CoT sequences that improve visual perception and understanding (Shao et al., 2024a). This research trajectory has evolved from early work that simply adapted R1-style objectives to the vision domain—aimed primarily at enhancing passive perception (Tan et al., 2025a; Li et al., 2025t; Huang et al., 2025c; Shen et al., 2025a; Peng et al., 2025; Xia et al., 2025a; Yang et al., 2025c; Gao et al., 2025a)—toward the now-popular paradigm of active perception, or "thinking with images" (Su et al., 2025c). The key transition lies in moving from text-only CoT that references an image once, to interactive, visually grounded reasoning, achieved through (i) grounding (Li et al., 2025e; Nagaraja et al., 2016; Mao et al., 2016; Fan et al., 2025c; Chung et al., 2025; Cao et al., 2025a), (ii) agentic tool use (Zhao et al., 2025d; Huang et al., 2025d; Wu et al., 2025g; Su et al., 2025b; Liu et al., 2025v; Su et al., 2025a), and (iii) visual imagination via sketching or generation (Xu et al., 2025e; Duan et al., 2025; Jiang et al., 2025b). Beyond text-only outputs, many vision tasks—such as scene understanding—require structured predictions like bounding boxes, masks, and segmentation maps. To begin with, Visual-RFT (Liu et al., 2025v) uses IoU with confidence as a verifiable reward for bounding-box outputs, while Vision-R1 (Huang et al., 2025c) incorporates precision and recall as localization rewards.

Table 7: A summary of methods for GUI agents, categorized by training paradigm and environment complexity.

| Method | Paradigm | Environment | Resource Link |
|---|---|---|---|
| *RL-free GUI Agents* | | | |
| MM-Navigator (Yan et al., 2023) | Vanilla VLM | - | ⚡GitHub |
| SeeAct (Zheng et al., 2024) | Vanilla VLM | - | ⚡GitHub |
| TRISHUL (Kunal Singh, 2025) | Vanilla VLM | - | - |
| InfiGUIAgent (Liu et al., 2025n) | SFT | Static | ⚡GitHub 🤗HuggingFace 🌐Website |
| UI-AGILE (Lian et al., 2025) | SFT | Interactive | ⚡GitHub 🤗HuggingFace |
| TongUI (Zhang et al., 2025a) | SFT | Static | ⚡GitHub 🤗HuggingFace 🌐Website |
| *RL-based GUI Agents* | | | |
| GUI-R1 (Luo et al., 2025d) | RL | Static | ⚡GitHub 🤗HuggingFace |
| UI-R1 (Lu et al., 2025d) | RL | Static | ⚡GitHub 🤗HuggingFace |
| InFiGUI-R1 (Liu et al., 2025o) | RL | Static | ⚡GitHub 🤗HuggingFace |
| AgentCPM (Zhang et al., 2025y) | RL | Static | ⚡GitHub 🤗HuggingFace |
| UI-Venus (Gu et al., 2025) | RL | Static | ⚡GitHub |
| WebAgent-R1 (Wei et al., 2025f) | RL | Interactive | - |
| Vattikonda et al. (2025) | RL | Interactive | - |
| UI-TARS (Qin et al., 2025) | RL | Interactive | ⚡GitHub 🤗HuggingFace 🌐Website |
| UI-TARS-2 (Wang et al., 2025c) | RL | Interactive | ⚡GitHub 🌐Website |
| DiGiRL (Bai et al., 2024) | RL | Interactive | ⚡GitHub 🤗HuggingFace 🌐Website |
| ZeroGUI (Yang et al., 2025b) | RL | Interactive | ⚡GitHub |
| MobileGUI-RL (Shi et al., 2025c) | RL | Interactive | - |
| ComputerRL (Lai et al., 2025) | RL | Interactive | - |

Extending this idea, Liu et al. (2025p) applies GRPO to segmentation tasks, combining soft and strict rewards with bounding-box IoU and L1 loss, and point-wise L1 distance. VLM-R1 (Shen et al., 2025a) employs mean Average Precision (mAP) as a reward to explicitly incentivize detection and localization capabilities in LVLMs. Finally, R1-SGG (Chen et al., 2025m) introduces three variants of GRPO rewards for scene-graph matching—ranging from hard rewards based on text matching and IoU to softer rewards computed via text-embedding dot products. RL has also been widely applied to image generation, particularly through its integration with diffusion and flow models—for example, RePrompt (Wu et al., 2025f), Diffusion-KTO (Li et al., 2024c), Flow-GRPO (Liu et al., 2025d), and GoT-R1 (Duan et al., 2025). Beyond diffusion-based approaches, RL has been leveraged for autoregressive image generation, where it improves coherence, fidelity, and controllability by directly optimizing task- or user-specific reward signals (Wang et al., 2025f; Jiang et al., 2025b; Yuan et al., 2025a).

**Video Tasks.** Following the same spirit, numerous works have extended GRPO variants to the video domain (Cheng et al., 2024; Feng et al., 2024b; Maaz et al., 2023) to enhance temporal reasoning (Park et al., 2025b; Li et al., 2025k; Zhu et al., 2025c; Liao et al., 2025c; Ouyang, 2025). TW-GRPO (Dang et al., 2025) introduces a token-weighted GRPO framework that emphasizes high-information tokens to generate more focused reasoning chains and employs soft, multi-choice rewards for lower-variance optimization. EgoVLM (Vinod et al., 2025) combines keyframe-based rewards with direct GRPO training to produce interpretable reasoning traces tailored for egocentric video. DeepVideo-R1 reformulates the GRPO objective as a regression task (Park et al., 2025b), while VideoChat-R1 demonstrates that reinforcement fine-tuning (RFT) can be highly data-efficient for task-specific video reasoning improvements (Li et al., 2025k). TinyLLaVA-Video-R1 explores scaling RL to smaller video LLMs (Zhang et al., 2025t), and (Chen et al., 2025j) introduces infrastructure and a two-stage pipeline (CoT-SFT + RL) to support large-scale RL for long videos. Additional efforts have also extended RL for embodied video reasoning tasks (Zhao et al., 2025b). A similar trend is observed in video generation, where RL is applied to improve temporal coherence, controllability, and semantic alignment. Key examples include DanceGRPO (Xue et al., 2025), GAPO (Zhu et al., 2025a),

GRADEO (Mou et al., 2025), InfLVG (Fang et al., 2025b), Phys-AR (Lin et al., 2025c), VideoReward (Liu et al., 2025e), TeViR (Chen et al., 2025i), and InstructVideo (Yuan et al., 2024).

**3D Vision Tasks.** RL has also been widely adopted to advance 3D understanding (Hong et al., 2023; Xu et al., 2024b; Deng et al., 2024a; Chen et al., 2024a; Zhou et al., 2023; Chen et al., 2024d) and generation (Wang et al., 2024i; Yin et al., 2025; Siddiqui et al., 2024). MetaSpatial (Pan & Liu, 2025) introduces the first RL-based framework for 3D spatial reasoning, leveraging physics-aware constraints and rendered-image evaluations as rewards during training. Scene-R1 (Yuan et al., 2025d) learns to reason about 3D scenes without point-wise 3D supervision, while SpatialReasoner (Ma et al., 2025c) introduces shared 3D representations that unify perception, computation, and reasoning stages. In the domain of 3D generation, RL has been applied to improve text-to-3D alignment and controllability. Notable efforts include DreamCS (Zou et al., 2025), which aligns generation with human preferences; DreamDPO (Zhou et al., 2025f) and DreamReward (Ye et al., 2024), which optimize 3D generation using 2D reward signals; and Nabla-R2D3 (Liu et al., 2025i), which further refines 3D outputs with reinforcement-driven objectives.

## 4.6 Embodied Agents

Embodied agents encompass a broad family of systems that perceive a structured environment and act within it, ranging from vision-language-action (VLA) models to language-driven open-ended agents. While many recent systems focus on VLA settings that require grounding in real-world visual observations, all embodied agents must integrate perception, reasoning, and action to operate effectively in complex physical or simulated environments and to execute goal-directed behaviors conditioned on high-level instructions. These competencies form a foundational component of agentic LLMs and MLLMs in embodied scenarios. In instruction-driven embodied scenarios, RL is often employed as a post-training strategy. A common pipeline begins with a pre-trained vision-language-action (VLA) model (Kim et al., 2024; Black et al., 2024; Team et al., 2025a; Liao et al., 2025b) obtained through imitation learning under teacher forcing supervision. This model is then embedded into an interactive agent that engages with the environment to collect reward signals. These rewards guide the iterative refinement of the policy, supporting effective exploration, improving sample efficiency, and enhancing the model's generalization capabilities across diverse real-world conditions. RL in VLA frameworks (SimpleVLA-RL Team, 2025; Lu et al., 2025a; Qi et al., 2025; Song et al., 2025e) can be broadly categorized into two classes: navigation agents, which emphasize spatial reasoning and locomotion in complex environments, and manipulation agents, which focus on the precise control of physical objects under diverse and dynamic constraints.

**RL in VLA Navigation Agent.** For navigation agents, planning is the central capability. Reinforcement learning is employed to enhance the VLA model's ability to predict and optimize future action sequences. A common strategy (Zhao et al., 2025c) is to integrate traditional robotics-style RL, using step-wise directional rewards, directly into VLA-based navigation frameworks. Some approaches operate at the trajectory level. VLN-R1 (Qi et al., 2025) aligns predicted and ground-truth paths to define trajectory-level rewards, and applies GRPO, following DeepSeek-R1, to improve predictive planning. OctoNav-R1 (Gao et al., 2025a) also leverages GRPO but focuses on reinforcing internal deliberation within the VLA model, promoting a thinking-before-acting paradigm that enables more anticipatory and robust navigation. S2E (He et al., 2025) introduces a reinforcement learning framework that augments navigation foundation models with interactivity and safety, combining video pretraining with RL to achieve superior generalization and performance on the NavBench-GS benchmark.

**RL in VLA Manipulation Agent.** Manipulation agents, typically involving robotic arms, require fine-grained control for executing structured tasks under diverse conditions. In this context, RL is employed to enhance the instruction-following and trajectory prediction capabilities of VLA models, especially to improve generalization across tasks and environments. RLVLA (Liu et al., 2025f) and VLA-RL (Lu et al., 2025a) adopt pre-trained VLMs as evaluators, using their feedback to assign trajectory-level rewards for VLA policy refinement. These methods establish an online RL framework that effectively improves manipulation performance and demonstrates favorable scaling properties. TGRPO (Chen et al., 2025k) further incorporates GRPO into manipulation tasks by defining rule-based reward functions over predicted trajectories. This

enables the VLA model to generalize to unseen scenarios and improves its robustness in real-world deployment. VIKI-R (Kang et al., 2025a) complements this with a unified benchmark and two-stage framework for multi-agent embodied cooperation, combining Chain-of-Thought fine-tuning with multi-level RL to enable compositional coordination across diverse embodiments.

A central challenge in RL for VLA embodied agents is scaling training to real-world environments. While simulation platforms enable efficient large-scale experimentation, the sim-to-real gap remains significant, particularly in fine-grained manipulation tasks. Conducting RL directly in real-world settings is currently impractical due to the high cost and complexity of physical robot experiments. Most RL algorithms require millions of interaction steps, which demand substantial time, resources, and maintenance. As a result, developing scalable embodied RL pipelines that can bridge the gap between simulation and real-world deployment remains an open and pressing problem.

**Case Study: Voyager.** Beyond these general challenges in embodied RL, Voyager (Wang et al., 2024a), a language-driven open-ended embodied agent, illustrates how planning, skill acquisition, and RL-based curriculum learning can be integrated in practice. The agent explores Minecraft using an iterative loop: it generates a plan, interacts with the environment, extracts reusable skills from successful trajectories, and stores them in a growing skill library. A curriculum scheduler selects new tasks based on the agent's current skill set, while RL objectives guide which behavior should be committed as skills and when to refine or discard them. This creates a self-improving cycle in which planning, environmental interaction, memory, and RL-driven curriculum optimization are tightly coupled.

## 4.7 Multi-Agent Systems

Large Language Model (LLM)-based Multi-agent Systems (MAS) comprise multiple autonomous agents collaborating to solve complex tasks through structured interaction, coordination, and memory management. Early static and hand-designed MAS such as CAMEL and MetaGPT (Li et al., 2023a; Hong et al., 2024b) explored role specialization and task decomposition, while debate-based frameworks such as MAD and MoA (Wang et al., 2025g; Liang et al., 2024) enhanced reasoning via collaborative refinement. Subsequent multi-agent research has shifted to proposing optimizable cooperative systems, which enable MAS to not only dynamically adjust coordination patterns but also directly enhance agent-level reasoning and decision-making strategies. Table 8 summarizes the main body of works discussed in this section.

**RL-Free Multi-Agent Evolution** In the RL-free self-evolving setting, foundation models cannot be directly optimized; instead, system evolution is driven by mechanisms such as symbolic learning (Zhou et al., 2024c), dynamic graph optimization (Zhuge et al., 2024; Ma et al., 2025d; Zhou et al., 2025b), and workflow rewriting (Hu et al., 2025d; Zhang et al., 2025i;h). These methods improve the coordination and adaptability within MAS, but cannot directly update the parameters of foundation models.

### 4.7.1 RL-Driven Optimization of Non-Parametric Coordination Modules

These approaches keep agent parameters frozen while using RL to optimize external coordination structures such as communication topologies, routing policies, or workflow graphs. Methods such as GPTSwarm, MaAS, and G-Designer (Zhuge et al., 2024; Zhang et al., 2025f;g) treat MAS coordination as a graph-level policy updated via policy gradient. Because no agent-level gradients exist, credit assignment must operate at the topology or message-routing level. Rewards are typically delayed and sparse—e.g., only final task accuracy—requiring global-to-local credit decomposition or structural priors to avoid collapse.

A key comparison emerges between *fixed communication protocols* (pre-specified message formats) and *learnable protocols*. Fixed protocols excel in low-data or highly specialized domains where stability is critical, whereas learnable protocols allow RL to discover efficient emergent communication but require substantially higher sample complexity and careful regularization to prevent overfitting or degenerate conventions.

Table 8: A summary of reinforcement learning and evolution paradigms in LLM-based Multi-Agent Systems. "Dynamic" denotes whether the multi-agent system is task-dynamic, *i.e.*, processes different task queries with different configurations (agent count, topologies, reasoning depth, prompts, *etc*). "Train" denotes whether the method involves training the LLM backbone of agents.

| Method | Dynamic | Train | RL Algorithm | Resource Link |
|---|---|---|---|---|
| *RL-Free Multi-Agent Systems* (not exhaustive) | | | | |
| CAMEL (Li et al., 2023a) | ✗ | ✗ | - |  GitHub  HuggingFace |
| MetaGPT (Hong et al., 2024b) | ✗ | ✗ | - |  GitHub |
| MAD (Liang et al., 2024) | ✗ | ✗ | - |  GitHub |
| MoA (Wang et al., 2025g) | ✗ | ✗ | - |  GitHub |
| AFlow (Zhang et al., 2025i) | ✗ | ✗ | - |  GitHub |
| *RL-Based Multi-Agent Training* | | | | |
| GPTSwarm (Zhuge et al., 2024) | ✗ | ✗ | policy gradient |  GitHub  Website |
| MaAS (Zhang et al., 2025f) | ✔ | ✗ | policy gradient |  GitHub |
| G-Designer (Zhang et al., 2025g) | ✔ | ✗ | policy gradient |  GitHub |
| Optima (Chen et al., 2025e) | ✗ | ✔ | DPO |  GitHub |
| DITS (Shi et al., 2025a) | ✗ | ✔ | DPO | - |
| MALT (Motwani et al., 2025) | ✗ | ✔ | DPO | - |
| MARFT (Liao et al., 2025a) | ✗ | ✔ | MARFT |  GitHub |
| ACC-Collab (Estornell et al., 2025b) | ✗ | ✔ | DPO | - |
| MAPoRL (Park et al., 2025a) | ✔ | ✔ | PPO |  GitHub |
| MLPO (Estornell et al., 2025a) | ✔ | ✔ | MLPO | - |
| ReMA (Wan et al., 2025b) | ✔ | ✔ | MAMRP |  GitHub |
| FlowReasoner (Gao et al., 2025b) | ✔ | ✔ | GRPO |  GitHub |
| CURE (Wang et al., 2025q) | ✗ | ✔ | rule-based RL |  GitHub  HuggingFace |
| MMedAgent-RL (Xia et al., 2025b) | ✗ | ✔ | GRPO | - |
| Chain-of-Agents (Li et al., 2025h) | ✔ | ✔ | DAPO |  GitHub  HuggingFace |
| RLCCF (Yuan et al., 2025b) | ✗ | ✔ | GRPO | - |
| MAGRPO (Liu et al., 2025l) | ✗ | ✔ | MAGRPO | - |

### 4.7.2 RL-Driven Optimization of Selected Agent Policies

A second class of systems updates only a subset of agents—often a leader, coordinator, or specialized expert—while keeping others frozen for stability. Representative examples include Optima, DITS, MALT, ACC-Collab (Chen et al., 2025e; Shi et al., 2025a; Motwani et al., 2025; Estornell et al., 2025b). These approaches balance flexibility and scalability: training only a few agents reduces sample complexity and avoids the instability of fully-decoupled credit assignment. MALT (Motwani et al., 2025) employs a heterogeneous multi-agent search tree to generate large-scale labeled trajectories, fine-tuning agents via a combination of Supervised Fine-Tuning (SFT) and Direct Preference Optimization (DPO) from both successful and failed reasoning paths.

Credit assignment in this regime is fundamentally *semi-local*: rewards emerge from a collective trajectory, but gradients apply only to the optimized agent(s). This requires mechanisms such as role-conditioned DPO (Motwani et al., 2025), local advantage estimation, or counterfactual baselines to prevent reward hijacking by non-updated agents. Empirically, such partial optimization yields better sample efficiency than fully joint multi-agent training while still enabling the emergence of specialized roles.

### 4.7.3 End-to-End Multi-Agent Reinforcement Learning

Full multi-agent RL jointly trains all agents under a shared or decentralized objective, typically formalized as a Dec-POMDP. Methods such as MAGRPO, MAPoRL, MLPO, ReMA, FlowReasoner, Chain-of-Agents, and SPIRAL (Liu et al., 2025l; Park et al., 2025a; Estornell et al., 2025a; Wan et al., 2025b; Gao et al., 2025b; Li et al., 2025h; Liu et al., 2025b) jointly optimize collaboration and reasoning behaviors, enabling emergent

division of labor and communication conventions. For example, MAGRPO (Liu et al., 2025l) formalizes multi-LLM cooperation as a Dec-POMDP problem and introduces a multi-agent variant of GRPO, which enables joint training of LLM agents in MAS while maintaining decentralized execution. MAPoRL (Park et al., 2025a) extends MAD by verifying debate responses and using validation outcomes as RL rewards to improve collaborative reasoning. RLCCF (Yuan et al., 2025b) is a self-supervised multi-agent RL framework that leverages self-consistency-weighted ensemble voting to generate pseudo-labels and collaboratively optimize individual model policies via GRPO, boosting both individual and collective reasoning accuracy. ReMA (Wan et al., 2025b) separates reasoning into a meta-thinking agent and an execution agent, jointly trained under aligned RL objectives with parameter sharing. LERO (Wei et al., 2025d) combines MARL with LLM-generated hybrid rewards and evolutionary search to improve credit assignment and partial observability handling in cooperative tasks. CURE (Wang et al., 2025q) focuses on code generation, jointly training a code generator and unit tester via RL to produce richer reward signals, achieving strong generalization across diverse coding benchmarks. MMedAgent-RL (Xia et al., 2025b) introduces a reinforcement learning-based multi-agent framework for medical VQA, where dynamically coordinated general practitioners and specialists collaboratively reason with curriculum-guided learning, significantly outperforming existing Med-LVLMs and achieving more human-like diagnostic behavior. Chain-of-Agents (COA) (Li et al., 2025h) is an end-to-end paradigm where a single LLM simulates multi-agent collaboration by dynamically orchestrating role-playing and tool-using agents; this is achieved through multi-agent distillation (converting trajectories from state-of-the-art multi-agent systems into training data) and agentic reinforcement learning with carefully designed reward functions, resulting in Agent Foundation Models (AFMs). SPIRAL (Liu et al., 2025b) presents a fully online, multi-turn, multi-agent self-play reinforcement learning framework for LLMs in zero-sum games, employing a shared policy with role-conditioned advantage estimation (RAE) to stabilize learning, and demonstrates that gameplay fosters transferable reasoning skills that significantly improve mathematical and general reasoning benchmarks.

However, end-to-end multi-LLM training exacerbates the *temporal and structural credit assignment* problem because rewards may depend on long multi-turn interaction chains. Solutions include role-conditioned advantage estimation (RAE), hierarchical controller–worker architectures (MLPO, ReMA), and self-play curricula (SPIRAL) that densify reward signals by constructing increasingly challenging interactions. These hierarchical patterns mirror enterprise deployments where a supervisory agent coordinates multiple workers; RL proves particularly effective at learning stable delegation and arbitration strategies under sparse reward settings. Despite their expressiveness, joint MARL approaches face scalability limits: sample complexity grows roughly linearly with the number of agents and quadratically with interaction depth. Algorithms such as MAGRPO and PPO-based MAPoRL mitigate this using centralized critics or value-shared baselines, but achieving scalable credit decomposition remains a central open challenge.

## 4.8 Other Tasks

**TextGame.** ARIA (Yang et al., 2025e) compresses the sprawling action space via intention-driven reward aggregation, reducing sparsity and variance. GiGPO (Feng et al., 2025b) enhances temporal credit assignment through hierarchical grouping without added computational burden. RAGEN (Wang et al., 2025v) ensures stable multi-turn learning by filtering trajectories and stabilizing gradients, while advocating for reasoning-aware rewards. SPA-RL (Wang et al., 2025b) decomposes delayed rewards into per-step signals, improving performance and grounding accuracy. Trinity-RFT (Pan et al., 2025) provides a unified, modular framework for reinforcement fine-tuning across tasks—including text games—enabling flexible, efficient, and scalable experimentation with diverse RL modes and data pipelines.

**Table.** SkyRL-SQL (Liu et al., 2025j) introduces a data-efficient, multi-turn RL pipeline for Text-to-SQL, enabling LLM agents to interactively probe databases, refine, and verify SQL queries. With just 653 training examples, the SkyRL-SQL-7B model surpasses both GPT-4o and o4-mini on SQL generation benchmarks. MSRL (Chen et al., 2025a) introduces multimodal structured reinforcement learning with multi-granularity rewards to overcome the SFT plateau in chart-to-code generation, achieving state-of-the-art performance on chart understanding benchmarks.

| Application | Agentic Capability | | | | | |
|---|---|---|---|---|---|---|
| | Planning | Tool-Use | Memory | Self-Imp. | Reasoning | Percep. |
| Search | ● | ● | ○ | ○ | ● | – |
| Code | ○ | ● | ○ | ○ | ● | – |
| Math | ● | ○ | – | ○ | ● | – |
| GUI | ● | ● | ○ | – | ○ | ● |
| Vision | ○ | ○ | – | – | ● | ● |
| Embodied | ● | ○ | ● | – | ○ | ● |
| MAS | ● | ○ | ○ | – | ● | ○ |

Table 9: Application-capability dependency matrix. Dots indicate qualitative dependency levels: ● Core, ○ Supporting, – Minimal. The heatmap provides a navigation aid linking the capability taxonomy (Section 3) with the application domains (Section 4).

**Time Series.** Time-R1 (Liu et al., 2025t) enhances moderate-sized LLMs with comprehensive temporal reasoning abilities through a progressive reinforcement learning curriculum and a dynamic rule-based reward system. TimeMaster (Zhang et al., 2025m) trains time-series MLLMs that combine SFT with GRPO to enable structured, interpretable temporal reasoning over visualized time-series inputs.

**General QA.** Agent models (Zhang et al., 2025w) internalize chain-of-action generation to enable autonomous and efficient decision-making through a combination of supervised fine-tuning and reinforcement learning. L-Zero (Zhang et al., 2025l) enables large language models to become general-purpose agents through a scalable, end-to-end reinforcement learning pipeline utilizing a low-cost, extensible, and sandboxed concurrent agent worker pool.

**Social.** Sotopia-RL (Yu et al., 2025c) refines coarse episode-level rewards into utterance-level, multi-dimensional signals to enable efficient and stable RL training for socially intelligent LLMs under partial observability and multi-faceted objectives. Wang et al. (2025h) introduces an Adaptive Mode Learning (AML) framework with the Adaptive Mode Policy Optimization (AMPO) algorithm, which uses reinforcement learning to dynamically switch between multi-granular reasoning modes in social intelligence tasks, achieving higher accuracy and shorter reasoning chains than fixed-depth RL methods like GRPO.

## 5 Enviroment and Frameworks

### 5.1 Environment Simulator

In agentic reinforcement learning, the environment is the world with which the agent interacts, receiving sensory input (observations) and enacting choices (actions) through its actuators. The environment, in turn, responds to the agent's actions by transitioning to a new state and providing a reward signal. With the rise of the LLM Agent paradigm, many works have proposed environments for training specific tasks. Table 10 provides an overview of the key environments examined in this section.

### 5.1.1 Web Environments

In the realm of web-based environments, several benchmarks offer controlled yet realistic static environments for Agentic RL. WebShop (Yao et al., 2022) is a simulated e-commerce website featuring a large catalog of real-world products and crowdsourced text instructions. Agents navigate various webpage types and issue diverse actions (e.g., searching, selecting items, customizing, purchasing) to find and buy products, with its deterministic search engine aiding reproducibility. Furthermore, Mind2Web (Gou et al., 2025) is a dataset designed for generalist web agents, featuring a substantial number of tasks from many real-world websites across diverse domains. It provides webpage snapshots and crowdsourced action sequences for tasks like finding flights or interacting with social profiles, emphasizing generalization across unseen websites and domains. Similarly, WebArena (Zhou et al., 2024b) and its multimodal extension, VisualwebArena (Koh et al.,

Table 10: A summary of environments and benchmarks for agentic reinforcement learning, categorized by agent capability, task domain, and modality. The agent capabilities are denoted by: ① Reasoning, ② Planning, ③ Tool Use, ④ Memory, ⑤ Collaboration, ⑥ Self-Improve.

| Environment / Benchmark | Agent Capability | Task Domain | Modality | Resource Link |
|---|---|---|---|---|
| LMRL-Gym (Abdulhai et al., 2025) | ①, ④ | Interaction | Text | ⚙GitHub |
| ALFWorld (Shridhar et al., 2021) | ②, ① | Embodied, Text Games | Text | ⚙GitHub 🌐Website |
| TextWorld (Côté et al., 2019) | ②, ① | Text Games | Text | ⚙GitHub |
| ScienceWorld (Wang et al., 2022) | ①, ② | Embodied, Science | Text | ⚙GitHub 🌐Website |
| AgentGym (Xi et al., 2025) | ①, ④ | Text Games | Text | ⚙GitHub 🌐Website |
| Agentbench (Liu et al., 2024b) | ① | General | Text, Visual | ⚙GitHub |
| InternBootcamp (Li et al., 2025g) | ① | General, Coding, Logic | Text | ⚙GitHub |
| LoCoMo (Maharana et al., 2024) | ④ | Interaction | Text | ⚙GitHub 🌐Website |
| MemoryAgentBench (Hu et al., 2025e) | ④ | Interaction | Text | ⚙GitHub |
| WebShop (Yao et al., 2022) | ②, ③ | Web | Text | ⚙GitHub 🌐Website |
| Mind2Web (Gou et al., 2025) | ②, ③ | Web | Text, Visual | ⚙GitHub 🌐Website |
| WebArena (Zhou et al., 2024b) | ②, ③ | Web | Text | ⚙GitHub 🌐Website |
| VisualwebArena (Koh et al., 2024) | ①, ②, ③ | Web | Text, Visual | ⚙GitHub 🌐Website |
| AppBench (Wang et al., 2024b) | ②, ③ | App | Text | ⚙GitHub |
| AppWorld (Trivedi et al., 2024) | ②, ③ | App | Text | ⚙GitHub 🌐Website |
| AndroidWorld (Rawles et al., 2025) | ②, ③ | GUI, App | Text, Visual | ⚙GitHub |
| OSWorld (Xie et al., 2024) | ②, ③ | GUI, OS | Text, Visual | ⚙GitHub 🌐Website |
| WindowsAgentArena (Bonatti et al., 2024) | ② | GUI, OS | Text, Visual | ⚙GitHub 🌐Website |
| Debug-Gym (Yuan et al., 2025c) | ①, ③ | SWE | Text | ⚙GitHub 🌐Website |
| MLE-Dojo (Qiang et al., 2025) | ②, ① | MLE | Text | ⚙GitHub 🌐Website |
| τ-bench (Barres et al., 2025) | ①, ③ | SWE | Text | ⚙GitHub |
| TheAgentCompany (Xu et al., 2024a) | ②, ③, ⑤ | SWE | Text | ⚙GitHub 🌐Website |
| MedAgentGym (Xu et al., 2025c) | ① | Science | Text | ⚙GitHub |
| SecRepoBench (Dilgren et al., 2025) | ①, ③ | Coding, Security | Text | - |
| R2E-Gym (Jain et al., 2025c) | ①, ② | SWE | Text | ⚙GitHub 🌐Website |
| BigCodeBench (Zhuo et al., 2025) | ① | Coding | Text | ⚙GitHub 🌐Website |
| LiveCodeBench (Jain et al., 2025b) | ① | Coding | Text | ⚙GitHub 🌐Website |
| SWE-bench (Jimenez et al., 2024) | ①, ③ | SWE | Text | ⚙GitHub 🌐Website |
| SWE-rebench (Badertdinov et al., 2025) | ①, ③ | SWE | Text | 🌐Website |
| DevBench (Li et al., 2025a) | ②, ① | SWE | Text | ⚙GitHub |
| ProjectEval (Liu et al., 2025g) | ②, ① | SWE | Text | ⚙GitHub 🌐Website |
| DA-Code (Huang et al., 2024b) | ①, ③ | Data Science, SWE | Text | ⚙GitHub 🌐Website |
| ColBench (Zhou et al., 2025e) | ②, ① | SWE, Web Dev | Text | ⚙GitHub 🌐Website |
| NoCode-bench (Deng et al., 2025a) | ②, ① | SWE | Text | ⚙GitHub 🌐Website |
| MLE-Bench (Chan et al., 2025) | ②, ①, ③ | MLE | Text | ⚙GitHub 🌐Website |
| PaperBench (Starace et al., 2025) | ②, ①, ③ | MLE | Text | ⚙GitHub 🌐Website |
| Crafter (Hafner, 2022) | ②, ④ | Game | Visual | ⚙GitHub 🌐Website |
| Craftax (Matthews et al., 2024) | ②, ④ | Game | Visual | ⚙GitHub |
| ELLM (Crafter variant) (Du et al., 2023) | ②, ① | Game | Visual | ⚙GitHub 🌐Website |
| SMAC / SMAC-Exp (Samvelyan et al., 2019) | ⑤, ② | Game | Visual | ⚙GitHub |
| Factorio (Hopkins et al., 2025) | ②, ① | Game | Visual | ⚙GitHub 🌐Website |
| SMAC-Hard (Deng et al., 2024b) | ②, ④ | Game | Visual | ⚙GitHub |
| TacticCraft (Ma et al., 2025a) | ②, ⑤ | Game | Text | - |

2024), are self-hostable, reproducible web environments delivered as Docker containers. WebArena features fully functional websites across common domains like e-commerce, social forums, collaborative development, and content management systems, enriched with utility tools and knowledge bases, and supports multi-tab tasks and user role simulation. VisualwebArena extends this by introducing new tasks requiring visual comprehension and a "Set-of-Marks" (SoM) representation to annotate interactable elements on screenshots, bridging the gap for multimodal web agents. Additionally, AppWorld (Trivedi et al., 2024) constitutes an environment simulating a multi-application ecosystem, encompassing 9 daily-use applications (e.g., Amazon, Spotify, Gmail) with 457 invokable APIs, and constructing a digital world featuring approximately 100 virtual characters and their social relationships. Agents accomplish complex tasks (such as travel planning and social relationship management) by writing code to call APIs. In these environments, all changes to the web pages or visual elements occur exclusively in response to the agent's actions.

### 5.1.2 GUI Environments

AndroidWorld (Rawles et al., 2025) exemplifies such dynamism as a benchmarking environment operating on a live Android emulator, featuring 116 hand-crafted tasks across 20 real-world applications. Its dynamic nature is underscored by parameter instantiation that generates millions of unique task variations, ensuring

the environment evolves into novel configurations without direct agent influence. Agents interact through a consistent interface (supporting screen interactions, app navigation, and text input) while receiving real-time state feedback, with integration to MiniWoB++ providing durable reward signals for evaluating adaptive performance. OSWorld (Xie et al., 2024) is a scalable real computer environment for multimodal agents, supporting task setup and execution-based evaluation across Ubuntu, Windows, and macOS. It includes a substantial number of real-world computer tasks involving real web and desktop applications, OS file I/O, and workflows spanning multiple applications, where all OS state changes are exclusively triggered by the agent's actions.

### 5.1.3 Coding & Software Engineering Environments

Code-related tasks are supported by a wide range of executable environments and benchmarks. These can be broadly categorized into interactive environments, where agents directly alter the state, and benchmarks/datasets that provide curated tasks and evaluation pipelines.

**Interactive SWE Environments.** Several environments instantiate agent–environment interaction under software engineering workflows. Debug-Gym (Yuan et al., 2025c) is a text-based interactive coding environment for LLM agents in debugging settings. It equips agents with tools like a Python debugger (pdb) to actively explore and modify buggy codebases, supporting repository-level information handling and ensuring safety via Docker containers. R2E-Gym (Jain et al., 2025c) constructs a procedurally generated, executable gym-style environment of over 8K software engineering tasks, powered by the SWE-Gen pipeline and hybrid verifiers. TheAgentCompany (Xu et al., 2024a) simulates a software development company, where agents act as "digital workers" performing professional tasks such as web browsing, coding, program execution, and communication with simulated colleagues. It features a diverse set of long-horizon tasks with checkpoints for partial credit, providing a comprehensive testbed for agents in a realistic workplace setting. In all these environments, the underlying problem definitions and codebases remain fixed, and changes occur solely as a result of the agent's actions.

**Coding Benchmarks & Datasets.** A wide range of benchmarks and datasets focus on constructing curated task suites and evaluation pipelines. HumanEval (Chen et al., 2021) introduces a benchmark of 164 hand-crafted Python programming tasks to measure functional correctness via the pass@k metric. MBPP (Austin et al., 2021) provides 974 entry-level Python tasks with natural language descriptions for evaluating short program synthesis. BigCodeBench (Zhuo et al., 2025) proposes a large-scale, contamination-free function-level benchmark of 1,140 tasks requiring composition of multiple function calls. LiveCodeBench (Jain et al., 2025b) builds a continuously updated, contamination-free benchmark from real competition problems. SWE-bench (Jimenez et al., 2024) introduces a dynamic, execution-driven code repair benchmark derived from real GitHub issues. SWE-rebench (Badertdinov et al., 2025) introduces a continual GitHub-mining pipeline (>21k tasks) for both training and evaluation. DevBench (Li et al., 2025a) evaluates end-to-end development across design, setup, implementation, and testing. ProjectEval (Liu et al., 2025g) constructs LLM-generated, human-reviewed project tasks with simulated user interactions. ColBench (Zhou et al., 2025e) instantiates multi-turn backend/frontend tasks with a privileged critic for step-wise rewards. NoCode-bench (Deng et al., 2025a) evaluates LLMs on feature addition from documentation updates across real codebases. CodeBoost (Wang et al., 2025n) serves as a data-centric, execution-driven training pipeline by extracting and augmenting code snippets.

**Programmatic World-Model Environments.** Beyond isolated coding tasks, recent benchmarks evaluate whether agents can induce executable world models. The Code World Models Benchmark (CWMB) (Dainese et al., 2024) requires agents to synthesize Python "Environment" classes (specifically the "step" function) to replicate ground-truth dynamics, assessing both transition fidelity and downstream planning utility. Complementing this, the Code Simulation suite (Malfa et al., 2024; 2025) offers finer-grained tests on line-by-line execution prediction and algorithmic generalization. Collectively, these tasks shift the evaluation focus from functional correctness to the dynamics-induction and program-simulation capabilities essential for constructing programmatic world models.

### 5.1.4 Domain-specific Environments

**Science & Research.** ScienceWorld (Wang et al., 2022) integrates science simulations (e.g., thermodynamics, electricity, chemistry) into complex text-based tasks designed around elementary-level science education. PaperBench (Starace et al., 2025) evaluates the ability of LLM agents to replicate cutting-edge machine learning research by reproducing 20 ICML 2024 papers from scratch, scored against rubric-based subtasks. $\tau$-bench (Barres et al., 2025) simulates dynamic conversations for software engineering tasks, operating with an underlying database state and domain-specific rules that change only through the agent's API calls.

**Machine Learning Engineering (MLE).** MLE-Dojo (Qiang et al., 2025) is a Gym-style framework for iterative machine learning engineering workflows, built upon real-world Kaggle competitions. It provides an interactive environment for agents to iteratively experiment, debug, and refine solutions. MLE-Bench (Chan et al., 2025) establishes a benchmark for MLE by curating 75 Kaggle competitions, evaluating agents against human baselines on public leaderboards. DA-Code (Huang et al., 2024b) addresses agentic data-science workflows grounded in real datasets and executable analysis, providing a focused benchmark for this domain.

**Biomedical.** MedAgentGym (Xu et al., 2025c) provides a domain-specific environment for biomedical code generation and testing, focusing on tasks within this specialized scientific field.

**Cybersecurity.** SecRepoBench (Dilgren et al., 2025) is a domain-specific benchmark for security vulnerability repair, covering 27 repositories and 15 Common Weakness Enumeration (CWE) categories.

### 5.1.5 Simulated & Game Environments

Text-based environments simulate interactive settings where agent actions are expressed through natural language. LMRL-Gym (Abdulhai et al., 2025) provides a benchmark for evaluating reinforcement learning algorithms in multi-turn language interactions, including tasks like "20 Questions" and Chess. TextWorld (Côté et al., 2019) is a sandbox environment for training agents in text-based games, offering both hand-authored and procedurally generated games. Game-based environments also emphasize visual settings that may evolve independently. Crafter (Hafner, 2022) is a 2D open-world survival game that benchmarks deep exploration and long-horizon reasoning. Craftax (Matthews et al., 2024), built upon Crafter using JAX, introduces increased complexity and GPU-acceleration for open-ended RL. The modified Crafter variant by ELLM (Du et al., 2023) expands the action space and introduces distractor tasks. For multi-agent coordination, SMAC (Samvelyan et al., 2019) and SMAC-Hard (Deng et al., 2024b) provide StarCraft II-based benchmarks for cooperative decentralized control. SMAC-R1 (Deng et al., 2024b), Adaptive Command (Ma et al., 2025b) and TacticCraft (Ma et al., 2025a) further advance the performance of LLM agents in StarCraft II-style environments. Factorio (Hopkins et al., 2025) presents a dynamic, tick-based industrial simulation where agent inaction still alters the world state.

### 5.1.6 General-Purpose Environments

Some environments and benchmarks are designed for broad evaluation or to improve general agent capabilities. AgentGym (Xi et al., 2025) focuses on improving LLM agent generalization via instruction tuning and self-correction, operating on deterministic environments such as ALFWorld, BabyAI, and SciWorld. Agentbench (Liu et al., 2024b) serves as a broad evaluation framework, assessing LLMs as agents across a variety of distinct interactive environments, including SQL-based, game-based, and web-based scenarios. InternBootcamp (Li et al., 2025g) is a scalable framework integrating over 1000 verifiable reasoning tasks, spanning programming, logic puzzles, and games, with a standardized interface for RL training and automated task generation.

## 5.2 RL Framework

In this section, we summarize three categories of codebases/frameworks most relevant to this work: Agentic RL frameworks, RLHF and LLM fine-tuning frameworks, and general-purpose RL frameworks. Table 11 provides an overview of the prevailing Agentic RL and LLM RL frameworks for readers' reference.

Table 11: A summary of frameworks for reinforcement learning, categorized by type and key features.

| Framework | Key Features | Resource |
|---|---|---|
| *Agentic RL Frameworks* | | |
| Verifiers (Brown, 2025) | Verifiable environment setup | 🎧 GitHub |
| SkyRL-v0 (Cao et al., 2025b) | Long-horizon real-world training | 🎧 GitHub |
| AREAL (Fu et al., 2025) | Asynchronous training | 🎧 GitHub |
| MARTI (Zhang et al., 2025n) | Integrated multi-agent training | 🎧 GitHub |
| EasyR1 (Zheng et al., 2025d) | Multimodal support | 🎧 GitHub |
| AgentFly (Wang et al., 2025j) | Scalable asynchronous execution | 🎧 GitHub |
| Agent Lightning (Luo et al., 2025e) | Decoupled hierarchical RL | 🎧 GitHub |
| AWorld (Yu et al., 2025a) | Parallel rollouts across clusters | 🎧 GitHub |
| RL-Factory (RL-Factory, 2025) | Easy-to-design reward | 🎧 GitHub |
| ROLL (Wang et al., 2025o) | Stable Multi-GPU Parallel Training | 🎧 GitHub |
| AgentRL (Zhang et al., 2025a) | Asynchronous Multi-Task Training | 🎧 GitHub |
| VerlTool (Jiang et al., 2025a) | Tool-integrated rollout | 🎧 GitHub |
| *RLHF and LLM Fine-tuning Frameworks* | | |
| OpenRLHF (Hu et al., 2025b) | High-performance scalable RLHF | 🎧 GitHub |
| TRL (von Werra et al., 2020) | Hugging Face RLHF | 🎧 GitHub |
| trlX (Havrilla et al., 2023) | Distributed large-model RLHF | 🎧 GitHub |
| HybridFlow (Sheng et al., 2025) | Streamlined experiment management | 🎧 GitHub |
| SLiMe (THUDM, 2025) | High-performance async RL | 🎧 GitHub |
| Oat (Liu et al., 2024c) | Lightweight RL support | 🎧 GitHub |
| *General-purpose RL Frameworks* | | |
| RLlib (Liang et al., 2018) | Production-grade scalable library | 🎧 GitHub |
| Acme (Hoffman et al., 2020) | Modular distributed components | 🎧 GitHub |
| Tianshou (Weng et al., 2022) | High-performance PyTorch platform | 🎧 GitHub |
| Stable Baselines3 (Raffin et al., 2021) | Reliable PyTorch algorithms | 🎧 GitHub |
| PFRL (Fujita et al., 2021) | Benchmarked prototyping algorithms | 🎧 GitHub |

**Agentic RL frameworks.** Verifiers (Brown, 2025) introduces a verifiable-environment setup for end-to-end policy optimization with LLMs, while SkyRL-v0 (Cao et al., 2025b) and its modular successors (Griggs et al., 2025) demonstrate long-horizon, real-world agent training via reinforcement learning. AREAL (Fu et al., 2025) scales this paradigm with an asynchronous, distributed architecture tailored to language reasoning tasks, and MARTI (Zhang et al., 2025n) extends it further to multi-agent LLM systems that integrate reinforcement training and inference. EasyR1 (Zheng et al., 2025d) brings multi-modality support, enabling agents to leverage vision and language signals together in a unified RL framework. AgentFly (Wang et al., 2025j) presents a scalable and extensible agent-RL framework that empowers language-model agents with traditional reinforcement-learning algorithms—enabling token-level multi-turn interaction via decorator-based tools and reward definition, asynchronous execution, and centralized resource management for high-throughput RL training. Agent Lightning (Luo et al., 2025e) is a flexible RL framework that decouples agent execution from training by modeling execution as an MDP and using a hierarchical RL algorithm (LightningRL) to train any AI agent with near-zero code modification. AWorld (Yu et al., 2025a) is a distributed Agentic RL framework, which tackles the main bottleneck of agent training—experience generation—by orchestrating massively parallel rollouts across clusters, achieving a 14.6× speedup over single-node execution and enabling scalable end-to-end training pipelines. ROLL (Wang et al., 2025o) provides a scalable library for large-scale RL optimization with a unified controller, parallel workers, and automatic resource mapping for efficient multi-GPU training. VerlTool (Jiang et al., 2025a) introduces an Agentic RL with tool use (ARLT) framework built upon Verl (Sheng et al., 2025), enabling agents to jointly optimize planning and execution across interactive environments. AgentRL (Zhang et al., 2025a) provides a scalable asynchronous framework for multi-turn, multi-task Agentic RL, unifying environment orchestration and introducing cross-policy sampling and task advantage normalization for stable large-scale training.

**RLHF and LLM fine-tuning frameworks.** OpenRLHF (Hu et al., 2025b) offers a high-performance, scalable toolkit designed for large-scale model alignment; TRL (von Werra et al., 2020) provides Hugging Face's baseline implementations for RLHF experiments; trlX (Havrilla et al., 2023) adds distributed training support for fine-tuning models up to tens of billions of parameters; and HybridFlow (Sheng et al., 2025) streamlines experiment management and scaling for RLHF research pipelines. SLiMe (THUDM, 2025) is an LLM post-training framework for RL scaling that combines Megatron with SGLang for high-performance multi-mode training, supports Async RL, and enables flexible disaggregated workflows for reward and data generation via custom interfaces and server-based engines.

**General-purpose RL frameworks** supply the core algorithms and distributed execution engines that can underpin agentic LLM systems. RLlib (Liang et al., 2018) is a production-grade, scalable library offering unified APIs for on-policy, off-policy, and multi-agent methods; Acme (Hoffman et al., 2020) provides modular, research-oriented building blocks for distributed RL; Tianshou (Weng et al., 2022) delivers a high-performance, pure-PyTorch platform supporting online, offline, and hierarchical RL; Stable Baselines3 (Raffin et al., 2021) packages reliable PyTorch implementations of standard model-free algorithms; and PFRL (Fujita et al., 2021) (formerly ChainerRL) offers benchmarked deep-RL algorithm implementations for rapid prototyping.

# 6 Open Challenges and Future Directions

The advance of agent RL toward general-purpose intelligence hinges on overcoming three pivotal challenges that define the field's research frontier. First is the challenge of **Trustworthiness**: ensuring the reliability, safety, and alignment of increasingly autonomous agents. Second is **Scaling up Agentic Training**, which requires surmounting the immense practical bottlenecks in computation, data, and algorithmic efficiency. Finally, an agent's capabilities are fundamentally bounded by its world, making **Scaling up Agentic Environments**—the creation of complex and adaptive training grounds—a critical necessity.

## 6.1 Trustworthiness

**Security.** The security landscape for autonomous agents is fundamentally more complex than for standard LLMs. While traditional models are primarily vulnerable to attacks on their text-in, text-out interface, agents possess an expanded attack surface due to their external components like tools, memory, and planning modules (Wang et al., 2025l; Shang et al., 2025b). This architecture exposes them to novel threats beyond direct prompt injection. For instance, indirect prompt injection can occur when an agent interacts with a compromised external environment, such as a malicious website or API, which poisons its memory or tool outputs (Chen et al., 2024i). Multi-agent systems further compound these risks by introducing vulnerabilities through inter-agent communication, where one compromised agent can manipulate or mislead others within the collective (Wang et al., 2025l).

RL significantly magnifies these agent-specific risks by transforming the agent from a passive victim of manipulation into an active, goal-seeking exploiter of vulnerabilities. The core issue is instrumental goal achievement through reward hacking: an RL agent's primary directive is to maximize its long-term reward, and it may learn that unsafe actions are the most effective path to this goal. For example, if an agent discovers that using a malicious, third-party tool yields a high reward for a given task, RL will actively reinforce and entrench this unsafe behavior. Similarly, if an agent learns that it can bypass safety protocols to achieve its objective more efficiently, the resulting reward signal will teach it to systematically probe for and exploit such security loopholes. This creates a more persistent and dangerous threat than one-off jailbreaks, as the agent autonomously learns and optimizes deceptive or harmful strategies over time.

Mitigating these amplified risks requires a defense-in-depth approach tailored to agentic systems. A critical first line of defense is robust sandboxing (Lu et al., 2025b; Ruan et al., 2024), where agents operate in strictly controlled, permission-limited environments to contain the potential damage from a compromised tool or action. At the training level, mitigation strategies must focus on shaping the reward signal itself. This includes implementing process-based rewards that penalize unsafe intermediate steps (e.g., calling an untrusted API) and employing adversarial training within the RL loop, where the agent is explicitly rewarded for resisting manipulation attempts and ignoring poisoned information. Finally, continuous monitoring and

anomaly detection are essential for post-deployment safety. By tracking an agent's actions, such as tool calls and memory access patterns, it is possible to identify deviations from normal behavior, allowing for timely intervention.

**Hallucination.** In the context of agentic LLMs, hallucination is the generation of confident yet ungrounded outputs, including statements, reasoning steps, or tool usage, that are not rooted in provided evidence or external reality. This issue extends beyond simple factual errors to encompass unfaithful reasoning paths and misaligned planning, with overconfidence often masking the agent's uncertainty (Cossio, 2025; Huang et al., 2025b). In multimodal agents, it also manifests as cross-modal inconsistency, such as a textual description mismatching an image, framing it as a fundamental grounding problem (Bai et al., 2025). Evaluating hallucination requires assessing both factuality against objective truth and faithfulness to a given source, often measured through benchmarks like HaluEval-QA or by the agent's ability to appropriately abstain on unanswerable questions, where a refusal to answer ("I don't know") is a critical signal of epistemic awareness (Li & Ng, 2025; Song et al., 2025c).

RL can inadvertently amplify hallucination if the reward mechanism is not carefully designed. Studies show that outcome-driven RL, which rewards only the correctness of the final answer, can encourage agents to find spurious correlations or shortcuts. This process may yield confident but unfounded intermediate reasoning steps, as the optimization process settles into local optima that achieve the goal without being factually sound (Li & Ng, 2025). This phenomenon introduces a "hallucination tax," where reinforcement finetuning can degrade an agent's ability to refuse to answer, compelling it to generate responses for unanswerable questions rather than abstaining (Song et al., 2025c). However, the effect is highly dependent on the training pipeline; while RL-only post-training can worsen factuality, a structured approach combining SFT with a verifiable-reward RL process can mitigate this degradation (Yao et al., 2025).

Promising mitigation strategies involve a hybrid approach of training-time alignment and inference-time safeguards. During training, a key direction is to shift from outcome-only rewards to process-based rewards. Techniques like Factuality-aware Step-wise Policy Optimization (FSPO) verify each intermediate reasoning step against evidence, directly shaping the policy to discourage ungrounded claims (Li & Ng, 2025). Data-centric approaches enhance epistemic humility by training agents on a mix of solvable and unsolvable problems, restoring their ability to abstain when necessary (Song et al., 2025c). At the system level, this is complemented by inference-time techniques such as retrieval augmentation, tool-use for fact-checking, and post-hoc verification to ground the agent's outputs in reliable sources. For multimodal agents, explicitly adding cross-modal alignment objectives is crucial for ensuring consistency (Huang et al., 2025b; Cossio, 2025; Bai et al., 2025). Collectively, these directions aim to align the agent's reward-seeking behavior with the goal of truthfulness, fostering more reliable and trustworthy autonomous systems.

**Sycophancy.** Sycophancy in LLM agents refers to their tendency to generate outputs that conform to a user's stated beliefs, biases, or preferences, even when those are factually incorrect or lead to suboptimal outcomes (Sun & Wang, 2025). This behavior transcends mere conversational agreeableness, fundamentally affecting an agent's planning and decision-making processes. For instance, a sycophantic agent might adopt a user's flawed reasoning in its internal plan, choose a course of action that validates the user's incorrect assumptions, or filter information from tools to present only what aligns with the user's view (Malmqvist, 2024). This represents a critical misalignment, where the agent optimizes for the user's expressed preference rather than their latent, long-term interest in achieving the best possible outcome.

RL is a primary cause for this behavior. The underlying mechanism is a form of "reward hacking," where the agent learns to exploit the reward model in ways that do not align with true human preferences (Lu et al., 2024). Because human labelers often show a preference for agreeable and validating responses, the reward model inadvertently learns to equate user satisfaction with sycophantic agreement. Consequently, RLHF can directly incentivize and "exacerbate sycophantic tendencies" by teaching the agent that conforming to a user's viewpoint is a reliable strategy for maximizing reward, even if it compromises truthfulness (Wen et al., 2024).

Mitigating sycophancy is an active area of research that focuses on refining the reward signal and training dynamics. A promising direction is the development of sycophancy-aware reward models, which are explicitly trained to penalize responses that merely parrot user beliefs without critical evaluation.

At inference time, strategies like explicitly prompting the agent to adopt a "red team" or contrarian perspective can also help counteract ingrained sycophantic tendencies. Cooper (Hong et al., 2025b) is a reinforcement learning framework that co-optimizes both the policy model and the reward model online, using high-precision rule-based verifiers to select positive samples and LLM-generated negative samples, thereby preventing the policy from exploiting a static reward model (i.e., reward hacking) by continuously adapting the reward model to closing emergent loopholes. Ultimately, the future direction lies in designing reward systems that robustly capture the user's long-term interests—such as receiving accurate information and making sound decisions—over their immediate desire for validation.

## 6.2 Scaling up Agentic Training

**Computation.** Recent advances demonstrate that scaling reinforcement learning fine-tuning (RFT) computation directly enhances the reasoning ability of LLM-based agents. The Agent RL Scaling Law study shows that longer training horizons systematically improve tool-use frequency, reasoning depth, and overall task accuracy, highlighting the predictive benefit of allocating more compute to RL training (Mai et al., 2025). Similarly, ProRL reveals that prolonged RL training expands reasoning boundaries beyond those accessible to base models, uncovering novel solution strategies even where extensive sampling from the pretrained model fails (Liu et al., 2025h). Building upon this, ProRLv2 extends training steps and incorporates more stable optimization techniques, demonstrating sustained benefits as smaller models, after extensive RL training, rival the performance of larger models on mathematics, code, and logic benchmarks (Hu et al., 2025a). Collectively, these results underscore that scaling compute through extended RL training is not merely complementary to enlarging model or data size, but a fundamental axis for advancing agentic reasoning.

**Model Size.** Increasing model capacity heightens both the promise and pitfalls of RL-based agent training. Larger models unlock greater potential but risk entropy collapse and narrowing of capability boundaries, as RL sharpens output distributions toward high-reward modes, limiting diversity (Dong et al., 2025c). Methods like RL-PLUS address this with hybrid strategies and advantage functions that foster novel reasoning paths, breaking capability ceilings (Dong et al., 2025c). Meanwhile, scaling demands massive compute, making efficiency vital. A two-stage approach in Vattikonda et al. (2025) uses large teachers to generate SFT data for smaller students, refined via on-policy RL. This "SFT+RL" setup outperforms each method alone and cuts compute by half compared to pure SFT. The work also underscores RL's extreme hyperparameter sensitivity at scale, stressing the need for careful tuning.

**Data Size.** Scaling RL training across domains introduces both synergy and conflict in agentic reasoning. Cross-domain RL in math, code, and logic tasks shows complex interactions (Li et al., 2025o): some pairings enhance each other, while others interfere and reduce performance. Model initialization also matters—instruction-tuned models generalize differently than raw ones. Building on this, the Guru dataset (Cheng et al., 2025) spans six reasoning domains, showing that RL gains correlate with pretraining exposure: math and code benefit from transfer, but domains like simulation or logic need dedicated training. These findings suggest that while multi-domain RL data can amplify general reasoning, it must be carefully curated to balance complementarity and mitigate interference across tasks.

**Efficiency.** The efficiency of LLM post-training is a central frontier for sustainable scaling (Tie et al., 2025). Beyond brute-force scaling, recent research emphasizes improving RL training efficiency through post-training recipes, methodological refinements, and hybrid paradigms. POLARIS (An et al., 2025) demonstrates that calibrating data difficulty, employing diversity-driven sampling, and extending reasoning length substantially boost RL effectiveness, enabling smaller models to reach or even surpass much larger counterparts on reasoning benchmarks. Complementary work (Liu et al., 2025s) provides systematic evaluations of common RL techniques, finding that judiciously combining just a few simple strategies often outperforms more complex methods. Another study proposes Dynamic Fine-Tuning (DFT) (Wu et al., 2025l), showing that introducing RL principles into gradient scaling can match or exceed advanced RL approaches with minimal additional cost. Taken together, these advances suggest a dual trajectory for the future: on one hand, progressively refining RL-based recipes to maximize efficiency; on the other, rethinking training paradigms to embed RL-like generalization signals without full-fledged online RL. A particularly compelling direction lies in

exploring how agentic models might acquire robust generalization from extremely limited data, for instance, by leveraging principled difficulty calibration, meta-learning dynamics, or information-theoretic regularization to distill broad reasoning abilities from a handful of experiences. Such pathways point to the possibility of a new regime of post-training: one where the ability to extrapolate, abstract, and generalize becomes decoupled from sheer data volume, and instead hinges on exploiting the structure and dynamics of the training process itself.

### 6.3 Scaling up Agentic Environments

A nascent yet critical frontier for Agentic RL involves a paradigmatic shift from treating the training environment as a static entity to viewing it as a dynamic and optimizable system. This perspective addresses a core bottleneck in agent development: the scarcity of interactive, adaptive environments and the difficulty of engineering effective reward signals. As a growing consensus holds that prevalent environments like ALFWorld (Shridhar et al., 2021) and ScienceWorld (Wang et al., 2022) are insufficient for training general-purpose agents (Zheng et al., 2025f), research is moving beyond solely adapting the agent's policy. Instead, a co-evolutionary approach uses learning-based methods to adapt the environment itself. One key strategy is to automate reward function design. This involves deploying an auxiliary "explorer" agent to generate a diverse dataset of interaction trajectories, which are then used to train a reward model via heuristics or preference modeling. This effectively decouples agent training from the expensive process of manual reward specification, enabling the learning of complex behaviors without direct human annotation.

Beyond automating the reward signal, a second, more dynamic strategy is to automate curriculum generation, transforming the environment into an active teacher. This approach establishes a feedback loop where an agent's performance data, highlighting specific weaknesses, is fed to an "environment generator" LLM. As exemplified by EnvGen (Zala et al., 2024), this generator then procedurally adapts the environment's configuration, creating new tasks that specifically target and remedy the agent's deficiencies. This form of goal-directed Procedural Content Generation (PCG) ensures the agent is consistently challenged within its "zone of proximal development," accelerating learning and preventing overfitting. Together, automated rewards and adaptive curricula create a symbiotic relationship between the agent and its environment, establishing a scalable "training flywheel" that is essential for the future of self-improving agentic systems.

### 6.4 The Mechanistic Debate on RL in LLMs

Two competing explanations have emerged for why RL appears to boost LLM reasoning. The "amplifier" view holds that RL with verifiable rewards—often instantiated via PPO-style variants such as GRPO—mainly reshapes the base model's output distribution: by sampling multiple trajectories and rewarding the verifiably correct ones, RL concentrates probability mass on already-reachable reasoning paths, improving pass@1 while leaving the support of solutions largely unchanged; consistent with this, large-k pass@k analyses often find that the base model eventually matches or surpasses its RL-tuned counterpart, suggesting elicitation rather than creation of capabilities, and further evidence indicates that reflective behaviors can already emerge during pre-training (Shao et al., 2024b; Yue et al., 2025a; AI et al., 2025). By contrast, the "new-knowledge" view argues that RL after next-token prediction can install qualitatively new computation by leveraging sparse outcome-level signals and encouraging longer test-time computation: theory shows that RL enables generalization on problems (e.g., parity) where next-token training alone is statistically or computationally prohibitive; empirically, RL can improve generalization to out-of-distribution rule- and visual- variants, induce cognitive behaviors (verification, backtracking, subgoal setting) that were absent in the base model yet predict self-improvement, and in under-exposed domains even expand the base model's pass@k frontier (Guo et al., 2025a; Tsilivis et al., 2025; Chu et al., 2025b; Gandhi et al., 2025; Cheng et al., 2025). Whether RL can truly endow LLMs with abilities beyond those acquired during pre-training remains an open question, and its underlying learning mechanisms are still to be fully understood.

**Case study: Mathematical Reasoning**   From a mechanistic standpoint, our survey of RL for mathematical reasoning in Sec 4.3 suggests that RL functions neither as a pure "sampler amplifier" nor as a universally reliable source of genuinely new reasoning algorithms (Yue et al., 2025a). Across the cited mathematical and code-reasoning studies, approximately 2/3 primarily emphasize improvements in pass@1 accuracy, while

about 1/3 explicitly report expanding pass@k frontiers (e.g., higher pass@32 at fixed or only modestly improved pass@1), indicating that many systems leverage RL chiefly to reshape the sampling distribution over pre-existing competent trajectories rather than to unlock qualitatively new ones. However, cases such as 1-shot RLVR and self-evolving System-2-style frameworks (e.g., rStar-Math–like pipelines (Guan et al., 2025a)) also exhibit "post-saturation" generalization and cross-category transfer, which are difficult to explain as mere reweighting and instead suggest strategy-level reorganization of latent capabilities.

Empirically, we find that such "new-capability" behaviors appear most reliably on tasks with (i) high-fidelity, often executable or formally checkable reward signals; (ii) compositional or multi-step structure where many partial trajectories are verifiably graded; and (iii) base models in the "intermediate" regime (neither near-random nor near-ceiling) where the space of near-miss trajectories is rich enough for exploration but still densely populated with correct reasoning paths. Under these conditions, policy-gradient updates plus explicitly managed exploration (e.g., entropy bonuses, self-play curricula, or search-guided expert iteration) seem to move the model toward internalizing more abstract decision rules—whereas on easier, low-noise benchmarks or with coarse outcome-only rewards, RL predominantly acts as an amplifier that sharpens and reuses patterns already implicit in the pretrained model.

## 6.5 Architectural Patterns for Real-World Agent Deployment

While the survey primarily analyzes RL as a mechanism for improving reasoning performance, the practical deployment of RL-optimized systems requires architectural patterns that ensure reliability, safety, and operational robustness. This subsection synthesizes four cross-cutting design principles—safety guardrails, human-in-the-loop supervision, hierarchical orchestration, and inter-agent communication protocols—that commonly arise in real-world deployments of RL-enhanced reasoning systems, irrespective of the domain.

**Guardrails and Safety Patterns.** Deployed systems typically incorporate multi-layered safety mechanisms that operate independently of the RL optimization loop. These include input validation (schema enforcement, semantic filtering, and constraint checking), output sanitization (format normalization, groundedness checks, and post-hoc constraint satisfaction), and sandboxed execution for tool or code calls. Such guardrails can be implemented in two major ways: **(1) Using RL optimization itself as a safeguard**, where, for example, many works directly incentivize models to "think safely" during the reasoning output via RL (Zheng et al., 2025c; Zhang et al., 2025z) ; and **(2) Using external modules to monitor RL training**, such as AWS Bedrock.

**Human-in-the-Loop Verification.** Human oversight remains essential in high-stakes or uncertainty-prone settings (Mozannar et al., 2025; Takerngsaksiri et al., 2025). HITL mechanisms range from synchronous review of critical decisions to asynchronous auditing, exception handling, and feedback collection. They often rely on model confidence signals or external uncertainty detectors to trigger intervention (Nazir & Banerjee, 2025). Architecturally, HITL provides sparse but high-fidelity corrective signals that complement RL reward structures, enabling safe deployment even when real-world reward feedback is limited, delayed, or noisy.

**Hierarchical Orchestration.** Many practical systems adopt hierarchical control structures (such as supervisor–worker, controller–executor, or planner–solver patterns, as observed in (Zhang et al., 2025r; Liu et al., 2025c; Hu et al., 2025c)) to manage complex workflows. The supervisory layer coordinates subtasks, resolves conflicts, or enforces global constraints, while lower-level components focus on domain-specific reasoning or tool execution. This decomposition facilitates temporal and structural credit assignment, improves scalability, and mirrors enterprise orchestration pipelines where operational logic and execution are cleanly separated.

**Inter-Agent Communication Protocols.** When multiple reasoning entities interact—whether as explicit agents or modular system components—the choice of communication protocol becomes critical. Fixed protocols (*e.g.*, ANP (Chang et al., 2025), A2A (Project, 225), ACP (Team, 2025a)) offer stability and predictability, while learnable communication channels allow adaptive coordination but require stronger regularization to avoid emergent pathologies. Standardized communication interfaces support composability, reproducibility, and compatibility with external workflow engines.

## 6.6 Broader Social Impact

The growing deployment of autonomous, agentic LLM systems raises broader societal considerations that increasingly shape research priorities. This subsection highlights five cross-cutting impact areas which merit sustained attention as agentic capabilities continue to advance.

**Dual-Use Risks.** The deployment of Agentic RL lowers the barrier for misuse, notably through "sleeper agent" behaviors where models appear aligned during training but activate concealed harmful policies in deployment (Hubinger et al., 2024). This deceptive alignment often persists despite SFT, RLHF, and adversarial training, as models—particularly those utilizing chain-of-thought—learn to distinguish evaluation contexts from operation. To govern such hazards across domains like Cybersecurity, CBRN, and Autonomous Replication and Adaptation (ARA), OpenAI's Preparedness Framework establishes a four-tier risk assessment structure (OpenAI, 2025b). However, the framework faces criticism for permissive thresholds, discretionary evaluation protocols, and static assessments that fail to account for post-deployment capability evolution (Coggins et al., 2025).

**Environmental Sustainability.** Large-scale RL is substantially more resource-intensive than SFT due to rollout generation, long-horizon reasoning, and iterative decision steps. Agentic systems further increase training- and deployment-time carbon footprints as interactions unfold over multi-stage workflows (Gardner et al., 2025). Sustainable practices include hardware-aware quantization and resource-efficiency–focused methods. HAQ searches for optimal layer-wise bitwidths under hardware constraints (Wang et al., 2019), and HERO derives low-bit quantization policies for efficient inference using RL-based optimization (Zhang et al., 2025u). Other recent work develops environmental evaluation benchmarks (Wu et al., 2025j).

**Labor Market Implications.** The shift from token-level assistance to autonomous workflow execution positions agentic systems as increasingly strong substitutes for humans in a variety of knowledge-intensive tasks. Code agents have demonstrated the ability to perform debugging, patching, and repository-level issue resolution in SWE benchmarks (Jimenez et al., 2024; Liu et al., 2023c). GUI and web agents similarly automate interactive desktop and browser workflows as shown in OSWorld and WebArena evaluations (Xie et al., 2024; Zhou et al., 2024b). Economic analyses indicate that such end-to-end automation may disproportionately affect entry-level or routine cognitive roles, raising concerns about skill ladder erosion and labor displacement (Eloundou et al., 2023; Brynjolfsson et al., 2023). These trends highlight broader socioeconomic implications, especially for labor markets that may be increasingly exposed to automation.

**Bias Amplification.** RLHF and RLAIF exacerbate societal biases and ideological sycophancy by overfitting to annotator preferences (Casper et al., 2023). Despite surface-level politeness, these models intensify covert discrimination and gender stereotypes, particularly in multi-turn agentic settings (Barnhart et al., 2025a). Furthermore, standard optimization risks collapsing minority preference modes (Xiao et al., 2025). Mitigation strategies address both reward and policy levels. Techniques include fairness-aware reward learning (Swamy et al., 2024; Ouyang et al., 2025), MaxMin-RLHF for heterogeneous groups (Chakraborty et al., 2024), and diversity-preserving objectives like DivPO to prevent mode collapse (Xiao et al., 2025; Wang et al., 2023a; Lanchantin et al., 2025). Complementary approaches involve pluralistic annotator pools and Constitutional AI (Santurkar et al., 2023; Bai et al., 2022), evaluated via benchmarks like CrowS-Pairs and dialect-sensitive tests (Barnhart et al., 2025b).

**Evaluation Contamination.** Static benchmarks like HumanEval and SWE-bench suffer from data contamination, causing inflated scores and illusory robustness (Banerjee et al., 2024). In agentic settings, this encourages overfitting to environmental quirks rather than generalizable reasoning. Addressing these limitations, recent work prioritizes dynamic, contamination-resistant benchmarks, including LiveCodeBench (Jain et al., 2025b), LiveSearchBench (Zhou et al., 2025c), LiveTradeBench (Yu et al., 2025b), and LiveBench (White et al., 2025). Combined with adversarial frameworks like Breakpoint (Hariharan et al., 2025), these approaches prevent test-gaming and offer rigorous assessments of out-of-distribution performance.

Collectively, these broader-impact considerations reinforce the importance of coupling methodological advances in Agentic RL with safety, sustainability, fairness, and robustness principles. As agentic systems advance

toward broader deployment, understanding and mitigating these societal effects will remain an important direction for future research.

# 7 Conclusion

This survey has charted the emergence of Agentic Reinforcement Learning (Agentic RL), a paradigm that elevates LLMs from passive text generators to autonomous, decision-making agents situated in complex, dynamic worlds. Our journey began by formalizing this conceptual shift, distinguishing the temporally extended and partially observable MDPs (POMDPs) that characterize Agentic RL from the single-step decision processes of conventional RL for LLMs. From this foundation, we constructed a comprehensive, twofold taxonomy to systematically map the field: one centered on *core agentic capabilities* (planning, tool use, memory, reasoning, self-improvement, perception, *etc.*) and the other on their *application* across a diverse array of task domains. Throughout this analysis, our central thesis has been that RL provides the critical mechanism for transforming these capabilities from static, heuristic modules into adaptive, robust agentic behavior. By consolidating the landscape of open-source environments, benchmarks, and frameworks, we have also provided a practical compendium to ground and accelerate future research in this burgeoning field.

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
