# OpenReview forum: "The Landscape of Agentic Reinforcement Learning for LLMs: A Survey"
_TMLR — Accepted by TMLR_

### Review · Reviewer_5bUW · 2025-11-23

**Summary Of Contributions:**

Survey examines the paradigm of Agentic RL for LLMs, presenting a systematic framework distinguishing agentic systems from conventional RLHF approaches. The paper synthesizes and proposes theoretical foundation based on the transition from single-step MDPs to temporally extended POMDPs.

The authors make significant contributions:

1. Theoretical Formalization: Establishes a rigorous distinction between conventional LLM-RL (T=1) and Agentic RL (T>1), providing clear definitions of state spaces, action spaces, and learning objectives.
2. Dual Taxonomy Framework: Introduces classification system examining (a) core agentic capabilities where RL serves as the optimization mechanism (planning, tool use, memory, reasoning, self-improvement), and (b) practical applications across domains (search, code generation, vision, embodied agents, multi-agent systems).
3. Resource Collection: Consolidates environments, frameworks, and benchmarks into an actionable reference.
4. Critical Analysis: Identifies key challenges including trustworthiness, scaling limitations, and the debate about whether RL amplifies existing capabilities or enables new behaviors.

Key Strengths:
- Clear formalization of PBRFT from Agentic RL through MDP/POMDP frameworks
- Comprehensive algorithm comparison (PPO, DPO, GRPO families) with trade-offs
- Good coverage of recent developments including GRPO variants and tool-integrated reasoning
- Valuable synthesis connecting foundations to real-world applications

Key Weaknesses(Addressed):
- ~~Requires more depth on architectural patterns and engineering challenges~~
- ~~If possible, coverage of emerging paradigms like Code World Models (CWM)~~
- ~~Taxonomy overlaps between capability and application sections~~

**Additional Comments:**

The revision comprehensively addresses all requested changes and broader impact concerns.

**Audience:**

Yes

**Audience Explanation:**

This survey addresses critical interests of multiple TMLR constituencies:

1. LLM Post-Training Researchers: The comprehensive comparison of RL algorithms (REINFORCE, PPO, DPO, GRPO families) provides actionable guidance for optimization method selection, particularly relevant given recent advances in preference-based training.
2. Agent Systems Developers: As someone working on the same  field, I can attest to the practical value. The classification by task domain and training paradigm directly supports experimental design.
3. Multi-Agent Systems Community: Section 4.7's coverage of MARL with LLMs addresses growing interest in collaborative AI systems, though there is less information regarding orchestration patterns, critical for enterprise-scale deployments.
4. Safety and Alignment Researchers: The trustworthiness discussion (Section 6.1) identifies critical vulnerabilities including security risks, hallucination amplification, and sycophancy.
5. Industry Practitioners: The practical compendium and workflow patterns provide immediate value for teams deploying agentic systems in real-world applications.

The paper's significance extends beyond academic interest, as evidenced by recent industry developments (OpenAI o3, DeepSeek-R1, production agentic deployments).

**Broader Impact Concerns:**

1. Dual-Use Risks
   - Original Request: Cite Anthropic's *Sleeper Agents*, OpenAI Preparedness Framework
   - Status: ADDRESSED
   - Evidence: Section 6.6 cites Hubinger et al. (2024) *Sleeper Agents*, OpenAI's Preparedness Framework, and criticism from Coggins et al. (2025).

2. Environmental Impact
   - Original Request: Carbon footprints, hardware-aware quantization
   - Status: ADDRESSED
   - Evidence: Section 6.6 discusses carbon footprints, HAQ, HERO quantization and environmental benchmarks.

3. Labor Market Effects
   - Original Request: Socioeconomic implications of code/GUI agents
   - Status: ADDRESSED
   - Evidence: Section 6.6 *Labor Market Implications* subsection with citations (Eloundou et al., 2023; Brynjolfsson et al., 2023).

4. Bias Amplification
   - Original Request: Fairness-constrained reward design, bias-aware quality metrics
   - Status: ADDRESSED
   - Evidence: Section 6.6 covers MaxMin-RLHF, DivPO, Constitutional AI and CrowS-Pairs evaluation.

5. Evaluation Contamination
   - Original Request: Benchmark overfitting, dynamic evaluation protocols
   - Status: ADDRESSED
   - Evidence: Section 6.6 discusses contamination, dynamic benchmarks (LiveCodeBench, LiveSearchBench, LiveTradeBench, LiveBench) and Breakpoint framework.

**Claims And Evidence:**

Yes

**Claims Explanation:**

The paper's claims are well-supported through comprehensive literature synthesis. The theoretical distinction between conventional LLM-RL and Agentic RL is mathematically grounded with clear definitions (Equations), distinguishing state spaces (trad <> agent), action spaces (text <> action), and learning objectives (trad <> agent).

The evidence is particularly strong in:
- Algorithm comparison (Table 2): Detailed analysis of RL methods with objectives and trade-offs
- Environment consolidation (Tables 9-10): Structured summaries of environments with characteristics and links
- Capability taxonomy with clear progression paths

However, the paper would benefit from:
- Quantitative comparisons of performance on standardized benchmarks
- Texts supporting the "critical mechanism" thesis for RL
- Deeper analysis of when Agentic RL outperforms alternative approaches

**Requested Changes:**

1. Add Selective Empirical Comparisons [Section 4]
   - Original Request: Include performance infographics (Pass@1 rates on HumanEval/MBPP, Accuracy on MATH/GSM8K)
   - Status: ADDRESSED
   - Evidence: Figure 7 shows pass@1 scores for SWE methods. Figure 8 shows pass@1 scores for Math methods.

2. Address Architectural Gaps [New Section 6.5]
   - Original Request: Guardrails and Safety Patterns, Human-in-the-Loop (HITL), Hierarchical Orchestration, Inter-Agent Communication (A2A)
   - Status: ADDRESSED
   - Evidence: Section 6.5 added with subsections for Guardrails, HITL Verification, Hierarchical Orchestration, and Inter-Agent Communication Protocols.

3. Enhance Taxonomy Navigation [Sections 3-4]
   - Original Request: Capability-application matrix showing primary <> secondary <> minimal dependencies
   - Status: ADDRESSED
   - Evidence: Table 9 added showing *Application-capability dependency matrix* with qualitative levels.

4. Strengthen Mechanistic Analysis [Section 6.4]
   - Original Request: Synthesize *amplifier vs new-knowledge* debate with pass@k vs pass@1 analysis
   - Status: ADDRESSED
   - Evidence: Section 6.4 expanded with Mathematical Reasoning case study.

5. Can deepen Multi-Agent Synthesis [Section 4.7]
   - Original Request: Comparative MARL analysis, credit assignment, communication protocols, hierarchical control patterns
   - Status: ADDRESSED
   - Evidence: Section 4.7 restructured into 4.7.1 (Non-Parametric Coordination), 4.7.2 (Selected Agent Policies), 4.7.3 (End-to-End MARL).

6. Can include Code World Models [Sections 4.2, 5.1]
   - Original Request: Facebook's 32B CWM, simulated execution environments, hardware-aware quantization
   - Status: ADDRESSED
   - Evidence: Section 4.2.4 *Emerging Paradigms* added covering GIF-MCTS, WorldCoder, Meta's 32B CWM. Section 5.1 includes CWMB benchmark.

7. Can add Integration Examples
   - Original Request: OpenAI o3, Voyager, multi-agent supervisor-worker architectures
   - Status: ADDRESSED
   - Evidence: *Integration Case Study: Voyager* in Section 4.6. *Integration Case Study: OpenAI Deep Research* in Section 4.1.2.

---

### Review · Reviewer_HPau · 2025-12-03

**Summary Of Contributions:**

**Summary of Contributions**

This paper provides the first comprehensive survey of Agentic Reinforcement Learning, a paradigm that reconceptualizes LLMs as autonomous, decision-making agents embedded within sequential decision processes rather than static conditional generators. The paper makes several key contributions: (1) it formalizes the distinction between conventional LLM-RL and Agentic RL through an MDP/POMDP framework; (2) it introduces a twofold taxonomy organizing the field around both core agentic capabilities (planning, tool use, memory, reasoning, self-improvement, and perception) and application domains (search, code, math, GUI, vision, embodied agents, and multi-agent systems); (3) it consolidates the landscape of open-source environments, benchmarks, and frameworks into a practical compendium for researchers; and (4) it synthesizes over five hundred recent works to identify open challenges and future directions.

**Strengths**

The dual taxonomy, i.e. organizing by capability (planning, tool use, memory, etc.) and by application domain (search, code, math, GUI, etc.), offers complementary ways to understand the literature. This structure makes the survey navigable for readers. Furthermore, the consolidation of environments, benchmarks, and frameworks (Section 5) provides actionable resources for practitioners and the tables serve as useful quick references linking methods, models, reward types, code repositories etc.

**Weaknesses**

See _Requested Changes_.

**Additional Comments:**

N/A

**Audience:**

Yes

**Audience Explanation:**

Agentic reinforcement learning for LLMs is a rapidly evolving and highly active research area, as evidenced by the volume of recent work the survey covers. A comprehensive synthesis that organizes the literature by both capabilities and applications, and consolidates available environments and frameworks would be valuable to researchers.

**Claims And Evidence:**

Yes

**Claims Explanation:**

This is a survey paper that synthesizes and organizes existing literature rather than presenting novel empirical or theoretical claims requiring validation.

**Requested Changes:**

1. Abstract error: There is a redundancy in the abstract: “the partially observable, temporally extended partially observable Markov decision process (POMDP)” contains “partially observable” twice. Please correct.

2. RLHF is cited as (Ouyang et al., 2022), which is the InstructGPT application paper. The foundational RLHF work should also cite Christiano et al. (2017) “Deep Reinforcement Learning from Human Preferences.”

3. When discussing RLAIF and DPO, the citations point to surveys rather than original work. Please add citations to the original Constitutional AI paper (Bai et al., 2022) and the original DPO paper (Rafailov et al., 2023).

4. Unsupported claim: The statement “procuring sufficient high-quality SFT data remains challenging” would benefit from a citation.

5. Section 2 implicit distinction: There is an implicit differentiation between “human feedback” and “data preferences” when discussing RLHF vs DPO. However, DPO is mathematically equivalent to the original RLHF formulation under certain assumptions—the optimization goal is arguably the same, just accomplished in one step versus two. This distinction could be clarified or softened.

6. Clarification needed for Figure 2: I fail to grasp what intuition Figure 2 is trying to convey. Why does it have this particular shape? What are the arrows pointing to? Please add an explanation in the caption.

7. Section 2.4 on PBRFT transitions: The claim that PBRFT has deterministic transitions (Equation 6) may be overstated. In the RLHF setup, if the same question is posed to multiple annotators, there is inherent stochasticity. For example, for the question “What are better, dogs or cats?”, if shown to 3 annotators and 2 prefer “dogs,” then $P(s_1|s_0, a=\text{dogs})=0.66$ , not $1$. Datasets like the collective alignment dataset or the community alignment dataset have this characteristic. Please clarify in the response, in case I misunderstood, or correct this claim.

8. REINFORCE is presented in Section 2.7 without citation. Please cite Williams (1992) “Simple statistical gradient-following algorithms for connectionist reinforcement learning.”

9. After Equation 12: It would be helpful to add a comment noting that $b(s_0)$ is typically instantiated as the value function $V(s_0)$ in practice.


10. After Equation 14: The text states the critic network is “of the same size” as the policy network. While this is common practice (and some libraries share parameters between policy and critic), it is not a requirement. Please either remove this claim or qualify it with “typically” or “often.”

11. Right before Equation 16, DeepSeek is mentioned without a citation to the relevant work.

12. IPO description (Table 2 and main text): The description of IPO is inaccurate. IPO does not merely “add a regularization term”—it proposes a fundamentally different optimization objective. Additionally, the table description claims IPO reduces dependence on human feedback by using LLMs as preference classifiers, but those LLMs must still be trained on human feedback, so the reduction in dependence is arguable. Please revise.

13. Clarification needed for Figure 3: Similarly, Figure 3 is confusing. What is it trying to convey? Also, what is the “Team” task domain supposed to represent?

14. Citation style inconsistency: Please revise citation style for consistency. For example, “Representative works like RAP Hao et al. (2023)” should be “Representative works like RAP (Hao et al., 2023).” This pattern appears in multiple places throughout the manuscript.

15. Section 3.1, MAP paragraph: The discussion of MAP feels out of place in a paragraph about RL, as the authors themselves note it is “not RL-based.” Consider moving this to a separate paragraph or subsection on non-RL approaches.

16. Page 11, Section 3.2, missing citations: The sentence “while early paradigms…” lacks citations for the paradigms being referenced. Please add citations.

17. Page 15, vague reference: The phrase “and others (Gou et al., 2024)” is unclear. Please clarify what “others” refers to and why Gou et al. is the representative citation.

18. Section 3.5, Reasoning - Misattribution of foundational concepts: The survey states “Reasoning in large language models can be broadly categorized into fast reasoning and slow reasoning, following the dual-process cognitive theory (Ke et al., 2025; Kumar et al., 2025).” This is a misattribution. Kumar et al. (2025) is itself a survey, not the originator of dual-process theory. The foundational work on fast vs. slow thinking (System 1 vs. System 2) should cite Kahneman’s “Thinking, Fast and Slow” (2011) or the earlier academic work by Kahneman, Tversky, Stanovich, and West. The surveys by Ke et al. and Kumar et al. may be relevant for how these concepts have been applied to LLMs, but this should be made explicit (e.g., “as discussed in recent surveys (Ke et al., 2025; Kumar et al., 2025), building on the dual-process cognitive theory of Kahneman (2011)”). Please revise to properly attribute the foundational cognitive science work.

19. In page 28, section “Hybrid reward RL”, please rewrite “Polu and Sutskever (Polu & Sutskever, 2020) introduces” as either
“Polu and Sutskever (2020) introduce…”
or
“Prior work (Polu & Sutskever, 2020) introduces…”

20. In page 33, MALT is in section RL free, while in the table 8, page 34, it is in RL Based. Please correct this.

21. In page 38, could you provide citations supporting the claim that Constitutional AI helps with sycophancy? Is there mixed evidence or is this a known result?

---

> ### Author Response · Authors · 2025-12-10
> **Response Q1-6 to Reviewer HPau**
>
> We sincerely thank Reviewer HPau for the insightful guidance, which was crucial in improving the quality of our paper. We have made the following revisions as suggested and have uploaded the updated PDF (with the corresponding changes highlighted in blue):
>
> ---
>
> **Q1**.Abstract error: There is a redundancy in the abstract: “the partially observable, temporally extended partially observable Markov decision process (POMDP)” contains “partially observable” twice. Please correct.
>
> **A1**: Thanks for pointing out. We've removed the redundancy.
>
> ---
>
> **Q2**.RLHF is cited as (Ouyang et al., 2022), which is the InstructGPT application paper. The foundational RLHF work should also cite Christiano et al. (2017) “Deep Reinforcement Learning from Human Preferences.”
>
> **A2**:We correct the citation of original RLHF to (Christiano et al., 2017), and leave (Ouyang et al., 2022) as an example of "In early RFT research..."
>
> ---
>
> **Q3**.When discussing RLAIF and DPO, the citations point to surveys rather than original work. Please add citations to the original Constitutional AI paper (Bai et al., 2022) and the original DPO paper (Rafailov et al., 2023).
>
> **A3**:Thank you for pointing this out. We have updated the citations so that RLAIF and DPO now reference their original works (Bai et al., 2022; Rafailov et al., 2023) at the appropriate locations. The survey citations are retained as sentence-level references to contextualize the broader discussion.
>
> ---
>
> **Q4**.Unsupported claim: The statement “procuring sufficient high-quality SFT data remains challenging” would benefit from a citation.
>
> **A4**:Thanks for poining out. We've added 3 citations to support the statement. specifically, (Maosongcao et al., 2025) mentioned "the availability of high‑quality human‑annotated SFT data has become a significant bottleneck for LLMs"; (Szep et al., 2025) mentioned "..., emphasizing that data quality (and not quantity) is often the limiting factor" and "Collecting high-quality human preference data is resource-intensive."; (Han et al., 2025) mentioned "Access to high-quality domain-specific data remains a significant bottleneck"
>
> ---
>
> **Q5**.Section 2 implicit distinction: There is an implicit differentiation between “human feedback” and “data preferences” when discussing RLHF vs DPO. However, DPO is mathematically equivalent to the original RLHF formulation under certain assumptions—the optimization goal is arguably the same, just accomplished in one step versus two. This distinction could be clarified or softened.
>
> **A5**:We have added a footnote in section 2 to clarify this point: "Although DPO is another form of optimization objective in RLHF, its complexity is optimized from the perspective of the training process, so it is necessary to distinguish between pure RLHF and DPO." We believe this addition helps to highlight the different perspectives without creating unnecessary confusion between the two approaches.
>
> ---
>
> **Q6**.Clarification needed for Figure 2: I fail to grasp what intuition Figure 2 is trying to convey. Why does it have this particular shape? What are the arrows pointing to? Please add an explanation in the caption.
>
> **A6**:Thank you for pointing out that the intuition behind Figure 2 was not sufficiently clear. We have revised the caption to explicitly explain (1) why the figure has a fan-shaped structure and (2) what the outward-pointing arrows indicate. The updated caption now clarifies that the shape reflects the progressive expansion of the RL formulation—from traditional RL, to LLM-RL, and finally to agentic RL—while the arrows illustrate the increasing breadth of interaction capabilities (e.g., tool use, web browsing, long-term exploration) as one moves outward. We also added explicit descriptions of the color-coded regions, which distinguish features of LLM-RL, features required for agentic RL, and representative implementations.
>
>
> ---

---

> ### Author Response · Authors · 2025-12-10
> **Response Q7-14 to Reviewer HPau**
>
> ---
>
> **Q7**.Section 2.4 on PBRFT transitions: The claim that PBRFT has deterministic transitions (Equation 6) may be overstated. In the RLHF setup, if the same question is posed to multiple annotators, there is inherent stochasticity. For example, for the question “What are better, dogs or cats?”, if shown to 3 annotators and 2 prefer “dogs,” then $P(s_1|s_0, a=\text{dogs})=0.66$ , not $1$. Datasets like the collective alignment dataset or the community alignment dataset have this characteristic. Please clarify in the response, in case I misunderstood, or correct this claim.
>
> **A7**:
> We understand the reviewer's confusion and offer the following explanation:
>
> Although the annotations by data annotators are random, they remain unchanged once the annotation process is complete. From the model's perspective, the observed data is always static and fixed, and no additional randomness is introduced during training. What we want to illustrate is that, in an agentic scenario, the data observed by the model can still continuously change due to the conditional probability transition function of the environment itself, introducing cognitive and accidental uncertainties into the training process.
>
> ---
>
> **Q8**.REINFORCE is presented in Section 2.7 without citation. Please cite Williams (1992) “Simple statistical gradient-following algorithms for connectionist reinforcement learning.”
>
> **A8**:We've added the citation (Williams, 1992) after REINFORCE.
>
> ---
>
> **Q9**.After Equation 12: It would be helpful to add a comment noting that $b(s_0)$ is typically instantiated as the value function $V(s_0)$ in practice.
>
> **A9**:Following your suggestion, we've added a comment after Eq(12) "In practice, $b(s)$ is commonly instantiated as the value function $V(s)$."
>
> ---
>
> **Q10**.After Equation 14: The text states the critic network is “of the same size” as the policy network. While this is common practice (and some libraries share parameters between policy and critic), it is not a requirement. Please either remove this claim or qualify it with “typically” or “often.”
>
> **A10**:Thank you for the suggestion. We have clarified this point by adding “typically, but not necessarily” to avoid implying a strict requirement.
>
> ---
>
> **Q11**.Right before Equation 16, DeepSeek is mentioned without a citation to the relevant work.
>
> **A11**:We've added the citation to DeepSeek as (Guo et al., 2025a).
>
> ---
>
> **Q12**.IPO description (Table 2 and main text): The description of IPO is inaccurate. IPO does not merely “add a regularization term”—it proposes a fundamentally different optimization objective. Additionally, the table description claims IPO reduces dependence on human feedback by using LLMs as preference classifiers, but those LLMs must still be trained on human feedback, so the reduction in dependence is arguable. Please revise.
>
> **A12**:As you suggested, we have revised the descriptions of IPO in the main text (paragraph under Eq. (15)) and Table 2.
>
> ---
>
> **Q13**.Clarification needed for Figure 3: Similarly, Figure 3 is confusing. What is it trying to convey? Also, what is the “Team” task domain supposed to represent?
>
> **A13**:Thank you for the feedback on Figure 3. We have revised the caption to clearly state that the figure illustrates the dynamic interaction loop between an agentic LLM and its environment, highlighting how core agentic capabilities support action generation and how environments provide rewards across task domains. We also replaced “Team” with “Collab.” and added an explicit explanation in the caption indicating that it refers to tasks involving division of labor and team collaboration.
>
>
> ---
>
> **Q14**.Citation style inconsistency: Please revise citation style for consistency. For example, “Representative works like RAP Hao et al. (2023)” should be “Representative works like RAP (Hao et al., 2023).” This pattern appears in multiple places throughout the manuscript.
>
> **A14**: Thank you for pointing out the citation style inconsistencies. We have reviewed all affected instances and standardized them according to the recommended pattern. Specifically, we converted occurrences of inline author–year text to \citep (except when the citation functions as a grammatical component, where \citet is retained). Examples corrected include RAP, LATS, ROLL, Goodfellow et al., and items in Table 5. We also ensured constructions such as “\citep{…}” are used consistently rather than “\cite{…}” for non-sentence-integrated citations.
>
> ---

---

> ### Author Response · Authors · 2025-12-10
> **Response Q15-21 to Reviewer HPau**
>
> ---
>
> **Q15**.Section 3.1, MAP paragraph: The discussion of MAP feels out of place in a paragraph about RL, as the authors themselves note it is “not RL-based.” Consider moving this to a separate paragraph or subsection on non-RL approaches.
>
> **A15**:We've moved the non-RL method MAP discussion to the first paragraph of Section 3.1, as an example of prompting-based methods.
>
> ---
>
> **Q16**.Page 11, Section 3.2, missing citations: The sentence “while early paradigms…” lacks citations for the paradigms being referenced. Please add citations.
>
> **A16**:Thanks for the suggestion, we've added 2 citations to support the statement (Chen et al., 2024g; Kavathekar et al., 2025)
>
> ---
>
> **Q17**.Page 15, vague reference: The phrase “and others (Gou et al., 2024)” is unclear. Please clarify what “others” refers to and why Gou et al. is the representative citation.
>
> **A17**:Thank you for the comment. We have removed the vague phrase "and others" and replaced it with the explicit description “and tool-interactive critiquing mechanisms”, which more accurately reflects what (Gou et al., 2024) represents in this context.
>
> ---
>
> **Q18**.Section 3.5, Reasoning - Misattribution of foundational concepts: The survey states “Reasoning in large language models can be broadly categorized into fast reasoning and slow reasoning, following the dual-process cognitive theory (Ke et al., 2025; Kumar et al., 2025).” This is a misattribution. Kumar et al. (2025) is itself a survey, not the originator of dual-process theory. The foundational work on fast vs. slow thinking (System 1 vs. System 2) should cite Kahneman’s “Thinking, Fast and Slow” (2011) or the earlier academic work by Kahneman, Tversky, Stanovich, and West. The surveys by Ke et al. and Kumar et al. may be relevant for how these concepts have been applied to LLMs, but this should be made explicit (e.g., “as discussed in recent surveys (Ke et al., 2025; Kumar et al., 2025), building on the dual-process cognitive theory of Kahneman (2011)”). Please revise to properly attribute the foundational cognitive science work.
>
> **A18**:Thank you for the comment. We've corrected the attribution and cited the relevant literature accordingly:
> "... building on the dual-process cognitive theory (Kahneman, 2011; Kahneman & Tversky, 1974; Stanovich & West, 2000), as discussed in recent surveys (Ke et al., 2025; Kumar et al., 2025)."
>
> ---
>
> **Q19**.In page 28, section “Hybrid reward RL”, please rewrite “Polu and Sutskever (Polu & Sutskever, 2020) introduces” as either “Polu and Sutskever (2020) introduce…” or “Prior work (Polu & Sutskever, 2020) introduces…”
>
> **A19**:Following your suggestion, we've rewrite it to "Prior work (Polu & Sutskever, 2020)".
>
> ---
>
> **Q20**.In page 33, MALT is in section RL free, while in the table 8, page 34, it is in RL Based. Please correct this.
>
> **A20**:Thanks for pointing out one of our important mistakes. We've moved MATL to RL-based discussions, under Section 4.7.2 RL-Driven Optimization of Selected Agent Policies. (We've reorginzed the Section 4.7 in the last revision to reflect the other reviewer's comment)
>
> ---
>
> **Q21**.In page 38, could you provide citations supporting the claim that Constitutional AI helps with sycophancy? Is there mixed evidence or is this a known result?
>
> **A21**: Thank you for raising this point. As we could not identify reliable peer-reviewed evidence supporting the claim, we have removed the original sentence beginning with "Another approach involves leveraging AI-driven feedback, such as in Constitutional AI, ..."
>
> Some unoffical documents include:
> https://github.com/eliseealex/sycophancy-reduction-cai
> https://docs.google.com/document/d/1_b853mHN2-Pq-DtuswMOX4BoeQWa0kCdCXqrm4NtgoY/edit?pli=1&tab=t.0
>
> ---

---

> ### Author Response · Authors · 2025-12-10
> **Thank you Reviewer 5bUW!**
>
> We have revised the relevant content in the article according to your instructions, and highlighted the changes in blue font. If you have any more concerns, please let us know. Thank you again for your detailed and constructive feedback!

---

### Review · Reviewer_wiWn · 2025-12-09

**Summary Of Contributions:**

This paper is an exhaustive survey of over five hundred recent works on Agentic RL applied to LLMs to enable them to become autonomous, decision-making agents (as opposed to just sequence generators). The paper examines the shift from traditional RL fine-tuning (e.g. RLHF) to Agentic RL through the lens of Markov Decision Processes (MDPs) and Partially Observable Markov Decision Processes (POMDPs). The survey is organized into a taxonomy in two parts: one organized by core agentic capabilities, and the other by their application across diverse domains like search, coding, mathematical reasoning, and GUI navigation. The paper does a good job of consolidating the landscape of environments and frameworks while outlining the key challenges and future directions -- including trustworthiness, scalability, and the mechanistic debate on the role of RL.

**Paper Strengths**: The main strength of the paper is its comprehensive (if a bit shallow, see below) treatment of Agentic RL applied to LLMs. The conceptual organization and grouping of topics is well-motivated and conducive to any readers interested in orienting themselves in this rapidly emerging direction.

**Paper Weaknesses**: The treatment is quite shallow on a technical level, with few -- if any -- technical details provided beyond a very basic introduction to the main building blocks (MDPs, POMDPs, PPO and variants). While the paper does a good job of *collecting* the relevant bibliographic references in one place, it does not help the reader interested in deeper technical details decide where to go next.

**Audience:**

Yes

**Audience Explanation:**

This is a **extremely** timely survey paper, one that should be of interest to anyone aiming to work on this rapidly evolving area of research. My reservations, perhaps with the exception of the somewhat shallow treatment of deeper technical issues in Section 2, are very minor.

**Broader Impact Concerns:**

The paper contains an extensive treatment of Trustworthiness issues, and the revised version has added an even more extensive section on  broader social impact which points the reader to the relevant literature.

**Claims And Evidence:**

Yes

**Claims Explanation:**

This is a survey paper, so the actual claims made in the paper are few. However, the taxonomical organization of the paper is top notch, and the overall organization aids the reader in navigating the literature.

**Requested Changes:**

These are related to the weaknesses cited above:

**Technical details in Section 2 a bit scarce and introductory**. The technical exposition limits itself to a very basic introduction to RL (MDP/POMPD/REINFORCE), with only PPO/DPO/GRPO detailed as more advanced examples of actual RL algorithms used in practice. While a comprehensive treatment of *all* variants included in Table 2 is totally infeasible, I would suggest commenting at the end of each minisection (PPO/DPO/GRPO) on the limitations of eac hvanilla algorithms and the *need* for variants in a general sense. This would also help interpret the final column of Table 2 much better for readers interested in the lower-level aspects of Agentic RL.

**Figures are somewhat shallow in content**: In particular, Figures 3, 5, and 6 -- although visually captivating -- do not really add anything to the organization of the paper beyond the textual and tabular descriptions of the same content. In my opinion it would be better to dedicate this space to visual depictions that aid the reader with the more technical aspects of the related frameworks, aspects, and interactions.

**NOTE**: The new figures added in revision (as I was working on my review) are much better in terms of technical content.

---

> ### Author Response · Authors · 2025-12-16
> **Response of Q1 to Reviewer wiWn**
>
> Q1: Technical details in Section 2 a bit scarce and introductory.
>
> A1: We thank the reviewer for this helpful suggestion. We agree that explicitly discussing the limitations of vanilla RL algorithms and the motivations for their variants improves both technical clarity and the interpretability of Table 2. In response, we have revised Section 2.7 by adding a brief concluding remark to each algorithm family (REINFORCE, PPO, DPO, GRPO), summarizing its key limitations and the general motivation for variants (e.g., high-variance gradients in REINFORCE, advantage-estimation sensitivity in PPO, dataset dependence in DPO, and noisy group-based advantages in GRPO). These additions help contextualize the variants listed in Table 2, while keeping Section 2.7 intentionally high-level and lightweight.
>
> The modified text is in orange font.

---

> ### Author Response · Authors · 2025-12-16
> **Response of Q2 to Reviewer wiWn**
>
> Q2: Figures are somewhat shallow in content
>
> A2:We thank the reviewer for this constructive suggestion and have revised the figures accordingly to make their technical roles more explicit.
>
> Figure 3. We revised the Action region (upper central) by replacing ad-hoc action descriptors with structured action categories and representative exemplars. We also reorganized the reward region (lower central) to align optimization method exemplars with the algorithm families discussed in Section 2.7.
>
> Figure 5. We augmented each capability with side panels illustrating representative mechanisms and interaction patterns, providing clearer and more interpretable visual explanations without changing the core structure.
>
> Figure 6. We updated the caption to explicitly indicate its chronological organization.
>
> The modified text is in orange font.

---

> ### Author Response · Authors · 2025-12-16
> **Thank you Reviewer wiWn!**
>
> We have revised the relevant content in the article according to your instructions, and highlighted the changes in orange font. If you have any more concerns, please let us know. Thank you again for your detailed and constructive feedback!

---

### Author Response · Authors · 2025-12-20
**Thank you for the reviewers' guidance.**

Dear TMLR editors and reviewers,
We sincerely thank you for your constructive feedback. We have revised the article based on all reviewers' comments and uploaded our responses in the latest PDF, marked in red, orange, and blue fonts.  Please let us know if you have any further questions.

---

### Decision · Action_Editor_3uaD · 2026-01-06

**Recommendation:** Accept with minor revision

**Additional Comments:**

This is a thorough survey paper, and the authors have addressed the reviewers' comments. They now simply need to upload a camera ready version.

**Audience:**

Yes

**Audience Explanation:**

The TMLR audience would surely find a survey paper on this topic useful.

**Claims And Evidence:**

Yes

**Claims Explanation:**

This is a survey paper, and the authors do a good job of providing an objective and clear evaluation of the research landscape in agentic LLMs.

---

> ### Author Response · Authors · 2026-01-14
> **Thank you to the reviewers and the editors.**
>
> We have uploaded the camera-ready version of the paper.